# The beginning of the endpoint bootstrap for conformal line defects

**Ryan A. Lanzetta,**[1,2,3] **Shang Liu**[2,4,5] **and Max A. Metlitski**[6]

[1]*Department of Physics, University of Washington, Seattle, WA 98195, USA*

[2]*Kavli Institute for Theoretical Physics, University of California, Santa Barbara, California 93106, USA*

[3]*Perimeter Institute for Theoretical Physics, Waterloo, Ontario N2L 2Y5, Canada*

[4]*Department of Physics and Institute for Quantum Information and Matter, California Institute of Technology, Pasadena, CA 91125, USA*

[5]*Institute of Physics, Chinese Academy of Sciences, Beijing 100910, China*

[6]*Department of Physics, Massachusetts Institute of Technology, Cambridge, MA 02139, USA*

*E-mail:* rlanzetta@pitp.ca, sliu.phys@gmail.com, mmetlits@mit.edu

ABSTRACT: A challenge in the study of conformal field theory (CFT) is to characterize the possible defects in specific bulk CFTs. Given the success of numerical bootstrap techniques applied to the characterization of bulk CFTs, it is desirable to develop similar tools to study conformal defects. In this work, we successfully demonstrate this possibility for *endable* conformal line defects. We achieve this by incorporating the endpoints of a conformal line defect into the numerical four-point bootstrap and exploit novel crossing symmetry relations that mix bulk and defect CFT data in a way that further possesses positivity, so that rigorous numerical bootstrap techniques are applicable. We implement this approach for the pinning field line defect of the $3d$ Ising CFT, obtaining estimates of its defect CFT data that agree well with other recent estimates, particularly those obtained via the fuzzy sphere regularization. An interesting consequence of our bounds is nearly rigorous evidence that the $\mathbb{Z}_2$-symmetric defect exhibiting long range order obtained as a direct sum of two conjugate pinning field defects is unstable to domain wall proliferation.

# 1   Introduction

Defect critical phenomena have been a subject of long-standing interest in theoretical physics [1–7]. The framework of the renormalization group (RG) applies equally well to defects as it does to bulk systems: in fact, one of the very first applications of RG ideas was to the Kondo impurity problem [2, 3]. Naturally, this induces a notion of defect universality classes characterized by possible defect and bulk fixed points and the RG flows between them, each with properties that are distinguished both qualitatively and quantitatively. Given the difficulty already in understanding bulk systems at criticality, the problem of describing defects in critical systems adds an additional layer of complexity and pushes the limits of existing theoretical tools.

The phases of matter that can be supported on $P$-dimensional defects in a given $D$-dimensional bulk are strictly richer than standalone $P$-dimensional systems since one possibility is that the bulk and defect are decoupled. Thus, phenomena that are excluded in "nice" standalone systems can potentially be exhibited on a defect embedded in a critical, higher-dimensional system. This possibility resembles a common theme of gapped, topological phases of matter. The degrees of freedom at the boundaries of these phases often exhibit phenomena that are forbidden in standalone systems with e.g. on-site symmetry action or a tensor-product decomposition of the Hilbert space, examples of systems that we might call "nice" in the context of lattice models. While the loophole for the boundaries of such gapped phases is typically that the bulk cancels some kind of anomaly, the addition of bulk gaplessness generically spoils the possibility of a conserved stress tensor on a boundary or defect, synonymous with the lack of locality.

A classic no-go result from statistical physics is that local, reflection-positive, classical lattice systems in one dimension do not order at finite temperature. For systems with a finite symmetry, the instability of the ordered phase is caused by the proliferation of domain walls [8, 9]. Upon breaking the assumption of local interactions, the possibility of an ordered phase can be restored in standalone classical one-dimensional systems [10–12]. Long-range interactions cause the energy cost of a pair of domain walls to grow with their separation, suppressing the possibility of domain wall proliferation, which can permit the existence of both low temperature ordered and high temperature disordered phases.

One of our interests in this work is to study whether ordering can occur on a one dimensional line defect in a local, higher-dimensional bulk, critical system. We will explore the possibility of an ordered line defect ($P = 1$) phase occurring in a defect conformal field theory (dCFT). For $P = 2$, there is a similar story for continuous symmetries and the corresponding no-go theorem preventing spontaneous symmetry breaking (SSB) in standalone two-dimensional systems is the Hohenberg-Mermin-Wagner-Coleman theorem [13–16]. This case has received attention in a number of recent works in the setting of plane defects and boundaries in CFTs [17–22].

Our focus will be on systems with a global $\mathbb{Z}_2$ symmetry, so naturally we will look at the Ising CFTs in various dimensions to examine the possibility of ordering on their conformal line

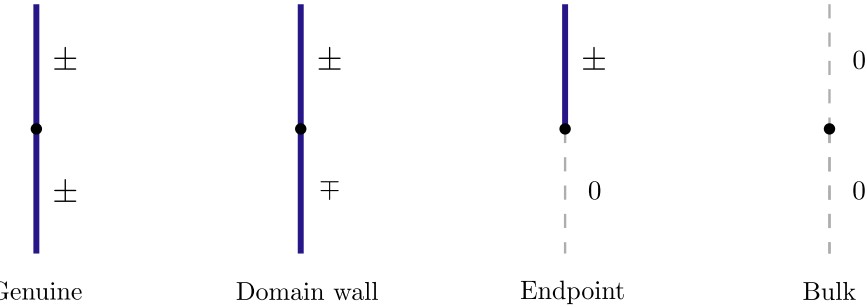

| Genuine | Domain wall | Endpoint | Bulk |

**Figure 1**: An illustration of the four types of defect-changing operators considered in this work.

defects. In order to assess this possibility, much like the analysis in classical one dimensional systems we will need to understand the effect of domain walls on a putative ordered defect fixed point. Intimately related to such ordered defect fixed points are those in which the $\mathbb{Z}_2$ symmetry is explicitly broken, which we will refer to as "pinning field" defects after [23]. Defects that explicitly break a $\mathbb{Z}_2$ symmetry come in pairs associated with a choice of sign, and are exchanged with each other upon a global application of the $\mathbb{Z}_2$ symmetry. We will call the Ising pinning field defects $\mathcal{D}^{\pm}$ depending on the choice of sign. In the Ising CFT, $\mathcal{D}^{\pm}$ can be obtained by perturbing the action with the order parameter $\sigma$ along a line. A junction where the sign of the defect changes can be considered, and this junction hosts a set of scaling operators that we will call domain wall operators (see Fig. 1), which are special cases of a general class of objects called *defect-changing operators*[1]. Whether or not the scaling dimension of the leading domain wall operator is relevant ($\Delta_0^{+-} < 1$) determines whether or not the corresponding *non-simple* fixed point $\mathcal{D}^+ \oplus \mathcal{D}^-$ is unstable, and thus whether ordering on the defect occurs without fine-tuning.[2]

In $D = 2$ and $D = 4 - \epsilon$ for $\epsilon \ll 1$ the calculation of $\Delta_0^{+-}$ can be carried out analytically. In $D = 2$, the calculation reveals that the leading domain wall is exactly marginal with $\Delta_0^{+-} = 1$ and the situation where order develops on the defect is fine-tuned, although in this case the "ordered" fixed point sits at the end of a line of dCFT fixed points continuously connected to the trivial defect [26]. In $D = 4 - \epsilon$ at the Wilson-Fisher (WF) fixed-point the leading domain wall is always relevant for small $\epsilon$, so again fine-tuning on the defect is required to display ordering. Exactly at $D = 4$, fine-tuning the bulk to the Gaussian fixed point possesses a line of "ordered" fixed points in which the scaling dimension of the domain wall can be tuned arbitrarily, although the marginally-irrelevant $\phi^4$ coupling in the bulk causes these fixed points to flow slowly to the trivial defect. For the case of $D = 3$,

---

[1]We denote defect-changing operators joining a defect $\mathcal{D}^a$ with $\mathcal{D}^b$ at a point by $\phi_i^{ab}$ where the label $i$ orders such operators by their scaling dimensions, i.e. $\Delta_i^{ab} \leq \Delta_j^{ab}$ when $i \leq j$. See section 2.1 for more details.

[2]In some microscopic realizations, the leading domain wall perturbation to the defect action can be prohibited by a combination of symmetries, such as internal $\mathbb{Z}_2$ and time-reversal, so that the long range ordered defect remains stable even if $\Delta_0^{+-} < 1$ [24, 25]. In the remainder of this paper, in our discussion of defect stability we will only assume the internal $\mathbb{Z}_2$ symmetry.

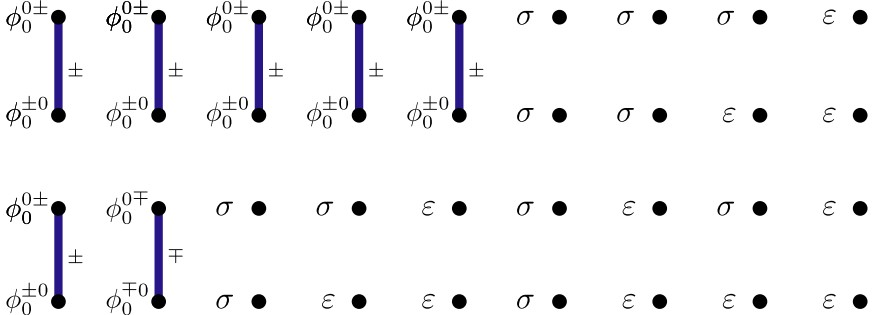

**Figure 2**: A graphical representation of the set of correlation functions involved in our numerical bootstrap calculations. We study correlation functions that involve four pinning field endpoint operators, two endpoint operators with two bulk operators, and four bulk operators. In all cases, the operators are inserted collinearly and all non-trivial line defects are straight. We suppress explicitly illustrating the trivial defect for clarity.

in this paper we will develop a novel numerical conformal bootstrap method that, subject to assumptions described below, conclusively demonstrates the instability of the long-range ordered line defect. Our results supplement recent analytical and numerical works that have addressed the question of ordering on a line defect in the $3d$ Ising CFT [27, 28], and others that have have studied the properties of the pinning field defect more broadly [23, 29–33]. Notably, the recent results from the fuzzy sphere regularization technique stand out as the highest standard of comparison for our work [27, 28, 33].

A challenge for conformal bootstrap methods has been to incorporate extended objects, thereby gaining the ability to study their properties in a way that manifestly makes use of conformal symmetry [34–37]. A number of works over the years have been devoted to this problem in the context of both boundary CFT (bCFT) and dCFT. Some works have performed bootstrap calculations using so-called bulk-to-defect/boundary crossing symmetry [18, 38–40], while others have studied the consequences of the usual crossing symmetry adapted to defect/boundary operators [40–45]. The former setup suffers from a lack of positivity, requiring either uncontrolled and non-rigorous methods [46] or non-generic positivity assumptions, and in the latter case there is very little input from the bulk CFT data, so obtaining tight bounds for specific theories can be difficult.

In this work[3], we will develop a numerical bootstrap approach to studying general line defects that includes defect-changing operators as external operators in the bootstrap calculations. In particular, since we study line defects that are *endable*, we include defect-changing operators corresponding to endpoints (i.e. defect-changing operators joining the trivial defect with the non-trivial defect in question). A correlation function involving combinations of endpoints and bulk operators admits an expansion in $SL(2, \mathbb{R})$ conformal blocks when the defect lines are straight and the operator insertions are all collinear, see Fig. 3. We apply

---

[3]Some results presented in this work were first published in the PhD thesis of RL [47].

this setup to the pinning field defect of the $3d$ Ising CFT by considering correlation functions involving the leading endpoint primaries $\phi_0^{0\pm}$ (see Fig. 1), whose scaling dimension we take to be a-priori unknown, together with the bulk operators $\sigma, \varepsilon$, whose scaling dimensions are known to very high precision [48]. We graphically enumerate these correlation functions in fig. 2. Additionally, in our setup we input scaling dimensions and OPE coefficients of higher bulk operators, which have been estimated in a number of works both with CFT techniques and more recently using the fuzzy sphere approach [49–51]. We point out that numerical bootstrap was first applied to correlation functions of line defect endpoints in [52], although in that work the bootstrap problem could be mapped to one involving local operators, partially related to the fact that the line defects studied were topological.

Altogether, our setup will provide sufficient constraining power, even with no non-trivial assumptions about the defect-changing operator spectrum of the pinning field defect beyond its RG stability ($\Delta_1^{++} \geq 1$), to isolate a small range of values of $\Delta_0^{0+}$ allowed by crossing symmetry. This range turns out to be in excellent agreement with existing numerical estimates [27, 30]. Consequently, for the other quantities that we bound as a function of $\Delta_0^{0+}$ the allowed regions include isolated islands. We especially highlight the fact that our setup allows us to put bounds on the defect $g$-function, which was first estimated for the $3d$ Ising pinning field quite recently [23], with additional numerical follow-up in a recent fuzzy sphere calculation [27].

The rest of the paper is organized as follows. After summarizing our main results in the remainder of this section, in section 2 we will review general aspects of conformal line defects that break a $\mathbb{Z}_2$ symmetry and also explain in detail our approach to studying the pinning field defect of the $3d$ Ising CFT. We will also give a more detailed overview of properties of the pinning field defect in $D = 2$ where analytical control is possible in a strongly-coupled setting. In section 3 we will present the results of our bootstrap calculations and provide details about our numerical implementation. In section 4 we use the $4 - \epsilon$ expansion to perturbatively compute properties of the pinning field defect of the Ising WF fixed point for comparison with our bootstrap results. Some of the results in this section are new, including estimates of various OPE coefficients and the dimension of the subleading endpoint primary operator, but others have already been computed elsewhere [28, 29]. Finally, in section 5 we provide concluding remarks and discuss future directions.

## 1.1 Summary of main results

### 1.1.1 Universal bounds on symmetry-breaking line defects

We begin by presenting in fig. 4 bounds that apply to general *stable* ($\Delta_1^{++} \geq 1$) $\mathbb{Z}_2$ symmetry-breaking line defects in arbitrary CFTs. The bounds are obtained by imposing unitarity and crossing symmetry on four-point functions of the leading endpoint operators $\phi_0^{0\pm}$ of the pinning field defects $\mathcal{D}^{\pm}$ (equations (B.1) and (B.2) in section B) using the techniques described in section 3. Notably, the crossing symmetry ensures that the four-point functions can be expanded in one-dimensional $SL(2,\mathbb{R})$ conformal blocks corresponding to either bulk

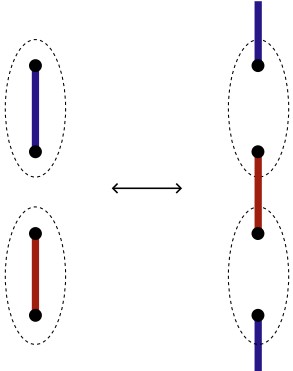

**Figure 3**: A graphical depiction of the central crossing symmetry relation exploited in this work for four-point functions of endpoints of non-trivial line defects of (possibly) distinct types (indicated by the red and blue lines). Schematically, the crossing transformation acts as a cyclic permutation of the endpoints. This leads to a direct relationship between the expansion of the four-point function in conformal blocks corresponding to the endpoints fusing either to bulk operators (left) or defect-changing operators (right).

| $D$ | 3 | 4 |
|---|---|---|
| $\Delta^{0+}_{\min}$ | 0.1209 | 0.2259 |

**Table 1**: Universal lower bound on endpoint operator scaling dimension for $\mathbb{Z}_2$ symmetry-breaking conformal line defects, in any CFT, satisfying $\Delta^{++}_1, \Delta^{+-}_0 \geq 1$ in $D = 3, 4$.

or defect-changing operators. This is how our bootstrap of dCFT data is sensitive to bulk CFT data, in contrast to previous bootstrap studies of four-point functions of local operators on line defects [41, 42, 44]. This crossing symmetry relation between the bulk and defect-changing operator expansions is illustrated schematically in fig. 3. We compute these universal bounds for CFTs in $D = 2, 3, 4$, where the spacetime dimension enters through the minimum scaling dimension allowed for bulk operators, which is set by the unitarity bound

$$\Delta \geq \frac{D-2}{2}. \tag{1.1}$$

We thus expect the bounds to become stronger as the ambient spacetime dimension increases simply because of the stronger gap assumptions.

There are a few general results that we can extract from these bounds. Recall that, roughly speaking, $g$ quantifies how many degrees of freedom are carried by a defect relative to the clean bulk: $g < 1$ indicates that the defect contains fewer degrees of freedom and vice versa. Excluding $D = 2$ where we only bound $g$ from above, we obtain non-trivial two-sided bounds on $g$ for stable, $\mathbb{Z}_2$-breaking line defects as a function of the scaling dimension of an endpoint operator $\Delta^{0+}$[4]. Via the state-operator correspondence, we can associate $\Delta^{0+}$

---

[4]In this simple calculation, it is not possible to encode the assumption that the external endpoint operator

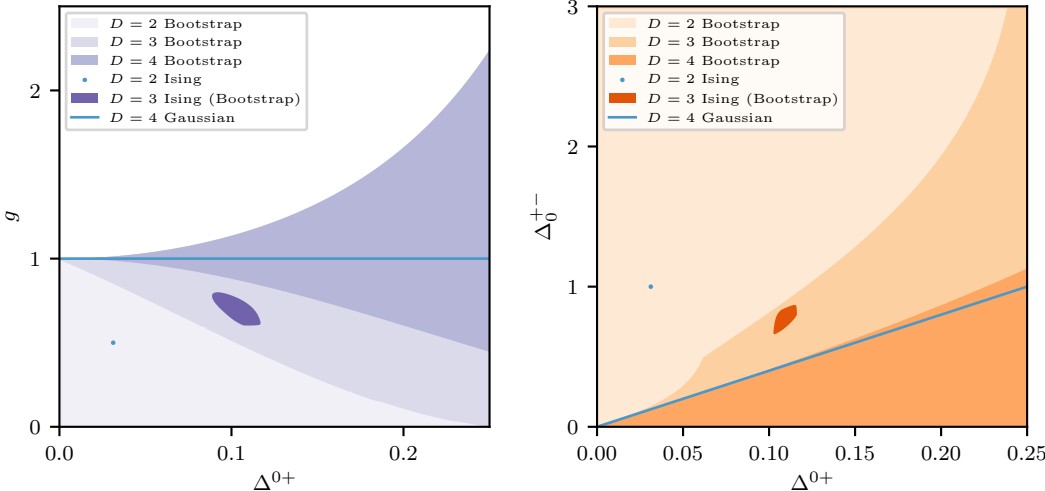

**Figure 4**: Universal bounds on stable ($\Delta_1^{++} \geq 1$) conformal line defects that explicitly break a $\mathbb{Z}_2$ symmetry in $D = 2, 3, 4$ compared with our most general bootstrap bounds for the $3d$ Ising pinning field defect. The shaded regions are allowed by bootstrap at derivative order $\Lambda = 45$. For the $D = 3$ Ising bootstrap bounds, we only plot the island expected to contain the pinning field defect created by a bulk $\sigma$ perturbation. There is a larger allowed region beyond the island that we do not show for clarity. **Left:** Universal lower and upper bounds on the defect $g$-function. For $D = 2$ we do not find a non-trivial lower bound. The $D = 2$ Ising point lies at $(\Delta_0^{0+}, g)_{2d} = (1/32, 1/2)$. In the $D = 4$ Gaussian theory $\Delta_0^{0+}$ may take any positive real value with $g_{4d} = 1$. For the $3d$ Ising bootstrap result we plot the allowed region where only $\Delta_1^{++} \geq 1$ is assumed. **Right:** Universal bounds on the leading domain wall primary operator $\Delta_0^{+-}$. In $D = 2$ Ising we have $\Delta_0^{+-} = 1$ and in the $D = 4$ Gaussian fixed point we have $\Delta_0^{+-} = 4\Delta_0^{0+}$. The $D = 3$ Ising bootstrap island is computed assuming $\Delta_1^{++} \geq 1$ and $\Delta_1^{+-} \geq 1$.

with the energy of placing a single defect on a spatial sphere $S^{D-1}$ via $E \sim \Delta^{0+}/R$, up to Casimir energy corrections from conformal anomalies in even dimensions. An interesting way to read our bounds on $g$ is to consider the inverse plot, which is a *lower* bound on $\Delta^{0+}$ as a function of $g$: what is the minimum energy cost required to add/remove degrees of freedom in a localized region in a CFT, while breaking a $\mathbb{Z}_2$ symmetry? In $D = 2$ we can only prove that it necessarily costs energy to increase $g$, while in $D > 2$ we see that tuning away from $g = 1$ in either direction always has finite energy cost. Our bound in $D = 2$ is complementary to bounds on $g$ computed by imposing open/closed duality of the annulus partition function, where it was also found that an *upper* bound on $g$ exists for stable boundary conditions (suitably interpreted using the folding trick to apply to line defects) as a function only of the

---

has the lowest scaling dimension amongst all other endpoint operators.

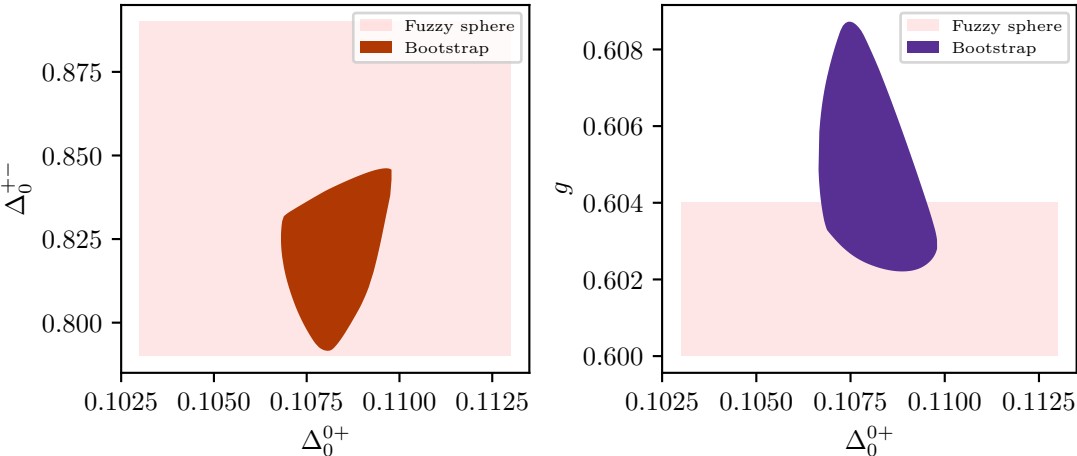

**Figure 5**: Bootstrap allowed regions for the scaling dimension of the leading domain wall primary $\Delta_0^{+-}$ and the defect $g$ function at $\Lambda = 45$, each computed as a function of the scaling dimension of the leading endpoint operator $\Delta_0^{0+}$. For comparison, we show in the light shaded regions estimates of the same quantities computed via the fuzzy sphere regularization technique [27]. The bounds are obtained using non-generic gap assumptions in the defect-changing sectors. In each bootstrap bound it is assumed that $\Delta_1^{++} \geq 1.4$, $\Delta_0^{+-} \geq 0.6$, and $\Delta_1^{0+} \geq \Delta_0^{0+} + 1.5$. The calculations use input from the OPE ratio bounds of Fig. 12, discussed in Sec. 3.4. To obtain the bound on $\Delta_0^{+-}$, we further assume that there is at most one relevant domain wall operator, i.e. $\Delta_1^{+-} \geq 1$.

bulk central charge $c$ [53][5].

We can also read off general constraints on when stable long range order (LRO) can be potentially supported on a conformal line defect via the bound on $\Delta_0^{+-}$ of Fig. 4. As we explain in more detail later, any defect that explicitly breaks a $\mathbb{Z}_2$ symmetry can be promoted to a non-simple "LRO defect". If the leading domain wall operator is relevant with $\Delta_0^{+-} < 1$, then the LRO defect is unstable and LRO is forbidden without fine-tuning. Our calculation reveals monotonically increasing upper bounds on $\Delta_0^{+-}$ as a function of the scaling dimension of an endpoint operator, except in $D = 2$ where our setup does not produce a non-trivial bound. From these bounds, we conclude that in order for stable LRO on a line defect to be possible, the dimension of any endpoint operator must satisfy the bounds from below listed in Tab. 1 for which the leading domain wall operator is not forced to be relevant.

### 1.1.2 Bounds on the 3$d$ Ising pinning field line defect

The primary results of this work are bootstrap bounds specialized to the case of the 3$d$ Ising pinning field defect. The most important among these are tight bounds on the defect $g$-function and the scaling dimension of the leading domain wall primary, plotted in Fig. 5. We obtain these by making non-generic assumptions about the defect-changing operator

---

[5]See also [54] for earlier bounds on the defect/boundary $g$-function in $D = 2$.

|  | Bootstrap bound | Bootstrap $g$-minimization | Fuzzy sphere |
|---|---|---|---|
| $g$ | 0.6056(32) | 0.60287 | 0.602(2) |
| $\Delta_0^{0+}$ | 0.1079(19) | 0.10726 | 0.108(5) |
| $\Delta_0^{+-}$ | 0.818(28) | 0.8346 | 0.84(5) |
| $\Delta_1^{++}$ | - | 1.6081 | 1.63(6) |
| $\Delta_1^{0+}$ | - | 2.2050 | 2.25(7) |
| $\Delta_1^{+-}$ | - | 3.2355 | 3.167 |
| $\Delta_2^{++}$ | - | 3.1081 | 3.12(10) |

**Table 2**: Comparison of bootstrap and fuzzy sphere predictions for the $g$-function and scaling dimensions of defect-changing operators of the $3d$ Ising CFT pinning-field defect. Our bootstrap bounds are computed with our strongest assumptions about the defect-changing operator spectrum $\Delta_0^{++} \geq 1.4$, $\Delta_0^{+-} \geq 0.6$, and $\Delta_1^{0+} \geq \Delta_0^{0+} + 1.5$. The values reported from $g$-minimization are obtained by reading off the extremal spectra resulting from minimizing $g$ at $\Delta_0^{0+} = 0.10726$. This value of $\Delta_0^{0+}$ corresponds to the value that minimizes the difference between our lower bound on $g$ with no assumptions at $\Lambda = 35$ and with the maximal assumptions at $\Lambda = 45$. The errors in our bootstrap bounds correspond with the width of the allowed region. For fuzzy sphere we quote the reported values and errors in [27, 33] except for $\Delta_1^{+-}$ which has no reported error bar. We report fuzzy sphere errors directly from [27, 33], and default to taking the results obtained directly from the energy spectrum, when available, for quantities where [27] does not provide mixed error estimates from different techniques.

spectrum $\Delta_0^{++} \geq 1.4$, $\Delta_0^{+-} \geq 0.6$, and $\Delta_1^{0+} \geq \Delta_0^{0+} + 1.5$, which are our most aggressive assumptions. We motivate these gap assumptions through a combination of self-consistency within bootstrap, existing data from other numerical methods [27, 31, 33], and the $4 - \epsilon$ expansion results reported in Ref. [23] and in the current work. For the bound on $\Delta_0^{+-}$ we additionally assume $\Delta_1^{+-} \geq 1$, i.e. there is at most a single relevant domain wall operator.

Our bound on $g$ indicates $1/2 < g < 1$, confirming that the island is consistent with the pinning field defect, which according to the $g$-theorem [55] must have $g < 1$, as it is obtained by perturbing the trivial defect with a relevant operator $\sigma$. In addition to the more rigorous evidence we show later, where fewer assumptions are made, our $\Delta_0^{+-}$ island demonstrates that the pinning field defect of the $3d$ Ising CFT does not give rise to a stable LRO defect since we show $\Delta_0^{+-} < 1$. It is natural to guess that when the $\mathcal{D}^+ \oplus \mathcal{D}^-$ defect theory is perturbed by the relevant domain wall operator it flows to the trivial defect (the latter is stable to all Ising symmetry preserving perturbations as $\Delta_\varepsilon > 1$.) Under this scenario, the $g$-theorem dictates $2g > 1$, which is again consistent with our bounds.

Note that all our claims here apply to the standard pinning field defect of the $3d$ Ising model, obtained by turning on $\sigma$ on a line, and the associated direct sum LRO defect. In principle, other $\mathbb{Z}_2$ symmetry breaking defects $\tilde{\mathcal{D}}^\pm$ in the $3d$ Ising model that are not accessible by weakly perturbing the bulk along a line could exist, although we have seen no evidence

for line defects of such a novel type in our explorations. Thus, strictly speaking, we cannot categorically rule out the existence of stable LRO line defects in the $3d$ Ising CFT.

Towards the goal of a general characterization of the $3d$ Ising pinning-field defect, we also have computed bootstrap estimates of scaling dimensions of a number of other defect-changing operators, summarized in table 2, which we report in addition to the numerical values of quantities already mentioned. We note that for $\Delta_1^{++}$ we derive an upper bound of $\Delta_1^{++} \leq 1.625$, but the bootstrap does not lead to an improvement of the lower bound beyond the lower bound assumption we make of $1.4 \leq \Delta_1^{++}$. In a similar spirit to the $c$-minimization principle for finding the $3d$ Ising CFT [56], our bounds indicate that a similar $g$-minimization principle could be reasonable for finding its pinning field defect: the associated extremal spectrum is reported in the second column of Tab. 2. In the main text, additional bounds on ratios of operator product expansion (OPE) coefficients involving the endpoint operators and bulk operators may be found in figs. 12 and 17.

**Note added:** We thank Zohar Komargodski, Fedor K. Popov, and Brandon C. Rayhaun for coordinating the arXiv submission of their related work [25] with ours.

## 2   $\mathbb{Z}_2$ symmetry-breaking conformal line defects

Here we make general remarks, from the point of view of symmetry, about conformal line defects that break a $\mathbb{Z}_2$ symmetry, either explicitly or spontaneously. We will provide a definition of spontaneous symmetry-breaking that is appropriate in the context of a line defect embedded within a gapless CFT, setting up our ability to later derive general constraints on when this is allowed using numerical bootstrap. This discussion will naturally lead us to introduce the notions of defect-changing operators and the defect $g$-function. We will finally examine the consequences of the internal and spacetime symmetries for correlation functions of defect-changing operators, deriving various selection rules that will later enter our bootstrap analysis.

We will consider throughout this work the situation where the combined bulk and defect system is described by a defect CFT (dCFT) in the IR. This situation is characterized by the partial breaking of the bulk conformal symmetry by degrees of freedom localized on the defect. For instance, when the geometry of the defect is a straight, infinite line, which we will henceforth take to be placed along the $\tau$ axis in coordinates $(\tau, x_1, ..., x_{D-1}) \in \mathbb{R}^D$, the minimal set of conformal symmetry generators required to have a dCFT describing a line defect are those generating dilatations $\hat{D}$, translations along the $\tau$ axis $\hat{P} \equiv \hat{P}^\tau$, and special conformal transformations along the $\tau$ axis $\hat{K} \equiv \hat{K}^\tau$. We will also assume invariance under the transverse rotations and reflections preserving the $\tau$ axis, which we denote $O(D-1)_T$. We denote the transverse rotation generators as $\hat{M}_{ij}$ and reflection generators as $\hat{R}_\mu$, including also reflections about a plane perpendicular to the $\tau$ axis. Altogether, this leads us to assume the spacetime symmetry group $SL^\pm(2, \mathbb{R}) \times O(D-1)_T$ where $SL^\pm(2, \mathbb{R})$ denotes the set of real $2 \times 2$ matrices $M$ with $\det M = \pm 1$. We will denote the $\hat{R}^\mu$ eigenvalues by $r^\mu = \pm 1$. We

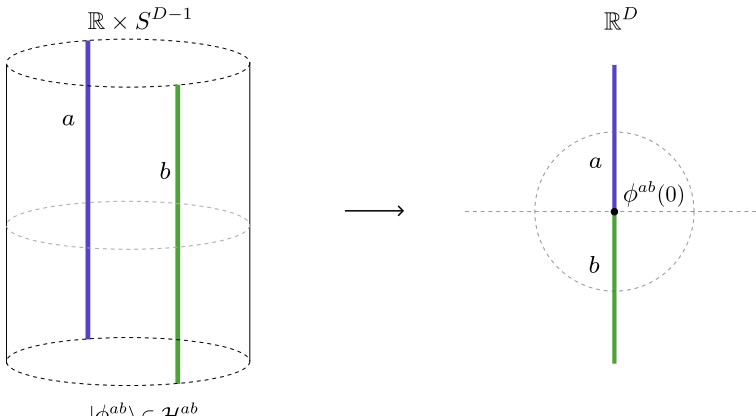

$\mathbb{R} \times S^{D-1}$ $\quad$ $\mathbb{R}^D$

$a$

$b$

$a$

$\phi^{ab}(0)$

$b$

$|\phi^{ab}\rangle \in \mathcal{H}^{ab}$

**Figure 6**: An illustration of the operator-state mapping with orientationless conformal defects.

will mainly be interested in the case $D = 3$, in which case we denote the $SO(3)$ and $SO(2)_T$ spins of bulk and defect operators living on a straight, infinite defect by $\ell$ and $s$, respectively.

## 2.1 Defect-changing operators

Let us consider a CFT placed on $\mathbb{R} \times S^{D-1}$. The CFT may contain various non-trivial line defects of distinct types $a$, which we label $\mathcal{D}^a$. In general, under a spacetime orientation-reversing transformation a line defect $\mathcal{D}^a$ can map to an orientation-reversed partner $\mathcal{D}^{\bar{a}}$, but in this work we will always have $a = \bar{a}$ corresponding to orientationless defects. We now imagine placing two, possibly distinct, conformal defects $\mathcal{D}^a, \mathcal{D}^b$ at the north and south poles of the $S^{D-1}$ that extend all throughout the $\mathbb{R}$ direction, which we choose as defining imaginary time. We denote the Hilbert space of states describing the quantization on $S^{D-1}$ with the pair of defects as $\mathcal{H}^{ab}$, which we will call the *defect-changing Hilbert space*. The usual Weyl transformation that implements the state-operator map between $\mathbb{R} \times S^{D-1}$ and $\mathbb{R}^D$ brings this defect configuration to a straight, infinite line defect in $\mathbb{R}^D$ containing a point-like junction at the origin accross which the defect type changes, illustrated in Fig. 6. We will refer to the operators hosted at these junctions as *defect-changing operators*, which we label as $\phi_i^{ab}$.

The notion of a defect-changing operator bears close resemblance with that of a boundary condition-changing operator in two dimensional theories [5, 6]. In two dimensions a defect-changing operator may be viewed as a boundary condition-changing operator via the folding trick [26, 57]. Much of the formalism used to describe correlation functions of boundary condition-changing operators in two dimensions carries over directly to our setting, since the $SL(2, \mathbb{R})$ conformal symmetry, which plays a significant role in determining the form of correlation functions involving operators on the defect, is present regardless of the ambient spacetime dimension.

Indeed, defect-changing operators may be organized into primary and descendant operators under the $SL(2, \mathbb{R})$ subgroup of the conformal symmetry preserved by the defect.

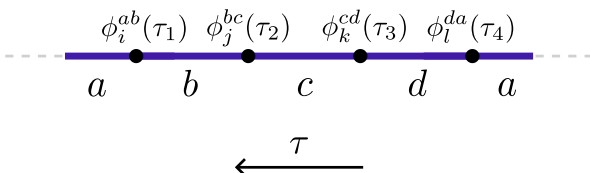

**Figure 7**: A graphical depiction of a four-point function of defect-changing operators. The imaginary time $\tau$ increases going to the left, which is consistent with the convention for time ordering for correlation functions mentioned in the main text.

Following the notation of [27], we will label the defect-changing primaries and their scaling dimensions by $\phi_i^{ab}$ and $\Delta_i^{ab}$, respectively, letting $\phi_0^{ab}$ be the operator with lowest scaling dimension in $\mathcal{H}^{ab}$ and ordering $\Delta_i^{ab} \leq \Delta_j^{ab}$ for $i < j$. We note a slight subtlety in defining the $SL(2, \mathbb{R})$ charges of defect-changing operator. The subtlety comes from the fact that performing the naive integral of the dilatation current $x_\mu T^{\mu\nu}$ around a sphere containing some primary operator will have additional UV-divergent contributions when the stress tensor insertion approaches the defect. These UV-divergent contributions do not contribute to the scaling dimension, representing instead a contribution coming from the "defect mass", so we thus *define* the defect-changing scaling dimension as the cutoff-independent piece in this scheme. In section 4.4 we will discuss this in more detail using the example of the WF fixed point near $D = 4$ .

We will from now on assume the defect-changing Hilbert spaces $\mathcal{H}^{ab}$ furnish a representation of the one-dimensional conformal symmetry algebra, generated by operators $\hat{D}, \hat{P}, \hat{K}$, acting on the defect-changing Hilbert spaces with the defect-changing operators transforming in the usual way

$$[\hat{D}, \phi_i^{ab}(\tau)] = (\tau\partial_\tau + \Delta)\phi_i^{ab}(\tau)$$
$$[\hat{P}, \phi_i^{ab}(\tau)] = \partial_\tau \phi_i^{ab}(\tau)$$
$$[\hat{K}, \phi_i^{ab}(\tau)] = (\tau^2\partial_\tau + 2\Delta\tau)\phi_i^{ab}(\tau)$$

We finally comment on our conventions for Hermitian conjugation. We will consider Hermitian conjugation in $\mathbb{R}^D$ using quantization on $\mathbb{R}^{D-1}$ planes and imaginary time evolution in the $\tau$ direction. In the correlation functions

$$\langle \phi_{i_1}^{a_1 b_1}(\tau_1)\phi_{i_2}^{a_2 b_2}(\tau_2)\cdots\rangle$$

we always assume that the $\tau$ coordinates are ordered $\tau_i \geq \tau_{i+1}$ and $b_i = a_{i+1}$, to be consistent with the graphical representation in fig. 7. For general defects (with orientation) we take Hermitian conjugation to act as

$$(\phi_i^{ab}(\tau))^\dagger = \phi_{\bar{i}}^{ba}(-\tau). \tag{2.1}$$

Note that it is always possible to choose $\bar{i} = i$, and for the case $a = b$ this corresponds to using a Hermitian basis of defect operators. This convention will be quite natural since for defect-changing operators considered in this work, all internal quantum numbers will transform

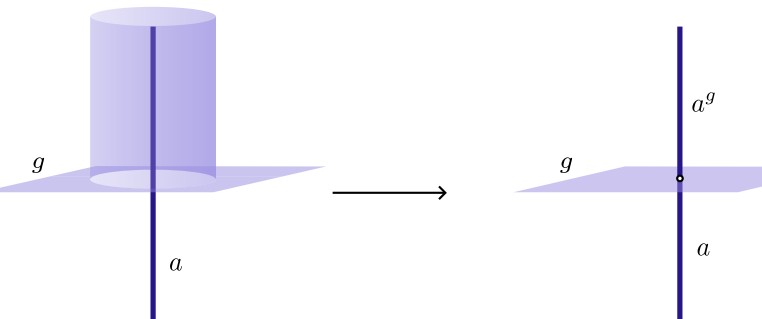

**Figure 8**: A topological surface operator for a symmetry group element $g$ that surrounds a line defect can be shrunk down onto the line $a$, implementing a symmetry transformation of the line leading to a potentially new line $a^g$. This process may also be done partially, as shown, to create a topological junction between the topological surface and the line defects $a, a^g$.

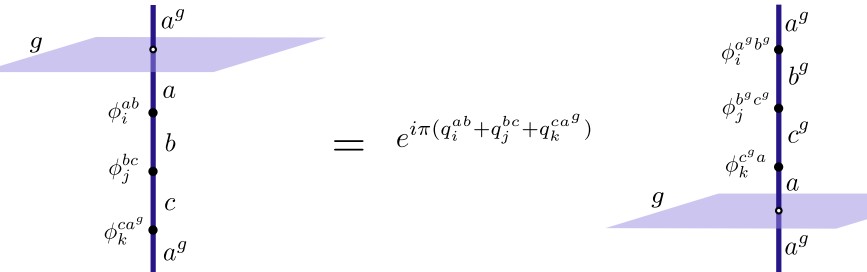

**Figure 9**: Sweeping the $\mathbb{Z}_2$ Ising topological surface defect past defect-changing operators inserted in a correlation function gives a constraint on three-point functions, illustrated here graphically.

trivially under Hermitian conjugation. In particular, only operators with $SO(2)_T$ spin $s = 0$ will be considered.

## 2.2 Explicit and spontaneous symmetry breaking on line defects

Defects generally break the symmetries of the bulk, as we saw before in the case of spacetime symmetries. Defects can also break global, internal symmetries. The particular degree of symmetry breaking gives rise to constraints on correlation functions involving defect-changing operators, as we will see. The general mathematical structure describing the interplay between non-topological defects and topological symmetry defects implementing finite symmetries is a developing subject, but for the simple case of a $\mathbb{Z}_2$ symmetry, which we study in this work, an intuitive description suffices. For recent work describing this structure in category-theoretic language, see e.g. [58, 59]. In two dimensions, one can use the mathematical language of module categories over fusion categories to describe the interplay of finite symmetries with boundaries/defects [60–63].

We first describe the consequences of global symmetry for defects that explicitly break the $\mathbb{Z}_2$ symmetry, such as the Ising pinning field defect in $2 \leq D \leq 4$. To create the pinning field defect, we can start from the clean Ising CFT and make the perturbation

$$\delta S = \int h(\tau)\, \sigma(\tau, 0)\, d\tau \qquad (2.2)$$

with $h(\tau) = h$ uniform to the action, with $\sigma$ the spin field of the Ising CFT. This perturbation is relevant in the RG sense when $D < 4$. In (2.2) the $\mathbb{Z}_2$ symmetry is explicitly broken; consequently, in the IR we may end up at one of two defect fixed points, which we call $\mathcal{D}^{\pm}$, depending on the sign of $h$.

In general, the bulk global symmetries act on the space of defects. For an invertible $G$ symmetry, we can abstractly define an action of $g \in G$ on the space of defect labels via $a^g \equiv g \cdot a$, making the space of defect labels a kind of $G$-module. The full structure needed to describe a symmetry action on a defect, however, is more sophisticated than a $G$-module since it contains additional information about properties of junctions between the topological and non-topological defects. It is interesting to note that, in the event that $a^g = a$, the symmetry must act trivially on the defect as an object that can be inserted in a correlation function $g : \mathcal{D}^a \to \mathcal{D}^a$ whenever its defect $g$-function obeys $g_a > 0$.

For the specific case of the pinning field defect, we have two defect labels $\pm$ which are mapped to each other by the generator $g$ of the $\mathbb{Z}_2$ symmetry $\pm^g = \mp$. We also will consider the trivial defect label 0, which obviously obeys $0^g = 0$. We can consider defect configurations where $h(\tau)$ is either uniform or non-uniform, and the corresponding defect-changing primary operators can either be bulk operators, genuine defect operators, domain walls or endpoints, illustrated in Fig. 1. Accordingly, the defect-changing operators transform under $\mathbb{Z}_2$ as $\phi_i^{ab} \mapsto e^{i\pi q_i^{ab}} \phi_i^{a^g b^g}$, illustrated in Fig. 9 for a case of multiple defect-changing operator insertions. When $(a, b) \neq (a^g, b^g)$ the quantities $q_i^{ab}$ represent the freedom to perform a change of basis by an overall phase, and otherwise $q_i^{ab}$ is physically meaningful and represents the $\mathbb{Z}_2$ charge. A slight exception to this are domain wall operators. For a domain wall operator, we can consider the combined action of the $R^\tau$ reflection and internal $\mathbb{Z}_2$ symmetry. We can similarly take the $\hat{R}^\tau$ reflection symmetry to act as $\phi_i^{ab} \mapsto e^{i\pi r_i^{ab}} \phi_i^{ba}$. Since applying this combination twice must do nothing[6] we conclude that $r_i^{\pm\mp} + q_i^{\mp\pm} = 0, 1 \mod 2$, representing an unambiguous conserved charge. In the cases we will consider, however, operators with a non-trivial value of this charge in the domain wall channel may not appear so we will typically set both $q_i^{ab}, r_i^{ab} = 0$ whenever $a$ or $b$ are non-trivial.

So far we have discussed properties of line defects that explicitly break the $\mathbb{Z}_2$ symmetry. We will see now that the construction introduced so far naturally leads to a notion of spontaneous symmetry breaking for line defects. We will first discuss the properties that we

---

[6] We assume that $\mathbb{Z}_2$ and $R_\tau$ symmetries commute and square to one on all defect changing operators. Such a symmetry implementation is certainly possible in the Ising CFT. In particular, this allows us to work in a defect changing operator basis where both symmetries do not act on any internal quantum numbers other than the defect labels.

expect for such a line defect, and then show how we may construct such a defect via a defect that explicitly breaks the symmetry.

Our starting assumption is that we have a bulk CFT with an unbroken $\mathbb{Z}_2$ symmetry. We will also assume the theory possesses a line defect $\mathcal{D}^l$ with long-range order (LRO) of the $\mathbb{Z}_2$ symmetry, in a sense that we will describe. We call this kind of defect an *LRO defect* without regard to its stability.

First, the combined bulk and defect system still preserves the $\mathbb{Z}_2$ symmetry, implying for instance that all $\mathbb{Z}_2$-odd local operators have a vanishing expectation value with or without $\mathcal{D}^l$. For there to be LRO of the $\mathbb{Z}_2$ symmetry, there necessarily exists some $\mathbb{Z}_2$-odd defect primary operator $\phi_{0,-}^{ll}$ whose two-point function obeys

$$\lim_{\tau \to \infty} \langle \phi_{0,-}^{ll}(\tau) \phi_{0,-}^{ll}(0) \rangle \sim \text{const.} \tag{2.3}$$

$\phi_{0,-}^{ll}$ thus represents the order parameter on the defect. Consistency with scale invariance dictates that $\Delta_{0,-}^{ll} = 0$, implying that it is a topological local operator. This additional topological local operator, which is distinct from the trivial operator since it carries $\mathbb{Z}_2$ charge, indicates that $\mathcal{D}^l$ is *non-simple*. A consequence of the non-simplicity and the additional non-trivial operator with $\Delta = 0$ is a twofold degeneracy of scaling dimensions of every operator on $\mathcal{D}^l$, since fusing any operator on $\mathcal{D}^l$ with $\phi_{0,-}^{ll}$ will give a distinct operator with the opposite $\mathbb{Z}_2$ charge. The presence of $\phi_{0,-}^{ll}$ will also generally lead bulk $\mathbb{Z}_2$-odd operators to acquire LRO of their connected two-point functions when measured parallel to the defect, which can be seen by noting that $\phi_{0,-}^{ll}$ would generally give the leading contribution to the defect operator expansion (DOE) of such an operator.

To discuss the stability of LRO defect fixed points, let us now explain how to describe the excitations of the LRO defect $\mathcal{D}^l$ in terms of defect-changing operators introduced before in the context of defects that explicitly break $\mathbb{Z}_2$. Consider again the radial quantization picture on the spacetime $\mathbb{R} \times S^{D-1}$. In a phase where a $\mathbb{Z}_2$ symmetry is spontaneously broken, an infinitesimal perturbation of the system that explicitly breaks the symmetry will drive the system towards one of the pinning field defects $\mathcal{D}^{\pm}$. We may then write $\mathcal{D}^l = \mathcal{D}^+ \oplus \mathcal{D}^-$. On $\mathbb{R} \times S^{D-1}$, the Hilbert space decomposes as

$$\mathcal{H} = \mathcal{H}^{++} \oplus \mathcal{H}^{+-} \oplus \mathcal{H}^{-+} \oplus \mathcal{H}^{--} \tag{2.4}$$

representing the choice of how we pin the sign of the defect at the north and south poles.

For a LRO defect, the states in the $\mathcal{H}^{++}$, $\mathcal{H}^{--}$, $\mathcal{H}^{+-}$, $\mathcal{H}^{-+}$ sectors correspond to genuine local operators, whereas for $\mathcal{D}^+$ ($\mathcal{D}^-$) the local operators are captured only by $\mathcal{H}^{++}$ ($\mathcal{H}^{--}$) and those in $\mathcal{H}^{+-}$, $\mathcal{H}^{-+}$ should be thought of as point-like defects that may only be generated by hand. Conversely, this suggests that if we are given any defect with explicitly-broken $\mathbb{Z}_2$ symmetry we may formally construct a LRO defect by, for instance, coupling the sign of the pinning field to a dynamical spin-1/2 living on the defect line.

The question now is whether this way of constructing a LRO defect is stable. Since for the non-simple LRO defect both defect and domain-wall excitations are allowed, stability

requires that the lightest non-trivial operator in either sector must be irrelevant with scaling dimension $\Delta > 1$.

## 2.3 Defect $g$-function

A fundamental quantity characterizing a conformal line defect is the value of its defect $g$-function [64], which can be defined as the vacuum expectation value of the defect on an $S^D$ spacetime where the defect is placed on an equator of $S^D$

$$g_a \equiv Z_{S^D}^a / Z_{S^D}^0, \tag{2.5}$$

with $Z_{S^D}^a$ denoting the partition function of the theory in the presence of the defect $\mathcal{D}^a$. Upon subtracting off the cosmological constant ambiguity, this quantity has been shown to be a RG monotone in $D \geq 2$ [55, 65, 66], and may be identified with the quantum dimension in the case when $\mathcal{D}^a$ is topological.

In defining $g$, or more general correlation functions of defect-changing operators, on an infinite defect line in $\mathbb{R}^D$, we must choose how to regulate the defect at infinity. If we demand the special conformal transformation generated by $\hat{K}_\tau$ to be a symmetry, as we will, we implicitly assume that correlation functions are constructed by inserting defect-changing operators on a small arc of the equator of the sphere $S^D$ and sending the sphere radius $R \to \infty$. In particular, this procedure leads us to a definition of $g_a$ that holds in flat space as the expectation value of the identity primary operator on the defect

$$g_a = \langle I^{aa} \rangle \tag{2.6}$$

where we use $I^{aa}$ to denote the identity operator in the defect Hilbert space $\mathcal{H}^{aa}$[7]. Other choices of IR regulator may give different answers for the expectation value of an infinite line defect in flat space, as has been demonstrated in various examples [67, 68]. All correlation functions of defect changing operators will be defined with respect to the defect-free bulk partition function, as in Eq. (2.5).

## 2.4 Correlation functions of defect-changing operators

Again, we will work in flat space with all defects placed along the $\tau$ axis in $\mathbb{R}^D$. Correlation functions of defect-changing operators may be described using very similar formalism to the $D = 2$ case of correlation functions of boundary condition changing operators [35, 69], except that in higher dimensions there is no Virasoro symmetry. As with local operators, there is a notion of an operator product expansion (OPE) of defect-changing operators. Let us consider a configuration where two defect-changing operators $\phi_i^{ab}$, $\phi_j^{bc}$ are inserted at nearby points, contained within a sphere without other operator insertions. At long distances, this creates a new operator in the $\mathcal{H}^{ac}$ defect Hilbert space. The OPE may be expressed as a sum over

---

[7]Sometimes we will also use $\phi_0^{aa}$ to refer to the identity primary.

primary and descendant operators of the form (assuming $\tau > 0$)[8]

$$\phi_i^{ab}(\tau) \times \phi_j^{bc}(0) = \delta_{ij}\delta_{ac}\alpha_i^{ab}\tau^{-2\Delta_i^{ab}}I^{aa} + \sum_{\substack{k \\ \phi_k^{ac} \neq I^{aa}}} \lambda_{kij}^{abc}\tau^{\Delta_k^{ac}-\Delta_i^{ab}-\Delta_j^{bc}}\sum_{n=0}^{\infty}\beta_{kij}^{abc,(n)}\tau^n\partial_\tau^n\phi_k^{ac}(0),$$

(2.7)

where the coefficients $\beta_{kij}^{abc,(n)}$ are fixed by $SL(2,\mathbb{R})$ symmetry to be[9]

$$\beta_{kij}^{abc,(n)} = \frac{(\Delta_k^{ac} + \Delta_i^{ab} - \Delta_j^{bc})_n}{n!\,(2\Delta_{\mathcal{O}})_n}.$$

(2.8)

One issue we must take care of is the normalization of defect-changing operators. From (2.6) and (2.7), a general two-point function of defect-changing primary operators takes the form

$$\langle\phi_i^{ab}(\tau_1)\phi_i^{ba}(\tau_2)\rangle = \frac{g_a\alpha_i^{ab}}{\tau_{12}^{2\Delta_i^{ab}}}.$$

(2.9)

where $\tau_{12} = \tau_1 - \tau_2$ and we assume $\tau_1 > \tau_2$. As noted before, in general, we will use a $\tau$-ordering convention inside correlation functions. Next we observe that the $SL(2,\mathbb{R})$ conformal symmetry may be used to cyclically permute the operators inside a correlation function, which allows us to swap the location of the operators in (2.9), leading to the relation

$$g_a\alpha_i^{ab} = g_b\alpha_i^{ba}$$

(2.10)

We may rescale $\phi_i^{ab}$ and $\phi_i^{ba}$ by e.g. $(g_a\alpha_i^{ab})^{-1/2}$, so that for non-trivial defect changing operators,

$$\langle\phi_i^{ab}(\tau_1)\phi_i^{ba}(\tau_2)\rangle = \frac{1}{\tau_{12}^{2\Delta_i^{ab}}}$$

(2.11)

and[10]

$$\phi_i^{ab}(\tau) \times \phi_j^{bc}(0) = \frac{\delta_{ij}\delta_{ac}}{g_a}\tau^{-2\Delta_i^{ab}}I^{aa} + \sum_{\substack{k \\ \phi_k^{ac} \neq I^{aa}}} \lambda_{kij}^{abc}\tau^{\Delta_k^{ac}-\Delta_i^{ab}-\Delta_j^{bc}}\sum_{n=0}^{\infty}\beta_{kij}^{abc,(n)}\tau^n\partial_\tau^n\phi_k^{ac}(0).$$

(2.12)

Continuing, a general 3-point function involving non-trivial defect-changing operators takes the form

$$\langle\phi_i^{ab}(\tau_1)\phi_j^{bc}(\tau_2)\phi_k^{ca}(\tau_3)\rangle = \frac{\lambda_{ijk}^{bca}}{\tau_{12}^{\Delta_{ijk}^{bca}}\tau_{13}^{\Delta_{ikj}^{abc}}\tau_{23}^{\Delta_{jki}^{cab}}} \qquad \Delta_{ijk}^{bca} = \Delta_i^{ab} + \Delta_j^{bc} - \Delta_k^{ca}.$$

(2.13)

---

[8]Note here we do not demand unit-normalization of the operators appearing on the right-hand side.

[9]$(x)_n$ denotes the Pochhammer symbol.

[10]Note that the OPE coefficients $\lambda_{kij}^{abc}$ would also be affected by the rescaling, but we will let this be implicit.

Using again the $SL(2, \mathbb{R})$ symmetry to cyclically permute the operators in (2.13) leads to the relations

$$\lambda_{ijk}^{bca} = \lambda_{jki}^{cab} = \lambda_{kij}^{abc}. \tag{2.14}$$

At this point it is also appropriate to mention the constraint coming from the $\hat{R}^\tau$ reflection symmetry. We assume that the $\hat{R}^\tau$ reflection leads to a relation

$$\langle \phi_i^{ab}(\tau_1)\phi_j^{bc}(\tau_2)\phi_k^{ca}(\tau_3)\rangle = e^{i\pi\left(r_i^{ab}+r_j^{bc}+r_k^{ca}\right)}\langle \phi_k^{ac}(-\tau_3)\phi_j^{cb}(-\tau_2)\phi_i^{ba}(-\tau_1)\rangle \tag{2.15}$$

for three-point functions. An operator $\phi_i^{ab}(0)$ cannot be associated with a definite $\hat{R}^\tau$ eigenvalue since the reflection acts as a map between different defect Hilbert spaces and $r_i^{ab}$ reflects an ambiguity in the choice of basis, except when $a = b$ and $e^{i\pi r_i^{aa}}$ is physically meaningful. The fact that $(\hat{R}^\tau)^2 = I$ leads to the relation

$$r_i^{ab} + r_i^{ba} = 0 \mod 2, \tag{2.16}$$

but we may simply set $r_i^{ab} = 0$ for $a \neq b$. These considerations, combined with those following from Hermitian conjugation, yield

$$\lambda_{ijk}^{bca} = e^{i\pi(r_i^{ab}+r_j^{bc}+r_k^{ca})}\lambda_{kji}^{cba} = (\lambda_{kji}^{cba})^*.$$

We also can impose the constraints coming from the $\mathbb{Z}_2$ Ising symmetry discussed in section 2.2 for the specific case of the pinning field defect, which lead to

$$\lambda_{ijk}^{\pm\pm\pm} = \lambda_{ijk}^{\mp\mp\mp} \qquad\qquad \lambda_{ijk}^{\pm\mp\pm} = e^{i\pi(q_j^{\pm\mp}+q_k^{\mp\pm})}\lambda_{ijk}^{\mp\pm\mp}$$

$$\lambda_{ijk}^{\pm 0\pm} = \lambda_{ijk}^{\mp 0\mp} \qquad\qquad \lambda_{ijk}^{\pm 0\mp} = e^{i\pi q_i^{\mp\pm}}\lambda_{ijk}^{\mp 0\pm}$$

$$\lambda_{ijk}^{0\pm 0} = e^{i\pi q_i^{00}}\lambda_{ijk}^{0\mp 0} \qquad\qquad \lambda_{ijk}^{000} = e^{i\pi(q_i^{00}+q_j^{00}+q_k^{00})}\lambda_{ijk}^{000}$$

and permutations thereof, which follow from the relation illustrated in Fig. 9. These relations will enter later when we derive the crossing equations. A consequence of the above OPE relations is that the OPE of the endpoints may contain both $\mathbb{Z}_2$-even and $\mathbb{Z}_2$-odd bulk operators, which is expected since the defect explicitly breaks the $\mathbb{Z}_2$ symmetry.

With these relations we are now prepared to discuss four-point functions, which are the main objects of study in our bootstrap calculations. A four-point function of defect-changing operators, as dictated by the $SL(2, \mathbb{R})$ symmetry, takes the general form

$$\langle \phi_i^{ab}(\tau_1)\phi_j^{bc}(\tau_2)\phi_k^{cd}(\tau_3)\phi_l^{da}(\tau_4)\rangle = \left(\frac{\tau_{24}}{\tau_{14}}\right)^{\Delta_{ij}^{abc}}\left(\frac{\tau_{14}}{\tau_{13}}\right)^{\Delta_{kl}^{cda}}\frac{G_{ijkl}^{abcd}(x)}{\tau_{12}^{\Delta_i^{ab}+\Delta_j^{bc}}\tau_{34}^{\Delta_k^{cd}+\Delta_l^{da}}} \tag{2.17}$$

where $x = \frac{\tau_{12}\tau_{34}}{\tau_{13}\tau_{24}}$, $\Delta_{ij}^{abc} = \Delta_i^{ab} - \Delta_j^{bc}$, and

$$G_{ijkl}^{abcd}(x) = \frac{1}{g_a}\delta_{ac}\delta_{ij}\delta_{kl} + \sum_{\substack{m \\ \phi_m^{ca}\neq I^{aa}}} \lambda_{ijm}^{bca}\lambda_{mkl}^{cda} f_{\Delta_m^{ca}}^{\Delta_{ij}^{abc},\Delta_{kl}^{cda}}(x). \tag{2.18}$$

The functions $f_\Delta^{\Delta_{12},\Delta_{34}}(x)$ are $SL(2,\mathbb{R})$ conformal blocks, given by

$$f_\Delta^{\Delta_{12},\Delta_{34}}(x) = x^\Delta \,_2F_1(\Delta - \Delta_{12}, \Delta + \Delta_{34}, 2\Delta, x).$$

Finally, four-point functions of defect-changing operators are subject to crossing symmetry, which again amounts to a cyclic permutation of the operator labels in addition to a change in the cross-ratio $x \to 1 - x$. Explicitly, crossing symmetry becomes the statement that

$$(1-x)^{\Delta_j^{bc}+\Delta_k^{cd}} G_{ijkl}^{abcd}(x) = x^{\Delta_i^{ab}+\Delta_j^{bc}} G_{lijk}^{dabc}(1-x). \tag{2.19}$$

We define for later convenience a modified set of conformal blocks that have definite crossing parity

$$F_{\Delta,\pm}^{\Delta_i^{ab}\Delta_j^{bc}\Delta_k^{cd}\Delta_l^{da}}(x) = (1-x)^{\Delta_j^{bc}+\Delta_k^{cd}} f_\Delta^{\Delta_{ij}^{abc},\Delta_{kl}^{cda}}(x) \pm x^{\Delta_j^{bc}+\Delta_k^{cd}} f_\Delta^{\Delta_{ij}^{abc},\Delta_{kl}^{cda}}(1-x). \tag{2.20}$$

In terms of the blocks in (2.20), the statement of crossing symmetry becomes

$$0 = \frac{1}{g_a}\delta_{ac}\delta_{ij}\delta_{kl} F_{0,\mp}^{\Delta_i^{ab}\Delta_j^{bc}\Delta_k^{cd}\Delta_l^{da}}(x) \pm \frac{1}{g_b}\delta_{bd}\delta_{il}\delta_{jk} F_{0,\mp}^{\Delta_l^{da}\Delta_i^{ab}\Delta_j^{bc}\Delta_k^{cd}}(x)$$

$$+ \sum_{\substack{m \\ \phi_m^{ca}\neq I^{aa}}} \lambda_{ijm}^{bca}\lambda_{mkl}^{cda}\, F_{\Delta_m^{ca},\mp}^{\Delta_i^{ab}\Delta_j^{bc}\Delta_k^{cd}\Delta_l^{da}}(x) \pm \sum_{\substack{m \\ \phi_m^{bd}\neq I^{bb}}} \lambda_{lim}^{abd}\lambda_{mjk}^{bcd}\, F_{\Delta_m^{db},\mp}^{\Delta_l^{da}\Delta_i^{ab}\Delta_j^{bc}\Delta_k^{cd}}(x). \tag{2.21}$$

From (2.21), it is clear why the $g$-function has not made it into prior bootstrap studies involving correlation functions of defect operators, since if line defects of only a single type are considered then multiplying the above expression by $g$ will lead to a crossing equation with a conventional contribution from the identity operator. With the introduction of multiple defect types, where defect-changing operators become involved, the $g_a$ can enter and we may constrain their physically allowed values, as we later demonstrate.

## 2.5 Bulk descendants as $SL(2,\mathbb{R})$ primaries

We primarily consider correlation functions of operators as constrained by $SL(2,\mathbb{R})$ conformal symmetry. However, the complete bulk conformal symmetry leads to additional constraints that can be incorporated for bulk local operators, which we now discuss.

First we discuss how to classify bulk $SL(2,\mathbb{R})$ primary operators by their $\hat{R}^\tau$ parity. In some instances such operators will be descendant operators under the full conformal symmetry of the bulk, so their $\hat{R}^\tau$ eigenvalue can differ from their parent primary operator. This analysis will later help us to decide in which OPE channels bulk descendant operators that are $SL(2,\mathbb{R})$ primaries may appear, which will be especially important when we explicitly incorporate known bulk operators of the 3$d$ Ising CFT in our bootstrap calculations. In the analysis below, we will only discuss operators with $SO(2)_T$ spin $s = 0$ that are even also under the transverse reflection symmetries, which will be the only types of operators entering out later bootstrap study.

Consider a bulk spin-$\ell$ primary operator $\mathcal{O}_a$, with $a$ representing its $SO(3)$ indices, parity $p$, and scaling dimension $\Delta$. A $\hat{R}^\tau$ transformation acts as

$$\hat{R}^\tau : \mathcal{O}_a(\tau, \vec{x}) \to (-1)^{p+n_\tau} \mathcal{O}_a(-\tau, \vec{x}),$$

where we use $n_\tau$ to denote the number of components set to $\tau$, giving $r^\tau = p + n_\tau$. Tracelessness and the requirement of being an $SO(2)_T$ singlet implies that at bulk descendant level zero $\mathcal{O}_a$ appears only with e.g. all its free indices set to $\tau$. Note that for the remainder of this discussion we will only consider parity-even ($p = 0$) bulk operators. Our ultimate strategy is to explicitly incorporate the scaling dimensions and OPE coefficients of low-lying bulk primaries and their descendants, and current evidence suggests that no parity-odd primaries appear below our cutoffs [51]. One can on general grounds rule out bulk pseudoscalar and pseudovector primaries and their descendants from appearing in the OPEs we consider, but (descendants of) higher-spin pseudotensor operators can in principle be allowed in the OPEs of endpoint operators fusing to bulk operators.

There are a few ways that we can generate $SL(2, \mathbb{R})$ primaries that are bulk conformal descendant operators. First let us assume that $\mathcal{O}_a$ is a scalar operator $\mathcal{O}$ with $\ell = 0$ and scaling dimension $\Delta$. In this case, we may construct $SO(2)_T$ singlet operators that are $SL(2, \mathbb{R})$ primaries by considering operators of the form

$$\tilde{\mathcal{O}}^{(2N),i}(x) = \sum_{n=0}^{N} a_n^i (\partial_\tau)^{2n} (\partial_j \partial^j)^{N-n} \mathcal{O}(x). \tag{2.22}$$

Such operators have scaling dimensions $\Delta + 2N$ and $r^\tau = 0$. It is not possible to obtain an $SL(2, \mathbb{R})$ primary at odd descendant level in the conformal tower of $\mathcal{O}$ that satisfies all of our symmetry requirements, so we will not need to consider $SL(2, \mathbb{R})$ primaries with $r^\tau = 1$ that are bulk descendants of $\mathcal{O}$. If $\mathcal{O}_a$ has $\ell > 0$, it is possible again to find $SL(2, \mathbb{R})$ primaries amongst the bulk descendant operators using the same construction as in (2.22) with scaling dimensions $\Delta + 2N$, in addition to others that come from contracting an even number of derivatives with the $SO(3)$ indices of $\mathcal{O}_a$ in a way that preserves $SO(2)_T$ invariance. All such operators will have $r^\tau = \ell \mod 2$. When $\ell > 0$, it is also possible to construct descendant operators that are $SL(2, \mathbb{R})$ primaries with $r^\tau = \ell + 1 \mod 2$ while still maintaining the other symmetry requirements. This can be done, for instance, by contracting an odd number of derivatives with the $SO(3)$ indices of $\mathcal{O}_a$ in an $SO(2)_T$-invariant way. This leads to operators with scaling dimensions $\Delta + 2N + 1$ and $r^\tau = \ell + 1 \mod 2$.

To summarize, we see that for spin-$\ell$ traceless symmetric tensor operators $\mathcal{O}_a$ with $\mathbb{Z}_2$ charge $q$ we have

$$\mathfrak{B}^{q,0} \ni \begin{cases} \partial^{2N} \mathcal{O}_a, & \ell \in 2\mathbb{Z} \\ \partial^{2N+1} \mathcal{O}_a, & \ell \in 2\mathbb{Z} + 1 \end{cases} \qquad \mathfrak{B}^{q,1} \ni \begin{cases} \partial^{2N} \mathcal{O}_a, & \ell \in 2\mathbb{Z} + 1 \\ \partial^{2N+1} \mathcal{O}_a, & 0 < \ell \in 2\mathbb{Z} \end{cases}$$

where we use the shorthand $\partial^n \mathcal{O}_a$ to denote an arbitrary $SL(2, \mathbb{R})$ primary at bulk descendant level $n$ in the conformal tower of $\mathcal{O}_a$.

In the above discussion, we have only treated operators as being primary with respect to the residual $SL(2,\mathbb{R})$ conformal symmetry preserved by the defect. However, an essential ingredient that connects the bulk and defect together in our bootstrap calculations is to include four-point functions of defect-changing operators together with bulk operators. But the OPE of bulk operators $\mathcal{O}_a, \mathcal{O}_b$ is constrained by the full $SO(D+1,1)$ symmetry

$$\mathcal{O}_i(x) \times \mathcal{O}_j(0) = \sum_k C^{3d}_{kij}(x, \partial_\mu)\mathcal{O}_k(0) \tag{2.23}$$

where $\mathcal{O}_k$ is a bulk primary operator and $C^{3d}_{kij}(x, \partial_\mu)$ is a differential operator which exactly determines the contributions of bulk descendants of $\mathcal{O}_k$ given the OPE coefficient $\lambda_{kij}$. On the other hand, when a pair of endpoint operators fuse to bulk operators, the OPE coefficients of bulk descendant operators that are $SL(2,\mathbb{R})$ primaries are no longer fixed in terms of the OPE coefficient involving the corresponding bulk primary.

In the limit where all operator insertions are along the $\tau$ axis, we can make use of the OPE of the form (2.7) for the special case of bulk operators. This expression contains less structure than (2.23) but, as we will see, is required in order to to consider a mixed bootstrap system of endpoint operators and bulk operators. The fact that some bulk descendant operators can be primary with respect to $SL(2,\mathbb{R})$ means the OPE coefficients of such primaries appearing in (2.7) will be fixed in terms of the OPE coefficients $\lambda_{kij}$, and we can make use of this information to give stronger bounds in bootstrap.

To do this, consider imposing crossing symmetry on a general set of four-point functions of bulk and defect-changing operators of the form (2.21). For this discussion, assume that at least some of the chosen four-point functions involve external defect endpoint operators and internal bulk operators. These equations can be expressed in a compact form as

$$0 = \sum_\mathcal{O} \mathrm{Tr}(P_\mathcal{O} \mathbf{V}_{\Delta_\mathcal{O}}(x)) + \text{defect-changing}, \tag{2.24}$$

where the sum runs over exchanged bulk $SL(2,\mathbb{R})$ primaries $\mathcal{O}$. We define the OPE matrices $P_\mathcal{O} = \vec{\lambda}_\mathcal{O} \cdot \vec{\lambda}_\mathcal{O}^\mathsf{T}$ with $\vec{\lambda}_\mathcal{O}$ a vector containing OPE coefficients involving $\mathcal{O}$, and $\mathbf{V}_{\Delta_\mathcal{O}}(x)$ are crossing vectors with matrix-valued entries that contain the contributions of conformal blocks. The crossing vectors are used directly as input in numerical bootstrap calculations.

Next, for bulk conformal primary operators $\mathcal{O}_p, \mathcal{O}_q, \mathcal{O}_r$ define

$$\alpha^{(n),i}_{\mathcal{O}_r \mathcal{O}_p \mathcal{O}_q} \equiv \frac{\lambda_{\tilde{\mathcal{O}}_r^{(n),i} \mathcal{O}_p \mathcal{O}_q}}{\lambda_{\mathcal{O}_r \mathcal{O}_p \mathcal{O}_q}}, \tag{2.25}$$

which depend entirely on the scaling dimensions and $SO(3)$ spins of $\mathcal{O}_p, \mathcal{O}_q, \mathcal{O}_r$ at any given level. These relations can be substituted into $P_{\tilde{\mathcal{O}}^{(n),i}}$, which then means that some of its OPE coefficients—excluding any involving non-trivial defect-changing operators—will be shared with those contained in $P_\mathcal{O}$. We can finally couple the contributions of $\mathcal{O}$ and its descendants that are $SL(2,\mathbb{R})$ primaries into a single crossing vector. To do this, choose some maximum

descendant level $K$ and, for each $\mathcal{O}$ as characterized by its scaling dimension and spin, solve

$$\text{Tr}\,(Q_{\mathcal{O}}\mathbf{W}_{\mathcal{O}}(x)) = \sum_{n=0}^{K}\sum_{i}\text{Tr}\,(P_{\mathcal{O}^{(n),i}}\mathbf{V}_{\mathcal{O}^{(n),i}}(x)) \tag{2.26}$$

for $\mathbf{W}_{\mathcal{O}}(x)$, where $Q_{\mathcal{O}}$ is the OPE matrix generated by the OPE vector of the remaining *independent* OPE coefficients involving descendants of $\mathcal{O}$ after imposing the relations (2.25). When (2.26) is substituted into (2.24) the contributions of bulk primaries and their descendants become non-trivially intertwined. Note that we must always choose $K$ to be finite in the above procedure, since the dimension of the OPE matrix $Q_{\mathcal{O}}$ grows with increasing $K$.

From this discussion we see why it is not possible, for numerical bootstrap purposes, to allow four-point functions involving only bulk operators to be taken away from the collinear limit at this stage. This would involve using full $3d$ conformal blocks, which automatically sum up the $SL(2,\mathbb{R})$ primaries that, as we showed, in this problem must be carefully intertwined with their contributions to the OPE of endpoint operators. However, the $3d$ conformal block decomposition can be used as long as the purely bulk four-point functions are included twice, once as in the above discussion using the decomposition into $SL(2,\mathbb{R})$ primaries and again using the full $SO(4,1)$ primary decomposition; the two decompositions would be intertwined through the discrete part of the spectrum. It is easy to see then that this would lead to a setup in which it is possible to achieve violations of crossing symmetry with semidefinite programming. This would generate additional computational cost, since one will then need to account for the different $SO(3)$ spin sectors, but this would make the sensitivity to the bulk operator spectrum at least as strong as the standard $3d$ bootstrap of local operators, which could be highly desirable in the future.

For simplicity, we will take into account knowledge of the descendant OPE coefficients only for level-two descendant operators of bulk scalar primaries in our work. Our first task is to determine, for a given bulk, scalar primary, what are the corresponding level-two $SL(2,\mathbb{R})$ primaries. Since ultimately we will consider four-point functions where all external bulk operators are parity-even scalars inserted collinearly, we only need the level-two bulk descendants that are $SO(2)_T$ transverse spin and transverse reflection singlets. A general level two descendant operator of a bulk, scalar primary $\mathcal{O}$ satisfying the spin and parity constraints takes the form

$$|\mathcal{O}^{(2)}\rangle = (a\,P_{\tau}^2 + b\,P_i^2)\,|\mathcal{O}\rangle. \tag{2.27}$$

The constraint that $|\mathcal{O}^{(2)}\rangle$ is an $SL(2,\mathbb{R})$ primary is equivalent to demanding

$$0 = \lim_{\vec{x}\to 0}\langle\mathcal{O}^{(2)}(\tau,\vec{x})\mathcal{O}(0,0)\rangle = \lim_{\vec{x}\to 0}(a\,\partial_{\tau}^2 + b\,\vec{\nabla}_{D-1}^2)\frac{1}{|x|^{2\Delta}} = \frac{2\Delta(a(2\Delta+1) - b(D-1))}{\tau^{2\Delta+2}}$$

which yields

$$a = \frac{D-1}{2\Delta+1}b. \tag{2.28}$$

We will define $|\tilde{\mathcal{O}}^{(2)}\rangle \equiv b\left(\frac{D-1}{2\Delta+1}P_\tau^2 + P_i^2\right)|\mathcal{O}\rangle$. The normalization of this state, which we will also need, may be easily computed using the conformal symmetry algebra to be

$$b = \sqrt{\frac{2\Delta+1}{8(D-1)\Delta(\Delta+1)(2(\Delta+1)-D)}} \tag{2.29}$$

which ensures that $|\tilde{\mathcal{O}}^{(2)}\rangle$ is a unit-normalized $SL(2,\mathbb{R})$ primary. We lastly determine the OPE coefficient of $\tilde{\mathcal{O}}^{(2)}$ appearing in the OPE of bulk, scalar primaries $\mathcal{O}_1, \mathcal{O}_2$, relative to $\lambda_{\mathcal{O}\mathcal{O}_1\mathcal{O}_2}$. To do this, we compute (assuming $\tau > 1$)

$$
\begin{aligned}
\lim_{\vec{x}\to 0}\langle\tilde{\mathcal{O}}^{(2)}(\tau,\vec{x})\mathcal{O}_1(1,0)\mathcal{O}_2(0,0)\rangle &= \lim_{\vec{x}\to 0}(a\partial_\tau^2 + b\vec{\nabla}_{D-1}^2)\langle\mathcal{O}(\tau,\vec{x})\mathcal{O}_1(1,0)\mathcal{O}_2(0,0)\rangle \\
&= -b(D-1)\frac{(\Delta+\Delta_{12})(\Delta-\Delta_{12})}{2\Delta+1}\frac{\lambda_{\mathcal{O}\mathcal{O}_1\mathcal{O}_2}}{(\tau-1)^{\Delta+2+\Delta_{12}}\tau^{\Delta+2-\Delta_{12}}}.
\end{aligned}
\tag{2.30}
$$

Note that the above functional form serves as an independent check that we correctly identified $\tilde{\mathcal{O}}^{(2)}$ as an $SL(2,\mathbb{R})$ primary. From (2.30) and comparing with (2.13) we extract

$$\alpha^{(2)}_{\mathcal{O}\mathcal{O}_1\mathcal{O}_2} = \frac{\lambda_{\tilde{\mathcal{O}}^{(2)}\mathcal{O}_1\mathcal{O}_2}}{\lambda_{\mathcal{O}\mathcal{O}_1\mathcal{O}_2}} = -b(D-1)\frac{(\Delta+\Delta_{12})(\Delta-\Delta_{12})}{2\Delta+1}. \tag{2.31}$$

### 2.6 Example: pinning field in $D = 2$

In two dimensions, the pinning field defect of the Ising CFT can be studied exactly using bCFT techniques. As a sanity check to confirm our above analysis about selection rules and crossing symmetry, we compute four-point functions of the endpoints of the pinning field defect both with and without bulk operators. Studying the case of $D = 2$, rather than $D = 4$ which also may be exactly solved, is especially useful for the purpose of verifying how $g$ enters in four-point functions. This is because for $D = 4$ the pinning field defect has $g = 1$.

In $D = 2$, the study of line defects is equivalent to the study of conformal interfaces between a given CFT $\mathcal{T}$ and itself, which is yet still equivalent to the study of conformal boundary conditions of $\mathcal{T}\otimes\bar{\mathcal{T}}$ via the folding trick. A special class of conformal interfaces are *factorized* interfaces, which correspond to separately imposing conformal boundary conditions on $\mathcal{T}$ and $\bar{\mathcal{T}}$ in the folded theory. It turns out that the pinning field line defect of the $2d$ Ising CFT is an example of a factorized interface, obtained by imposing the continuum version of a fixed-spin boundary condition on each copy. We will, for the rest of this section, use $\mathcal{T}$ to denote the $2d$ Ising CFT.

Recall that the $2d$ Ising CFT supports three simple conformal boundary conditions, which may be described in the Cardy state formalism via

$$|\pm\rangle = \frac{1}{\sqrt{2}}|I\rangle\rangle + \frac{1}{\sqrt{2}}|\varepsilon\rangle\rangle \pm \frac{1}{2^{1/4}}|\sigma\rangle\rangle$$

$$|f\rangle = |I\rangle\rangle - |\varepsilon\rangle\rangle.$$

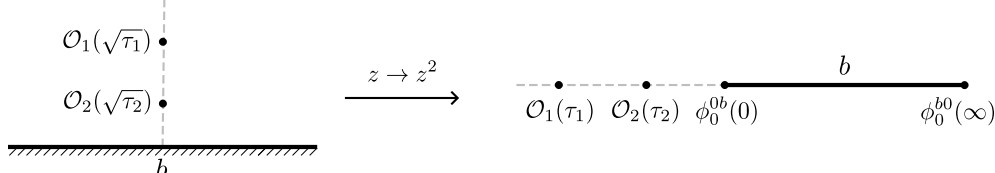

**Figure 10**: The coordinate transformation $z \to z^2$ folds an arbitrary conformal boundary condition $b$ into a semi-infinite line defect, which we, abusing notation, denote with the same label for such factorized line defects. In the presence of additional bulk operator insertions, this prepares a correlation function of the bulk operators in addition to the leading endpoint operator.

The states $|\mathcal{O}\rangle\!\rangle$ are known as Ishibashi states and satisfy the condition $(L_n - \bar{L}_{-n})|\mathcal{O}\rangle\!\rangle = 0$ [70]. For a conformal boundary condition $B$, its boundary $g$-function may be defined as the overlap of the vacuum state with the corresponding Cardy state $|B\rangle$

$$g = \langle 0|B\rangle, \tag{2.32}$$

which has the interpretation of the disc partition function. In the folded theory, the pinning field defect corresponds to the Cardy state

$$|\mathcal{D}^{\pm}\rangle = |\pm\rangle \otimes |\pm\rangle \tag{2.33}$$

from which it is straightforwrard to extract $g = 1/2$ for the pinning field line defect. The determination of other pieces of dCFT data is accomplished straightforwardly, for the most part, using tricks associated with the factorized nature of the pining field defect. A generic line defect of the 2$d$ Ising CFT may be described using the description of $\mathcal{T} \otimes \bar{\mathcal{T}}$ as an orbifold of a compact boson and studying the corresponding set of conformal boundary conditions [26].

We now show how to compute additional pieces of dCFT data for the pinning field defect, making extensive use of the following trick. Consider placing a single copy of $\mathcal{T}$ on the upper half plane (UHP), and impose the "+" boundary condition without loss of generality. Next note that a $z \mapsto z^2$ coordinate transformation maps the negative real line onto the positive real line, which in turn corresponds to placing the pinning field defect along the positive real line with endpoints at the origin and at infinity, see Fig. 10. Since we did not place a non-trivial boundary primary operator at the origin or at infinity, this procedure produces endpoint primary operators $\phi_0^{0+}$ with minimal scaling dimension. With this trick, a general four-point function involving two endpoint primaries, as above, and two bulk operators can be computed as follows. To compare the results with the setup we outlined earlier, we will take the bulk operators to be inserted collinearly with the defect, which amounts to taking the bulk operators to be inserted along the imaginary axis before the $z \mapsto z^2$ transformation. For a general four-point function involving bulk scalar operators $\mathcal{O}_1(\tau_1), \mathcal{O}_2(\tau_2)$ with $\tau_1 < \tau_2 < 0$,

we then have

$$\langle\phi_0^{0+}|\mathcal{O}_1(\tau_1)\mathcal{O}_2(\tau_2)|\phi_0^{0+}\rangle \equiv \lim_{L\to\infty} L^{2\Delta_{0+}}\langle\phi_0^{0+}(L)\phi_0^{+0}(0)\mathcal{O}_2(\tau_2)\mathcal{O}_1(\tau_1)\rangle = \frac{\langle\mathcal{O}_1(\sqrt{\tau_1})\mathcal{O}_2(\sqrt{\tau_2})\rangle_+}{2^{\Delta_1+\Delta_2}|\tau_1|^{\frac{\Delta_1}{2}}|\tau_2|^{\frac{\Delta_2}{2}}},$$

(2.34)

where the correlator on the right is computed in the 2d Ising CFT on the half-plane with the fixed + boundary condition. Note that we only use a single complex coordinate to specify the positions of operators, defining $\mathcal{O}(z) \equiv \mathcal{O}(z, z^*)$, since in the presence of a boundary only a single copy of the Virasoro algebra is preserved.

Let us use this result to compute various quantities for the case of the 2d Ising pinning field defect. We first use (2.34) to extract the scaling dimension of the leading endpoint primary $\phi_0^{0+}$. For the present case of a factorized defect, this turns out to be a universal quantity that only depends on central charge $\Delta = \frac{c}{16}$, giving the result $\Delta_0^{0+} = \frac{1}{32}$ for the Ising pinning field defect. To see this, we consider a slight modification of the relation (2.34) to account for the anomalous transformation of the stress tensor under conformal transformations. Noting that $\langle T(z)\rangle_B = 0$ for any conformal boundary condition $B$, only the Schwarzian derivative term from the stress tensor conformal transformation contributes to the three-point function of two endpoints and $T(z)$[11]

$$\langle\phi_0^{0+}|T(z)|\phi_0^{0+}\rangle = \frac{c}{12}\{\sqrt{z}; z\} = \frac{1}{64}\frac{1}{z^2}.$$

(2.35)

Along with the identical result from the antiholomorphic component of the stress tensor, we may read off $\Delta_0^{0+} = 1/32$.

We next compute OPE coefficients of $\sigma, \varepsilon$ in the OPE of the endpoint operators. This is done using the general form of the one-point function of a bulk primary operator (which must have $s = 0$ to have a non-vanishing one-point function) in the presence of a boundary

$$\langle\mathcal{O}(z)\rangle_B = \frac{a_\mathcal{O}^B}{|z - z^*|^\Delta} \implies \langle\phi_0^{0+}|\mathcal{O}(\tau)|\phi_0^{0+}\rangle = \frac{a_\mathcal{O}^+}{2^{2\Delta}}\frac{1}{|\tau|^\Delta}.$$

(2.36)

Using $\Delta_\sigma = 1/8$, $\Delta_\varepsilon = 1$, $a_\sigma^+ = 2^{1/4}$, and $a_\varepsilon^+ = 1$ we find $\lambda_{\sigma00}^{0+0} = 1$ and $\lambda_{\varepsilon00}^{0+0} = 1/4$ [34].

The dCFT quantities computed above will help to verify the crossing symmetry of four-point functions involving the endpoint operators, as we now demonstrate. To compute four-point functions involving two endpoints and two bulk operators, we need first to know the two-point functions of bulk operators in the presence of the fixed boundary condition. A general such two-point function has the form

$$\langle\mathcal{O}_1(z_1)\mathcal{O}_2(z_2)\rangle_B = \frac{G_{\mathcal{O}_1\mathcal{O}_2}^B(\xi)}{|z_1 - z_1^*|^{\Delta_1}|z_2 - z_2^*|^{\Delta_2}}, \qquad \xi = \frac{|z_1 - z_2|^2}{|z_1 - z_1^*||z_2 - z_2^*|}.$$

(2.37)

---

[11]Here we are using the normalization of $T(z)$ conventional to 2d CFT, see Ref. [71].

For the 2d Ising CFT with fixed-spin boundary, we have [71]

$$G_{\sigma\sigma}^{+}(\xi) = \sqrt{\left(\frac{\xi}{1+\xi}\right)^{1/4} + \left(\frac{\xi}{1+\xi}\right)^{-1/4}}, \tag{2.38}$$

$$G_{\varepsilon\sigma}^{+}(\xi) = \frac{1}{2^{3/4}}\left(\left(\frac{\xi}{1+\xi}\right)^{1/2} + \left(\frac{\xi}{1+\xi}\right)^{-1/2}\right), \tag{2.39}$$

$$G_{\varepsilon\varepsilon}^{+}(\xi) = \frac{1+\xi+\xi^2}{\xi(1+\xi)}. \tag{2.40}$$

Comparing (2.17) with (2.34) and (2.37) we get

$$G_{00\mathcal{O}_2\mathcal{O}_1}^{0+00}(x) = \frac{x^{\Delta_1+\Delta_2}}{4^{\Delta_1+\Delta_2}(1-x)^{\Delta_2}}G_{\mathcal{O}_1\mathcal{O}_2}^{+}(\xi(x)), \qquad \xi(x) = \frac{1}{4}\left((1-x)^{1/4} - (1-x)^{-1/4}\right)^2.$$

Then using (2.38–2.40),

$$G_{00\sigma\sigma}^{0+00}(x) = \frac{1}{(1-x)^{1/8}}, \tag{2.41}$$

$$G_{00\varepsilon\sigma}^{0+00}(x) = \frac{x^{1/8}(2-x)}{4(1-x)}, \tag{2.42}$$

$$G_{00\varepsilon\varepsilon}^{0+00}(x) = 1 + \frac{x^2}{16(1-x)}. \tag{2.43}$$

Expanding the above expressions for $x \to 0$, which corresponds to the OPE limit of the bulk operators approaching each other, we see that in terms of $SL(2,\mathbb{R})$ blocks in the bulk channel we have

$$G_{00\sigma\sigma}^{0++0}(x) = 1 + \frac{1}{128}f_2^{0,0}(x) + \frac{1}{8}\left(f_1^{0,0}(x) + \frac{1}{384}f_3^{0,0}(x)\right) + O(x^4), \tag{2.44}$$

$$G_{00\varepsilon\sigma}^{0++0}(x) = \frac{1}{2}\left(f_{1/8}^{0,7/8}(x) + \frac{1}{20}f_{17/8}^{0,7/8}(x) + O(x^{33/8})\right), \tag{2.45}$$

$$G_{00\varepsilon\varepsilon}^{0++0}(x) = 1 + \frac{1}{16}f_2^{0,0}(x) + O(x^4). \tag{2.46}$$

We see the expected appearance of the OPE coefficients $\lambda_{\sigma00}^{0+0} = 1, \lambda_{\varepsilon00}^{0+0} = 1/4$ that were derived above, noting that $\lambda_{\varepsilon\varepsilon\varepsilon} = 0$ and $\lambda_{\varepsilon\sigma\sigma} = 1/2$. An additional consistency check is that in Eqs. (2.44)-(2.46) $SL(2,\mathbb{R})$ primaries appear only at even bulk Virasoro descendant levels, consistent with the fact that by $180°$ rotation symmetry only even spin operators appear in the OPE of $\phi_0^{0+}$ and $\phi_0^{+0}$ endpoint operators.

Using the crossing relation (2.19) we can also obtain from Eqs. (2.41)-(2.43) expressions for $G_{0\mathcal{O}_2\mathcal{O}_10}^{+000}(x)$. Considering the expansion as $x \to 0$, which corresponds to the OPE limit of the bulk operators approaching the endpoint operators, we see that in terms of $SL(2,\mathbb{R})$

blocks in the endpoint channel we have

$$G_{0\sigma\sigma0}^{+000}(x) = f_{1/32}^{-3/32,3/32}(x) + \frac{1}{136} f_{65/32}^{-3/32,3/32}(x) + O(x^{97/32}), \tag{2.47}$$

$$G_{0\varepsilon\sigma0}^{+000}(x) = \frac{1}{4} \left( f_{1/32}^{-31/32,3/32}(x) - \frac{2}{17} f_{65/32}^{-31/32,3/32}(x) + O(x^{97/32}) \right), \tag{2.48}$$

$$G_{0\varepsilon\varepsilon0}^{+000}(x) = \frac{1}{16} \left( f_{1/32}^{-31/32,31/32}(x) + \frac{32}{17} f_{65/32}^{-31/32,31/32}(x) + O(x^{97/32}) \right). \tag{2.49}$$

We see that the lowest dimension endpoints $\phi_0^{0+}$ appear with the correct OPE coefficients $\lambda_{\sigma00}^{0+0}, \lambda_{\varepsilon00}^{0+0}$.

The final, more non-trivial case is the four-point function involving only endpoint operators where the trick used above is no longer applicable. This was computed in a general setting in [53][12] and gives us

$$G_{0000}^{0+0\pm}(x) = 2 \left( \frac{x^2}{2^8(1-x)} \right)^{\frac{1}{48}} Z_{+\pm}(t(x)), \qquad t(x) = \frac{{}_2F_1(1/2,1/2,1;1-x)}{{}_2F_1(1/2,1/2,1;x)} \tag{2.50}$$

where $Z_{++}(t)$ $(Z_{+-}(t))$ is the partition function of the Ising CFT on a right cylinder with $++$ $(+-)$ boundary conditions and $t = 2\ell/\beta$, where $\ell$ is the length of the open cylinder direction and $\beta$ is the length of the periodic cylinder direction. Explicitly [6],

$$Z_{+\pm}(t) = \frac{1}{2}\chi_0(it) + \frac{1}{2}\chi_{1/2}(it) \pm \frac{1}{\sqrt{2}}\chi_{1/16}(it). \tag{2.51}$$

The functions $\chi_h(\tau)$ are the $c = 1/2$ Virasoro characters listed in appendix A, and we will need the first few $q$-series terms in their $q$-series expansions [71]

$$\chi_0(\tau) = q^{-1/48} + q^{95/48} + q^{143/48} + O(q^{191/48}), \tag{2.52}$$

$$\chi_{1/16}(\tau) = q^{1/24} + q^{25/24} + q^{49/24} + O(q^{73/24}), \tag{2.53}$$

$$\chi_{1/2}(\tau) = q^{23/48} + q^{71/48} + q^{119/48} + O(q^{167/48}). \tag{2.54}$$

where $q = e^{2\pi i\tau}$. The $x \to 0$ limit of $G_{0000}^{0+0\pm}(x)$ is controlled by the OPE where the endpoints shrink down to bulk operators. Indeed, the expansion in this limit may be carried out for a few terms to check for the expected appearance of the bulk primaries. The regime of small $x$ in $\chi_h(it(x))$ is dominated by the terms with the smallest powers of $q$. Expanding in small $x$ and writing the result as a sum of $SL(2,\mathbb{R})$ blocks gives

$$G_{0000}^{0+0\pm}(x) = 1 \pm f_{1/8}^{0,0}(x) + \frac{1}{16} f_1^{0,0}(x) + \frac{1}{512} f_2^{0,0}(x) + O(x^{17/8}). \tag{2.55}$$

In the above, the leading contribution of each Virasoro primary is exactly in agreement with our calculation of the OPE coefficients.

---

[12]We thank Yifan Wang for pointing this out. Here we use a normalization of the defect operators specified in section 2.4.

To be sure that everything works as expected, in particular that the contribution from the defect identity operator comes with the factor $1/g$ in the crossed channel (see Eq. (2.18)), we also compute the conformal block expansion in the crossed channel where the endpoint operator product is expanded in defect/domain wall operators. This can be done by noting that $t(x) = 1/t(1-x)$ and using

$$Z_{++}(t) = \chi_0(i/t), \tag{2.56}$$

$$Z_{+-}(t) = \chi_{1/2}(i/t). \tag{2.57}$$

Again making use of (2.19) we obtain

$$G_{0000}^{+0+0}(x) = 2\left(1 + \frac{1}{512}f_2^{0,0}(x) + O(x^4)\right), \tag{2.58}$$

$$G_{0000}^{-0+0}(x) = \frac{1}{8}\left(f_1^{0,0}(x) + \frac{1}{1536}f_3^{0,0}(x) + O(x^5)\right) \tag{2.59}$$

We thus see that the contributions from the identity operators in the two OPE channels (2.55) and (2.58) differ by a factor of $g$, as they should according to Eq. (2.18). We also observe the lowest dimension non-trival local defect operator with $\Delta_1^{++} = 2$ and the OPE coefficient $\lambda_{010}^{++0} = 1/16$ and the lowest dimension domain wall with $\Delta_0^{+-} = 1$ and $\lambda_{000}^{+-0} = 1/(2\sqrt{2})$.

## 3 Bootstrap bounds on the pinning field defect

We now discuss the conformal bootstrap approach to studying the pinning field defect in the $D = 3$ Ising CFT, with the goal of both obtaining bounds on the leading defect CFT data and presenting the results of spectrum extraction on the higher operator spectrum from $g$-minimization.

Our bootstrap analysis proceeds using the standard formulation of numerical conformal bootstrap problems as polynomial-matrix problems (PMPs), which are internally converted to semidefinite programming problems (SDPs) and solved using the solver SDPB [72]. These techniques will allow us to numerically impose necessary conditions that any hypothetical set of CFT data must satisfy, and to optimize quantities of our choosing subject to the constraints.

The calculations involve numerically searching for a linear functional $\alpha$ of the form

$$\alpha[\mathbf{V}] = \sum_{m=0}^{\Lambda}\sum_i a_m^i \partial_x^m [\mathbf{V}(x)]_i \Big|_{x=\frac{1}{2}} \tag{3.1}$$

that acts on the space of crossing vectors and satisfies certain positive-semidefiniteness constraints. The output of $\alpha$ acting on a crossing vector is a matrix whose entries are, usually[13],

---

[13]We will shortly discuss constraints for which we assign a different meaning to the continuous parameter in SDPB.

polynomials in the scaling dimension of exchanged operators. We refer to the maximum number of derivatives $\Lambda$ as the *derivative order*, which we may increase to yield stronger bounds. The task of searching for $\alpha$ is performed using SDPB. There is a further truncation parameter, which we call $\kappa$, which represents the polynomial order of the numerator in the rational (times positive prefactor) approximation of $SL(2,\mathbb{R})$ blocks, which can be written as

$$\partial_x^m f_\Delta^{\Delta_{12},\Delta_{34}}(x)\Big|_{x=1/2} \approx \frac{(1/2)^\Delta}{\chi_\kappa(\Delta)} P_{m,\kappa}(\Delta_{12},\Delta_{34},\Delta), \tag{3.2}$$

where $\chi_\kappa(\Delta)$ is chosen to factor out all poles (which occur for $\Delta < 0$) and $P_{m,\kappa}(\Delta_{12},\Delta_{34},\Delta)$ are polynomials in $\Delta$. We will choose $\kappa = \Lambda + 10$ throughout this work. Importantly, $\chi_\kappa(\Delta) \geq 0$ for $\Delta \geq 0$, allowing us to express all positivity constraints only in terms of $P_{m,\kappa}$ whenever $\Delta$ is not explicitly fixed, suitable for input into SDPB. Depending on the particular calculation, we will require $\alpha$ to satisfy different constraints, which we will outline in the following subsections in addition to discussing our bounds.

## 3.1 Crossing equations

In our bootstrap calculations, we will consider correlation functions involving the leading endpoint operator $\phi_0^{0+}$ with transverse $SO(2)_T$ spin $s = 0$ together with the leading bulk, scalar primary operators $\sigma$ and $\varepsilon$ of the $3d$ Ising CFT.

Let us assign some labels to the different sets of operators appearing in the dCFT scenario that we consider, refined by the symmetry charges and defect labels. Recall again that all operators under consideration must be $SO(2)_T$ singlets with $s = 0$ and must also be even under reflections about the transverse directions, so we do not label the quantum numbers associated with these symmetries. We denote the set of bulk $SL(2,\mathbb{R})$ primary operators with $\mathbb{Z}_2$ charge $q$ and $\hat{R}^\tau$ eigenvalue $r^\tau$ by $\mathfrak{B}^{q,r^\tau}$. We denote the sets of endpoint, defect, and domain wall operators by $\mathfrak{D}^{0+}$, $\mathfrak{D}^{++}$ and $\mathfrak{D}^{+-}$, respectively. We do not need to additionally specify the $\mathbb{Z}_2$ or $\hat{R}^\tau$ charges of operators in these sectors. The $\mathbb{Z}_2$ symmetry is broken, and all operators in $\mathfrak{D}^{++}$, for which $\hat{R}^\tau$ is a good symmetry, appear with the same (trivial) charge in our setup. From the analysis of section 2.5, we can finally express the content of each of the OPEs as

$$\begin{aligned}
\phi^{+0} \times \phi^{0+} &\subset \mathfrak{D}^{++} & \sigma \times \phi^{0+} &\subset \mathfrak{D}^{0+} \\
\phi^{+0} \times \phi^{0-} &\subset \mathfrak{D}^{+-} & \varepsilon \times \phi^{0+} &\subset \mathfrak{D}^{0+} \\
\phi^{0+} \times \phi^{0+} &\subset \mathfrak{B}^{0,0} \cup \mathfrak{B}^{1,0} & \sigma \times \varepsilon &\subset \mathfrak{B}^{1,0} \cup \mathfrak{B}^{1,1} \\
\varepsilon \times \varepsilon &\subset \mathfrak{B}^{0,0} & \sigma \times \sigma &\subset \mathfrak{B}^{0,0}.
\end{aligned} \tag{3.3}$$

Having accounted for the various symmetry constraints and selection rules, we are ready to express the system of crossing symmetry constraints in a manner suitable for our bootstrap analysis. To do so, we must enumerate all four-point functions involving the external operators $\phi_0^{0+}, \sigma$, and $\varepsilon$, and impose the crossing symmetry relation (2.21). Suppressing the coordinate

dependence, the four-point functions that we consider are

$$\langle\phi^{+0}\phi^{0+}\phi^{+0}\phi^{0+}\rangle, \langle\phi^{+0}\phi^{0-}\phi^{-0}\phi^{0+}\rangle, \langle\sigma\phi^{0+}\phi^{+0}\sigma\rangle, \langle\varepsilon\phi^{0+}\phi^{+0}\sigma\rangle, \langle\varepsilon\phi^{0+}\phi^{+0}\varepsilon\rangle,$$
$$\langle\sigma\sigma\sigma\sigma\rangle, \langle\sigma\sigma\varepsilon\varepsilon\rangle, \langle\sigma\varepsilon\sigma\varepsilon\rangle, \langle\varepsilon\varepsilon\varepsilon\varepsilon\rangle.$$

All others are related to the above set by crossing symmetry. We could also imagine augmenting the above set of correlators by other important operators in the game such as the displacement operator $D_i$, or the other leading primary operators such as $\phi_1^{++}$ or $\phi_0^{+-}$. These larger systems of correlators certainly deserve more study in the future. We list the explicit decomposition of the above four-point functions into conformal blocks, and provide the expressions for the crossing vectors, in Appendix B, and present here the result in a compactified form

$$0 = g^{-1}\mathbf{V}_0^{++}(x) + \mathbf{V}_0^{0,0}(x) + \text{Tr}\Big(P_{\phi_0^{0+}\sigma\varepsilon}\mathbf{V}_{\phi_0^{0+}\sigma\varepsilon}(x)\Big)$$
$$+ \text{Tr}\big(P_{\varepsilon'}\mathbf{V}_{\varepsilon',\Delta_{\varepsilon'}}(x)\big) + \text{Tr}(P_T\mathbf{V}_T(x)) + \text{Tr}\Big(P_{\Delta_{T'}}^{0,0}\mathbf{V}_{\Delta_{T'}}^{0,0}(x)\Big)$$
$$+ \sum_{I,\varepsilon,\tilde{\varepsilon}^{(2)},\varepsilon',\tilde{\varepsilon}'^{(2)}T,T'\neq\mathcal{O}\in\mathfrak{B}^{0,0}}\text{Tr}\Big(P_{\mathcal{O}}^{0,0}\mathbf{V}_\Delta^{0,0}(x)\Big) + \sum_{\sigma,\tilde{\sigma}^{(2)}\neq\mathcal{O}\in\mathfrak{B}^{1,0}}\text{Tr}\Big(P_{\mathcal{O}}^{1,0}\mathbf{V}_\Delta^{1,0}(x)\Big) + \sum_{\mathcal{O}\in\mathfrak{B}^{1,1}}|\lambda_{\mathcal{O}\sigma\varepsilon}|^2\mathbf{V}_\Delta^{1,1}(x)$$
$$+ \sum_{m>0}(\lambda_{m00}^{+0+})^2\mathbf{V}_\Delta^{++}(x) + \sum_m(\lambda_{m00}^{+0-})^2\mathbf{V}_\Delta^{+-}(x) + \sum_{m>0}\text{Tr}\big(P_m^{0+}\mathbf{V}_\Delta^{0+}(x)\big).$$

$$(3.4)$$

The OPE vectors appearing in (3.4) are

$$\vec{\lambda}_T = \begin{pmatrix}\lambda_{T00}^{0+0} & \lambda_{T\sigma\sigma}\end{pmatrix}^{\mathsf{T}} \qquad \vec{\lambda}_{\phi_0^{0+}\sigma\varepsilon} = \begin{pmatrix}\lambda_{\sigma00}^{0+0} & \lambda_{\varepsilon00}^{0+0} & \lambda_{\varepsilon\sigma\sigma} & \lambda_{\tilde{\sigma}^{(2)}00}^{0+0} & \lambda_{\tilde{\varepsilon}^{(2)}00}^{0+0}\end{pmatrix}^{\mathsf{T}}$$

$$\vec{\lambda}_{\varepsilon'} = \begin{pmatrix}\lambda_{\varepsilon'00}^{0+0} & \lambda_{\varepsilon'\sigma\sigma} & \lambda_{\varepsilon'\varepsilon\varepsilon} & \lambda_{\tilde{\varepsilon}'^{(2)}00}^{0+0}\end{pmatrix}^{\mathsf{T}}$$

$$\vec{\lambda}_m^{0+} = \begin{pmatrix}\lambda_{m0\sigma}^{+00} & \lambda_{m0\varepsilon}^{+00}\end{pmatrix}^{\mathsf{T}} \qquad \vec{\lambda}_{\mathcal{O}}^{0,0} = \begin{pmatrix}\lambda_{\mathcal{O}00}^{0+0} & \lambda_{\mathcal{O}\sigma\sigma} & \lambda_{\mathcal{O}\varepsilon\varepsilon}\end{pmatrix}^{\mathsf{T}} \qquad \vec{\lambda}_{\mathcal{O}}^{1,0} = \begin{pmatrix}\lambda_{\mathcal{O}00}^{0+0} & \lambda_{\mathcal{O}\sigma\varepsilon}\end{pmatrix}^{\mathsf{T}}.$$

Let us comment on some assumptions that go into the derivation of the crossing equation in terms of crossing vectors. First, the crossing vector $\mathbf{V}_{\phi_0^{0+}\sigma\varepsilon}(x)$ contains the contribution to the crossing equation from $\phi_0^{0+}$ and the bulk operators $\sigma,\varepsilon$, which appear both as external and exchanged primaries, as well as the level-2 descendant operators $\tilde{\sigma}^{(2)},\tilde{\varepsilon}^{(2)}$. In deriving $\mathbf{V}_{\phi_0^{0+}\sigma\varepsilon}(x)$, we make use of (2.31) when accounting for the contributions from $\tilde{\sigma}^{(2)},\tilde{\varepsilon}^{(2)}$. Next, we have assumed that the ratio of OPE coefficients $r_{\sigma\varepsilon} = \lambda_{\varepsilon\varepsilon\varepsilon}/\lambda_{\varepsilon\sigma\sigma}$ is known exactly, which can be extracted from table 4. This allows us to remove the dependence on $\lambda_{\varepsilon\varepsilon\varepsilon}$. We further simplified the contribution from the stress tensor $T$ by explicitly incorporating the OPE ratio $\lambda_{T\varepsilon\varepsilon}/\lambda_{T\sigma\sigma} = \Delta_\varepsilon/\Delta_\sigma$, which again we take to be known exactly.

A final trick we employed in deriving $\mathbf{V}_{\phi_0^{0+}\sigma\varepsilon}(x)$ is to add to our set of crossing equations

$$\lambda_{\varepsilon\sigma\sigma}^2 - (1.05185442)^2 = 0 \qquad (3.5)$$

which encodes the assumption that $\lambda_{\varepsilon\sigma\sigma}$ takes a fixed value given by the hybrid bootstrap prediction [73]. In our setup, $\lambda_{\varepsilon\sigma\sigma}$ enters as an OPE coefficient whose value can be arbitrary

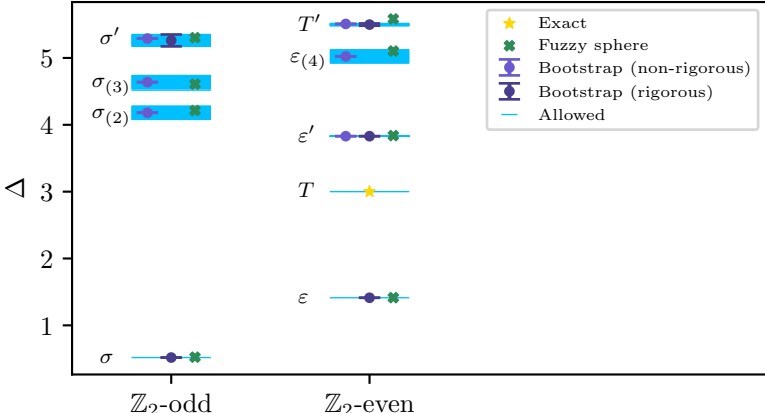

**Figure 11**: A graphical illustration of the dimensions of known bulk primary operators of the $3d$ Ising CFT as computed using the conformal bootstrap and fuzzy sphere. We use $(\sigma_{(\ell)}, \varepsilon_{(\ell)})$ to denote the leading $\mathbb{Z}_2$-(odd,even) spin-$\ell$ traceless symmetric tensor operator of the $3d$ Ising CFT in the instances where the operator does not have a more conventional symbol. Each horizontal light blue line represents an operator dimension explicitly allowed to appear in our crossing equations. The fuzzy sphere data are taken from [51] and the bootstrap data with (non)-rigorous error are taken from ([49])[50]. For clarity, we only explicitly mark "allowed" operators that are bulk primaries in this graphic; their descendants also appear in our crossing equations, as explained in the main text.

when we impose the positivity constraint, and it is not possible to completely eliminate it in favor of its known value without introducing additional unknown parameters, but this approach circumvents this issue.

## 3.2 Incorporating $3d$ Ising CFT data

One of the main goals of our work is to develop the ability to study conformal defects in a way that can effectively and rigorously incorporate details about the bulk operator spectrum. To accomplish this, we take advantage of the wealth of information known about the local operator spectrum of the $D = 3$ Ising CFT to study its pinning-field defect, including scaling dimensions and OPE coefficients of low-lying bulk operators.

Firstly, the scaling dimensions and OPE coefficients involving the leading $\mathbb{Z}_2$-odd and even scalar primaries $\sigma, \varepsilon$ are known to high precision from previous conformal bootstrap calculations, taking values

$$(\Delta_\sigma, \Delta_\varepsilon, \lambda_{\varepsilon\sigma\sigma}, \lambda_{\varepsilon\varepsilon\varepsilon}) = (0.5181489(\mathbf{10}), 1.412625(\mathbf{10}), 1.0518537(\mathbf{41}), 1.532435(\mathbf{19})) \quad (3.6)$$

where the bold uncertainties represent rigorous errors [48]. In our setup, $\sigma$ and $\varepsilon$ appear both as external and exchanged primary operators in the four-point functions we consider.

| $\mathcal{O}$ | $\Delta_{\mathcal{O}}^{\mathrm{boot}}$ | $\delta_{\mathcal{O}}$ | $\epsilon_{\mathcal{O}}$ |
|---|---|---|---|
| $\varepsilon'$ | 3.82951 | **0.00061** | 0.0000813 |
| $\sigma_{(2)}$ | 4.180305 | 0.1 | 0.002 |
| $\sigma_{(3)}$ | 4.63804 | 0.1 | 0.002 |
| $\varepsilon_{(4)}$ | 5.022665 | 0.1 | 0.002 |
| $\sigma'$ | 5.262 | **0.089** | 0.0022 |
| $T'$ | 5.499 | **0.017** | 0.0023 |

**Table 3**: Numerical values of the higher operator dimensions that appear explicitly in our crossing equations, along with the error ranges as encoded by $\delta_{\mathcal{O}}, \epsilon_{\mathcal{O}}$. Bold values of $\delta_{\mathcal{O}}$ are rigorous errors taken from [50], and remaining operator dimensions are taken from [49].

| $\lambda_{\sigma\varepsilon\varepsilon}$ | $\lambda_{\varepsilon\varepsilon\varepsilon}$ | $\lambda_{T\sigma\sigma}/\lambda_{T\varepsilon\varepsilon}$ | $\lambda_{\varepsilon'\sigma\sigma}/\lambda_{\varepsilon'\varepsilon\varepsilon}$ | $\lambda_{T'\sigma\sigma}/\lambda_{T'\varepsilon\varepsilon}$ |
|---|---|---|---|---|
| 1.05185442 | 1.53243407 | 0.36679891 | 0.03453(**14**) | 0.0155607(**51**) |

**Table 4**: Values of bulk OPE coefficients explicitly used in our bootstrap calculations. As explained in the main text, we fix the values of $(\Delta_\sigma, \Delta_\varepsilon, \lambda_{\sigma\varepsilon\varepsilon}, \lambda_{\varepsilon\varepsilon\varepsilon})$ to the navigator minimum reported in [73], which also fixes $\lambda_{T\sigma\sigma}/\lambda_{T\varepsilon\varepsilon} = \Delta_\sigma/\Delta_\varepsilon$. For the remaining OPE coefficients we incorporate the rigorous errors derived in [50].

The fact that these operators are external in our setup makes it somewhat difficult to fully account for the uncertainty in $(\Delta_\sigma, \Delta_\varepsilon)$, since this would involve performing a scan over these quantities during the optimization of any defect quantity. Recently, the navigator bootstrap method has been developed to efficiently deal with precisely this type of problem [74, 75]. However, the precision we are able to achieve for any defect quantity is many orders of magnitude less than the precision of quantities in (3.6), making the error introduced by fixing the values of these quantities insignificant for the purposes of this work. We postpone a more sophisticated navigator bootstrap treatment to future work, opting instead to choose values from the navigator bootstrap minimum of [73]

$$(\Delta_\sigma, \Delta_\varepsilon, \lambda_{\varepsilon\sigma\sigma}, \lambda_{\varepsilon\varepsilon\varepsilon}) = (0.518148884, 1.41262383, 1.05185442, 1.53243407) \qquad (3.7)$$

for all bootstrap calculations performed in our work[14].

Beyond the leading bulk operator data, scaling dimensions of subleading operators of the Ising CFT have been estimated using a variety of methods [49–51]. Our strategy will

---

[14]During the completion of this work after our numerics were finished, very high-precision numerical bootstrap bounds were obtained [76] that rule out the navigator bootstrap minimum of [73] whose values we input directly in this work. The discrepancies in these reported values occur at the eighth, sixth, sixth, and fifth decimal places for $(\Delta_\sigma, \Delta_\varepsilon, \lambda_{\varepsilon\sigma\sigma}, \lambda_{\varepsilon\varepsilon\varepsilon})$, respectively.

be to explicitly assume that a number of the lowest-lying bulk operators appear as internal operators in our crossing equation (3.4), allowing also for uncertainty in the exact values of the scaling dimensions of these operators. We note that an analogous strategy was used to study the space of boundary conditions of specific rational CFTs, where it is especially effective since all bulk operator dimensions are known exactly [53]. In our context, while it will be essential to make discrete bulk spectrum assumptions to yield strong bounds on the defect CFT data, we must exercise caution in how much we assume.

First, excluding the cases where rigorous errors have been obtained [50], the errors in the scaling dimensions of the remaining sub-leading bulk operators from bootstrap techniques are non-rigorous [49]. Another issue is the possibility that the set of operators present in the OPEs $\sigma \times \sigma$, $\sigma \times \varepsilon$, and $\varepsilon \times \varepsilon$, which are the only OPEs studied in [49], does not exhaust the complete set of low-lying bulk operators. The main reason this is a possibility is because bulk $\mathbb{Z}_2$-even tensor primaries with odd $\ell$ are kinematically excluded from these OPEs, and in principle our setup would be sensitive to such operators[15]. A similar statement applies to potential pseudo-tensor operators with relatively low scaling dimension.

In light of these uncertainties about the bulk operator spectrum, we will work with the following set of assumptions about the scaling dimensions of low-lying bulk primaries of the $D = 3$ Ising CFT. The recent results from the fuzzy sphere regularization technique seem to account for all operators predicted in [49] with $\Delta \leq 7$ and $\ell \leq 4$ [51], and further do not uncover any additional operators in this range, giving justification to our assumption that all operators in this range are known. An example of such a potential unknown operator is the leading $\mathbb{Z}_2$-even, vector primary, whose scaling dimension is now expected to be at least $\Delta \geq 7$ [51, 77]. Since there is no existing data that could completely cover the spectrum of operators with $\ell \geq 5$ within some higher range of $\Delta$, we will assume a completely general spectrum of operators with $\Delta \geq 6$, coinciding with the $\ell = 5$ unitarity bound $\Delta \geq \ell + 1$ in $D = 3$.

Our task now is to incorporate the known bulk operators with $\Delta < 6$ and $\ell \leq 4$ into our setup. Our spectrum assumptions in this regime are illustrated graphically in Fig. 11. For each subleading bulk primary $\mathcal{O}$ with scaling dimension $\Delta_{\mathcal{O}}$, we let $\delta$ represent the error in $\Delta_{\mathcal{O}}$ and further use a small discrete sampling width of around $\epsilon = 0.002$, but in some cases where the error is small we use a much smaller sampling[16]. We set $\delta$ according to the rigorous error of $\Delta_{\mathcal{O}}$ whenever possible, and otherwise we uniformly choose $\delta = 0.1$, which is roughly one to four orders of magnitude larger than any non-rigorous error. As seen in Fig. 11, this choice guarantees our error windows include estimates of scaling dimensions from a variety of methods, including conformal bootstrap and fuzzy sphere.

We also incorporate known rigorous bounds on ratios of OPE coefficients of various low-lying operators [50]. This is accomplished as follows. We define the following reduction

---

[15]More precisely, our setup is sensitive to descendants of such operators—see the discussion in section 2.5.

[16]Changes to $\varepsilon, \delta$ in this range have very little effect on our final results.

matrices

$$M_{\varepsilon'}(r_{\varepsilon'}) = \begin{pmatrix} 1 & 0 & 0 & 0 \\ 0 & 1 & r_{\varepsilon'} & 0 \\ 0 & 0 & 0 & 1 \end{pmatrix} \qquad M_{T'}(r_{T'}) = \begin{pmatrix} 1 & 0 & 0 \\ 0 & 1 & r_{T'} \end{pmatrix} \tag{3.8}$$

where $r_{\varepsilon'} = \lambda_{\varepsilon'\varepsilon\varepsilon}/\lambda_{\varepsilon'\sigma\sigma}$ and $r_{T'} = \lambda_{T'\varepsilon\varepsilon}/\lambda_{T'\sigma\sigma}$. The ratios $r_{\varepsilon'}, r_{T'}$ are constrained to take values within the ranges listed in table 4. We finally write

$$\mathrm{Tr}\big(P_{\varepsilon'}\mathbf{V}_{\varepsilon',\Delta_{\varepsilon'}}(x)\big) = \mathrm{Tr}\Big(P'_{\varepsilon'}M_{\varepsilon'}(r_{\varepsilon'})\mathbf{V}_{\varepsilon',\Delta_{\varepsilon'}}(x)M_{\varepsilon'}(r_{\varepsilon'})^{\mathsf{T}}\Big) = \mathrm{Tr}\Big(P'_{\varepsilon'}\mathbf{V}'_{\varepsilon',\Delta_{\varepsilon'}}(x;r_{\varepsilon'})\Big) \tag{3.9}$$

$$\mathrm{Tr}\Big(P^{0,0}_{T'}\mathbf{V}^{0,0}_{\Delta_{T'}}(x)\Big) = \mathrm{Tr}\Big(P'_{T'}M_{T'}(r_{T'})\mathbf{V}^{0,0}_{\Delta_{T'}}(x)M_{T'}(r_{T'})^{\mathsf{T}}\Big) = \mathrm{Tr}\Big(P'_{T'}\mathbf{V}'_{T',\Delta_{T'}}(x;r_{T'})\Big) \tag{3.10}$$

$$\vec{\lambda}'_{\varepsilon'} = \Big(\lambda^{0+0}_{\varepsilon'00} \ \lambda_{\varepsilon'\sigma\sigma} \ \lambda^{0+0}_{\tilde{\varepsilon}'(2)00}\Big)^{\mathsf{T}} \qquad \vec{\lambda}'_{T'} = \Big(\lambda^{0+0}_{T'00} \ \lambda_{T'\sigma\sigma}\Big)^{\mathsf{T}}, \tag{3.11}$$

so the ratios of OPE coefficients $r_{\varepsilon'}, r_{T'}$ now enter as explicit free parameters. We will allow these ratios to take all values within the ranges specified in table 4. This is accomplished using a method we discuss in more detail in section 3.4.

In addition to each bulk primary operator that we include explicitly, we also must consider the contributions of its descendant operators whose scaling dimensions obey $\Delta \leq 6$, some of which will be $SL(2, \mathbb{R})$ primary operators as we discussed before. Altogether, this leads to a number of constraints that are fixed across all of our calculations

$$\alpha[\mathbf{V}_T(x)] \succeq 0 \tag{3.12}$$

$$\alpha[\mathbf{V}'_{\varepsilon',\Delta}(x;r)] \succeq 0 \qquad \Delta \in \mathsf{D}_{\varepsilon'}, r \in [r^{\min}_{\varepsilon'}, r^{\max}_{\varepsilon'}] \tag{3.13}$$

$$\alpha[\mathbf{V}'_{T',\Delta}(x;r)] \succeq 0 \qquad \Delta \in \mathsf{D}_{T'}, r \in [r^{\min}_{T'}, r^{\max}_{T'}] \tag{3.14}$$

$$\alpha[\mathbf{V}^{0,0}_{\Delta}(x)] \succeq 0 \qquad \Delta \in \{\Delta_\varepsilon + 4, 5\} \cup \mathsf{D}_{\varepsilon_{(4)}} \cup [6, \infty) \tag{3.15}$$

$$\alpha[\mathbf{V}^{1,0}_{\Delta}(x)] \succeq 0 \qquad \Delta \in \{\Delta_\sigma + 4\} \cup \mathsf{D}_{\sigma_{(2)}} \cup \mathsf{D}_{\partial\sigma_{(3)}} \cup \mathsf{D}_{\sigma'} \cup [6, \infty) \tag{3.16}$$

$$\alpha[\mathbf{V}^{1,1}_{\Delta}(x)] \succeq 0 \qquad \Delta \in \mathsf{D}_{\sigma_{(3)}} \cup \mathsf{D}_{\partial\sigma_{(2)}} \cup [6, \infty) \tag{3.17}$$

where $\mathsf{D}_{\mathcal{O}} = \{\Delta^{\mathrm{boot}}_{\mathcal{O}} + n\epsilon_{\mathcal{O}} : n \in \mathbb{Z}, |n\epsilon_{\mathcal{O}}| \leq \delta_{\mathcal{O}}\}$. Note that we set $\delta_{\partial^n\mathcal{O}} = \delta_{\mathcal{O}}, \epsilon_{\partial^n\mathcal{O}} = \epsilon_{\mathcal{O}}$. For the numerical values of $\Delta^{\mathrm{boot}}_{\mathcal{O}}, \delta_{\mathcal{O}}, \epsilon_{\mathcal{O}}$ see table 3. Further, note that when $\mathcal{O} = \varepsilon', T'$ in the above we incorporate the rigorous errors on the ratios of OPE coefficients $\lambda_{\varepsilon'\varepsilon\varepsilon}/\lambda_{\varepsilon'\sigma\sigma}$ and $\lambda_{T'\varepsilon\varepsilon}/\lambda_{T'\sigma\sigma}$, which we quote in table 4 [50].

Finally, we mention in passing that we also could have used a continuous interval positivity constraint to account for uncertainties instead of a discrete sampling [78], but for some bulk operators we will use such a continuous interval constraint for ratios of OPE coefficients, as we will explain shortly, and it is not as straightforward to combine these distinct continuous constraints in SDPB. We find very little sensitivity in our setup to changes in $\epsilon$ or $\delta$, so the discrete sampling is sufficient for our purposes.

## 3.3 Defect assumptions

Having outlined the spectrum assumptions we make about the set of bulk operators, we now turn to describing our limited set of assumptions about defect-changing operators. Recall

that our main goal is to describe the IR fixed-point of a defect RG flow triggered by a relevant, explicit $\mathbb{Z}_2$ symmetry-breaking perturbation. This physical scenario implies two main consequences for the low-lying defect CFT data:

1. The IR defect fixed-point lacks a non-trivial, relevant defect operator preserving both parity/reflection and $SO(2)_T$ spacetime symmetries. Our setup is not sensitive to parity/reflection and $SO(2)_T$ violating operators, so we assume $\Delta_1^{++} \geq 1$.

2. The defect $g$-theorem guarantees that $g < 1$ at the IR defect fixed-point.

The first consequence will be explicitly assumed, and the second will guide our search for finding the physical values of the Ising pinning field defect's CFT data based on those that *imply* $g < 1$ as a non-trivial consequence of crossing symmetry.

Based on these simple physical consequences, we impose the following set of constraints on $\alpha$ in the most general setting. All of our calculations will begin with the assumption that there exists an endpoint operator $\phi_0^{0+}$ with lowest scaling dimension $\Delta_0^{0+}$. Then the constraints on $\alpha$ become

$$\alpha[\mathbf{V}_\Delta^{0+}] \succeq 0 \qquad \Delta \geq \Delta_{1,\text{min}}^{0+} \tag{3.18}$$

$$\alpha[\mathbf{V}_\Delta^{++}] \succeq 0 \qquad \Delta \geq \Delta_{1,\text{min}}^{++} \tag{3.19}$$

$$\alpha[\mathbf{V}_\Delta^{+-}] \succeq 0 \qquad \Delta \geq \Delta_{0,\text{min}}^{+-}. \tag{3.20}$$

The gap assumptions that lose no generality thus are $\Delta_{1,\text{min}}^{0+} = \Delta_0^{0+}$, $\Delta_{1,\text{min}}^{++} = 1$, and $\Delta_{0,\text{min}}^{+-} = 0$. We will also later make various non-generic gap assumptions in the defect-changing operator sectors to obtain stronger bounds.

## 3.4  $\Delta_0^{0+}$ island and bounds on $|\lambda_{\varepsilon00}^{0+0}/\lambda_{\sigma00}^{0+0}|$

The first set of bounds we present constrain the ratio of the OPE coefficients $\lambda_{\sigma00}^{0+0}$ and $\lambda_{\varepsilon00}^{0+0}$, shown in Fig. 12. The reason to introduce these bounds first is because we will use the results in all of our subsequent calculations as a way to encode information about the lack of degeneracy of operators with scaling dimension equal to $\Delta_\sigma$ or $\Delta_\varepsilon$ appearing in the $\phi_0^{0+} \times \phi_0^{+0}$ OPE. The main effect of this additional assumption will be to give a stronger restriction on the allowed values of $\Delta_0^{0+}$.

There are a few main takeaways from the bounds of Fig. 12. First, they reveal an island of allowed values of $\Delta_0^{0+}$ that is separated from the remaining allowed values, even when no non-trivial gap assumptions for defect-changing operators are made[17]. We claim that this island indeed contains the physical pinning-field defect solution (up to our fixed assumptions about the bulk) based on the additional evidence that we will summarize from our other bounds, and also the fact that the fuzzy sphere estimate of $\Delta_0^{0+}$ and $\lambda_{\varepsilon00}^{0+0}/\lambda_{\sigma00}^{0+0}$ agree very well with the values contained within the island. It is further noteworthy that the allowed

---

[17]The existence of the island in the allowed region is not contingent on the extra assumption encoded by fixing $|\lambda_{\varepsilon00}^{0+0}/\lambda_{\sigma00}^{0+0}|$, but its width in $\Delta_0^{0+}$ is reduced.

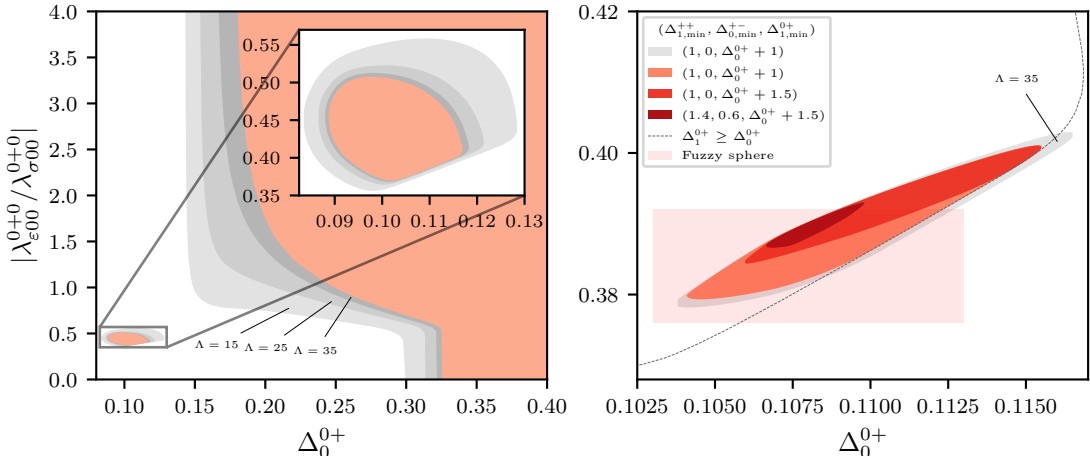

**Figure 12**: Bounds on the ratio of OPE coefficients of the $3d$ Ising pinning field endpoint primaries $\phi_0^{0+}$ fusing to $\varepsilon$ and $\sigma$. We compute this for various gap assumptions and choices of $\Lambda$. Unless otherwise indicated the bounds are computed with $\Lambda = 45$. **Left:** The most general bound where only $\Delta_1^{++} \geq 1$ is assumed. **Right:** The same calculation with stronger gap assumptions. The dashed line represents the boundary of the $\Lambda = 45$ result from the left figure. We show the $\Lambda = 35$ bound with the gap assumptions $(1, 0, \Delta_0^{0+} + 1)$ to demonstrate that the region that is robust to increasing gaps/$\Lambda$ does not lie on the boundary (dashed line) of the bound with the weakest assumptions. The fuzzy sphere data $(\lambda_{\varepsilon 00}^{0+0}, \lambda_{\sigma 00}^{0+0}) = (0.3334(9), 0.869(19))$ are taken from [27], and we use standard error propagation to determine the final fuzzy sphere estimate $|\lambda_{\varepsilon 00}^{0+0}/\lambda_{\sigma 00}^{0+0}| = 0.384(8)$ shown in the plot.

"continent" that exists for values of $\Delta_0^{0+}$ exceeding those contained in the island does not seem to possess any feature that is robust to an increase in $\Lambda$. This makes it difficult to say whether any potentially new physical solutions corresponding to as-of-yet unknown line defects lie in this region.

We now describe how the bounds on $\lambda_{\varepsilon 00}^{0+0}/\lambda_{\sigma 00}^{0+0}$ are derived. Let us consider the contribution of operators $\phi_0^{0+}, \sigma, \varepsilon$ to the crossing equation (3.4) through the OPE coefficients

$$\vec{\lambda}_{\phi_0^{0+}\sigma\varepsilon} = \left( \lambda_{\sigma 00}^{0+0} \; \lambda_{\varepsilon 00}^{0+0} \; \lambda_{\varepsilon\sigma\sigma} \; \lambda_{\tilde{\sigma}^{(2)}00}^{0+0} \; \lambda_{\tilde{\varepsilon}^{(2)}00}^{0+0} \right)^{\mathsf{T}}, \tag{3.21}$$

given by $\mathrm{Tr}\left( P_{\phi_0^{0+}\sigma\varepsilon} \mathbf{V}_{\phi_0^{0+}\sigma\varepsilon}(x) \right)$. Recall that $P_{\phi_0^{0+}\sigma\varepsilon} = \vec{\lambda}_{\phi_0^{0+}\sigma\varepsilon} \cdot \vec{\lambda}_{\phi_0^{0+}\sigma\varepsilon}^{\mathsf{T}}$. As was pointed out in [48], if in some calculation we only impose $\alpha[\mathbf{V}_{\phi_0^{0+}\sigma\varepsilon}(x)] \succeq 0$, this allows for solutions to crossing containing contributions of the form

$$\sum_i \mathrm{Tr}\left( P_{\phi_0^{0+}\sigma\varepsilon}^i \mathbf{V}_{\phi_0^{0+}\sigma\varepsilon} \right), \tag{3.22}$$

which has the interpretation that multiple operators appear with dimensions equal to $\Delta_\sigma, \Delta_\varepsilon$, each with their own distinct OPE vector $\vec{\lambda}_{\phi_0^{0+}\sigma\varepsilon}^i$ of the form (3.21) but with varying values of

the OPE coefficients. Clearly then the uniqueness of the operators with scaling dimensions equal to $\Delta_\sigma, \Delta_\varepsilon$ is not encoded in this treatment. The uniqueness of operators with dimensions equal to $\Delta_\sigma, \Delta_\varepsilon$ in the $\phi_0^{0+} \times \phi_0^{+0}$ OPE is an important physical assumption, so imposing it can potentially eliminate unphysical solutions and give stronger bounds.

We will extract consequences of the uniqueness of $\sigma, \varepsilon$ appearing in the $\phi^{0+} \times \phi^{+0}$ OPE by explicitly fixing $|\lambda_{\varepsilon00}^{0+0}/\lambda_{\sigma00}^{0+0}|$ to different values and determining which choices are disallowed by crossing symmetry. Note that we bound only the absolute value because for any solution where $\lambda_{\varepsilon00}^{0+0}/\lambda_{\sigma00}^{0+0}$ is allowed $\lambda_{\varepsilon00}^{0-0}/\lambda_{\sigma00}^{0-0} = -\lambda_{\varepsilon00}^{0+0}/\lambda_{\sigma00}^{0+0}$ must also be allowed by symmetry. Thus, we may assume $\lambda_{\varepsilon00}^{0+0}/\lambda_{\sigma00}^{0+0} > 0$ without loss of generality. The main external scanning parameter in our problem is the dimension of the leading endpoint operator $\Delta_0^{0+}$. For any given $\Delta_0^{0+}$, we compute upper and lower bounds on $|\lambda_{\varepsilon00}^{0+0}/\lambda_{\sigma00}^{0+0}|$ using a "cutting-curve" algorithm adapted from an essentially identical algorithm introduced in [79], which we describe in Appendix C.

To implement this scan, we fix some $\theta_{\sigma\varepsilon}^{\phi_0^{0+}}$ [18] and let

$$\left(\lambda_{\sigma00}^{0+0} \ \lambda_{\varepsilon00}^{0+0}\right) = |\lambda|\left(\sin\theta_{\sigma\varepsilon}^{\phi_0^{0+}} \ \cos\theta_{\sigma\varepsilon}^{\phi_0^{0+}}\right). \tag{3.23}$$

Substituting (3.23) into $\vec{\lambda}_{\phi_0^{0+}\sigma\varepsilon}$, we define the reduced OPE vector

$$\vec{\lambda}_{\phi_0^{0+}\sigma\varepsilon}^{\theta} \equiv \left(|\lambda| \ \lambda_{\varepsilon\sigma\sigma} \ \lambda_{\tilde{\sigma}^{(2)}00}^{0+0} \ \lambda_{\tilde{\varepsilon}^{(2)}00}^{0+0}\right)^{\mathsf{T}} = M^\theta(\theta_{\sigma\varepsilon}^{\phi_0^{0+}}) \cdot \vec{\lambda}_{\phi_0^{0+}\sigma\varepsilon} \qquad M^\theta(\theta_{\sigma\varepsilon}^{\phi_0^{0+}}) = \begin{pmatrix} \sin\theta_{\sigma\varepsilon}^{\phi_0^{0+}} & \cos\theta_{\sigma\varepsilon}^{\phi_0^{0+}} & 0 & 0 & 0 \\ 0 & 0 & 1 & 0 & 0 \\ 0 & 0 & 0 & 1 & 0 \\ 0 & 0 & 0 & 0 & 1 \end{pmatrix}.$$

We finally replace

$$\mathrm{Tr}\left(P_{\phi_0^{0+}\sigma\varepsilon}\mathbf{V}_{\phi_0^{0+}\sigma\varepsilon}(x)\right) \to \mathrm{Tr}\left(P_{\phi_0^{0+}\sigma\varepsilon}^\theta \mathbf{V}_{\phi_0^{0+}\sigma\varepsilon}^\theta(x;\theta_{\sigma\varepsilon}^{\phi_0^{0+}})\right) \tag{3.24}$$

with $\mathbf{V}_{\phi_0^{0+}\sigma\varepsilon}^\theta(x;\theta_{\sigma\varepsilon}^{\phi_0^{0+}}) = M^\theta(\theta_{\sigma\varepsilon}^{\phi_0^{0+}}) \cdot \mathbf{V}_{\phi_0^{0+}\sigma\varepsilon}(x) \cdot M^\theta(\theta_{\sigma\varepsilon}^{\phi_0^{0+}})^{\mathsf{T}}$ in the crossing equation to account for our explicit choice of $\theta_{\sigma\varepsilon}^{\phi_0^{0+}}$. To compute the bounds shown in Fig. 12, we run the "cutting curve" algorithm subject to (3.12–3.20) in addition to

$$\alpha[\mathbf{V}_0^{0,0}(x)] = 1 \tag{3.25}$$
$$\alpha[\mathbf{V}_0^{++}(x)] \succeq 0, \tag{3.26}$$

as our normalization and to account for the identity operator in the defect channel with unspecified $g$, and finally

$$\alpha[\mathbf{V}_{\phi_0^{0+}\sigma\varepsilon}^\theta(x;\theta_{\sigma\varepsilon}^{\phi_0^{0+}})] \succeq 0 \tag{3.27}$$

---

[18]The trigonometric parameterization is convenient for the purpose of bounding $|\lambda_{\varepsilon00}^{0+0}/\lambda_{\sigma00}^{0+0}|$ since it makes the scanning region bounded.

where $\theta_{\sigma\varepsilon}^{\phi_0^{0+}}$ takes the role of the parameter $\gamma$ in the notation of Appendix C.

In calculations of other defect quantities, we will make use of the bounds of Fig. 12 in the following way. Again, without using a more sophisticated setup such as the navigator method, it is somewhat impractical to scan over fixed values of $\theta_{\sigma\varepsilon}^{\phi_0^{0+}}$ as we did above when computing bounds on other quantities. We settle instead for allowing $|\lambda_{\varepsilon00}^{0+0}/\lambda_{\sigma00}^{0+0}|$ to take any values within the allowed regions implied by Fig. 12. In SDPB each constraint supports up to one continuous parameter that appears polynomially. The continuous parameter usually takes the role of the scaling dimension of an exchanged primary operator, but since in $\mathbf{V}_{\phi_0^{0+}\sigma\varepsilon}(x)$ all operator dimensions are fixed, we are free to use $r_{\sigma\varepsilon}^{\phi_0^{0+}} \equiv |\lambda_{\varepsilon00}^{0+0}/\lambda_{\sigma00}^{0+0}|$ as a continuous parameter. It is straightforward to modify the derivation of the constraint $\mathbf{V}_{\phi_0^{0+}\sigma\varepsilon}^{\theta}(x, \theta_{\sigma\varepsilon}^{\phi_0^{0+}})$ to arrive on one that depends directly on $r_{\sigma\varepsilon}^{\phi_0^{0+}}$

$$\mathbf{V}_{\phi_0^{0+}\sigma\varepsilon}^r(x, r_{\sigma\varepsilon}^{\phi_0^{0+}}) = M^r(r_{\sigma\varepsilon}^{\phi_0^{0+}}) \cdot \mathbf{V}_{\phi_0^{0+}\sigma\varepsilon}(x) \cdot M^r(r_{\sigma\varepsilon}^{\phi_0^{0+}})^{\mathsf{T}} \qquad M^r(r_{\sigma\varepsilon}^{\phi_0^{0+}}) = \begin{pmatrix} 1 & r_{\sigma\varepsilon}^{\phi_0^{0+}} & 0 & 0 & 0 \\ 0 & 0 & 1 & 0 & 0 \\ 0 & 0 & 0 & 1 & 0 \\ 0 & 0 & 0 & 0 & 1 \end{pmatrix},$$
(3.28)

which enters in the crossing equation via the term

$$(\vec{\lambda}_{\sigma\varepsilon}^r(r_{\sigma\varepsilon}^{\phi_0^{0+}}))^{\mathsf{T}} \cdot \mathbf{V}_{\phi_0^{0+}\sigma\varepsilon}^r(x, r_{\sigma\varepsilon}^{\phi_0^{0+}}) \cdot \vec{\lambda}_{\sigma\varepsilon}^r(r_{\sigma\varepsilon}^{\phi_0^{0+}}) \qquad \vec{\lambda}_{\sigma\varepsilon}^r(r_{\sigma\varepsilon}^{\phi_0^{0+}}) = \begin{pmatrix} \lambda_{\sigma00}^{0+0} & \lambda_{\varepsilon\sigma\sigma} & \lambda_{\tilde{\sigma}^{(2)}00}^{0+0} & \lambda_{\tilde{\varepsilon}^{(2)}00}^{0+0} \end{pmatrix}.$$

Note that $r_{\sigma\varepsilon}^{\phi_0^{0+}}$ appears quadratically in $\mathbf{V}_{\sigma\varepsilon}^r(x; r_{\sigma\varepsilon}^{\phi_0^{0+}})$. We would like to impose

$$r_{\sigma\varepsilon}^{\phi_0^{0+}}(\Delta_0^{0+}) \in [r_{\min}(\Delta_0^{0+}), r_{\max}(\Delta_0^{0+})],$$

where the upper and lower bounds are set by Fig. 12 and depend on the choices of $\Lambda$ and gap assumptions made in a particular calculation. In SDPB, however, the continuous parameter is assumed to take values in $y \in [0, \infty)$. By letting

$$r_{\sigma\varepsilon}^{\phi_0^{0+}}(\Delta_0^{0+}) = r_{\min}(\Delta_0^{0+}) + \left(r_{\max}(\Delta_0^{0+}) - r_{\min}(\Delta_0^{0+})\right) \frac{y}{1+y} \tag{3.29}$$

we can convert the interval constraint on $r_{\sigma\varepsilon}^{\phi_0^{0+}}$ to a form suitable for input to SDPB, further without introducing any discretization error, which would otherwise be required. Upon making the substitution (3.29) the crossing vector $\mathbf{V}_{\phi_0^{0+}\sigma\varepsilon}^r(x, r_{\sigma\varepsilon}^{\phi_0^{0+}})$ depends non-polynomially on $y$, which is not suitable for input to SDPB. This is easily fixed by choosing $1/(1+y)^2$ as a positive prefactor, and imposing

$$\alpha[(1+y)^2 \mathbf{V}_{\phi_0^{0+}\sigma\varepsilon}^r(x, y)(\Delta_0^{0+})] \succeq 0 \qquad \forall y \geq 0 \tag{3.30}$$

in our subsequent calculations. With this strategy, the uniqueness of $\sigma, \varepsilon$ in the $\phi_0^{0+} \times \phi_0^{+0}$ OPE is only partially accounted for, since again solutions with multiple OPE vectors in the spirit of (3.22) are allowed, but this nonetheless gives improved constraining power in our other bounds.

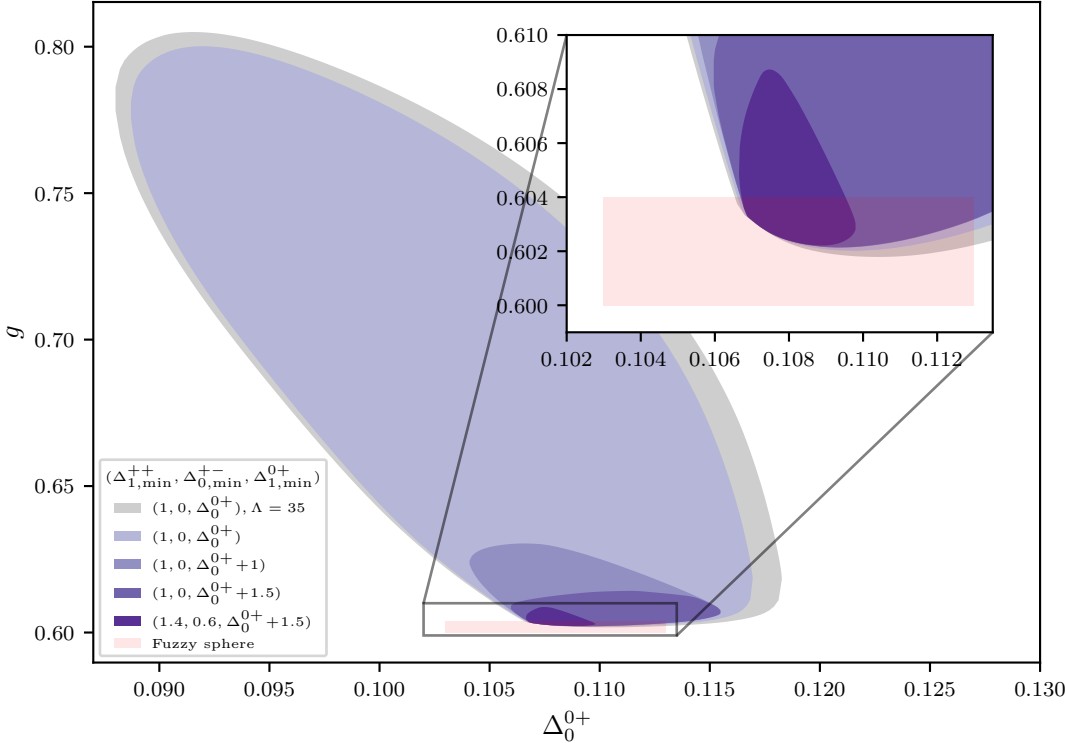

**Figure 13**: Bounds on the defect $g$-function with various defect spectrum assumptions. The bounds are obtained using $\Lambda = 45$, unless otherwise indicated.

## 3.5 Bounds on $g$-function

Next we explore bounds on the defect $g$-function, which are shown in Fig. 13. A number of interesting physical implications follow from these bounds. First, we learn that all values of $\Delta_0^{0+}$ sitting within the islands of Fig. 12, even for the range obtained with no non-trivial assumptions about the defect operator spectrum, are inconsistent with $g \geq 1$. This means that a generic point within the island will satisfy all of the necessary requirements we expect of the true pinning field defect. Together with existing analytical [23] and numerical [27] evidence, which predicts values for $\Delta_0^{0+}$ and $g$ that are overall very consistent with the island seen in Fig. 13, we can safely reason that the physical solution indeed lies inside of the island. Another interesting feature is that $g < 1/2$ is totally ruled out within the island. This is significant because it means that the non-simple LRO defect $\mathcal{D}^+ \oplus \mathcal{D}^-$ with $g_{\text{LRO}} = 2g_\pm > 1$ *can* flow to the trivial defect under the $\phi_0^{+-}$ domain wall perturbation that we will soon show to be relevant.

Another interesting feature of our bound on $g$ is the region where the bounds appear to saturate. We can see that there is a kink in the island along its lower edge, and there is a sizable region near this kink where the lower bound on $g$ is essentially insensitive to increasing $\Lambda$ and imposing stronger gap assumptions. This behavior serves as strong evidence that our

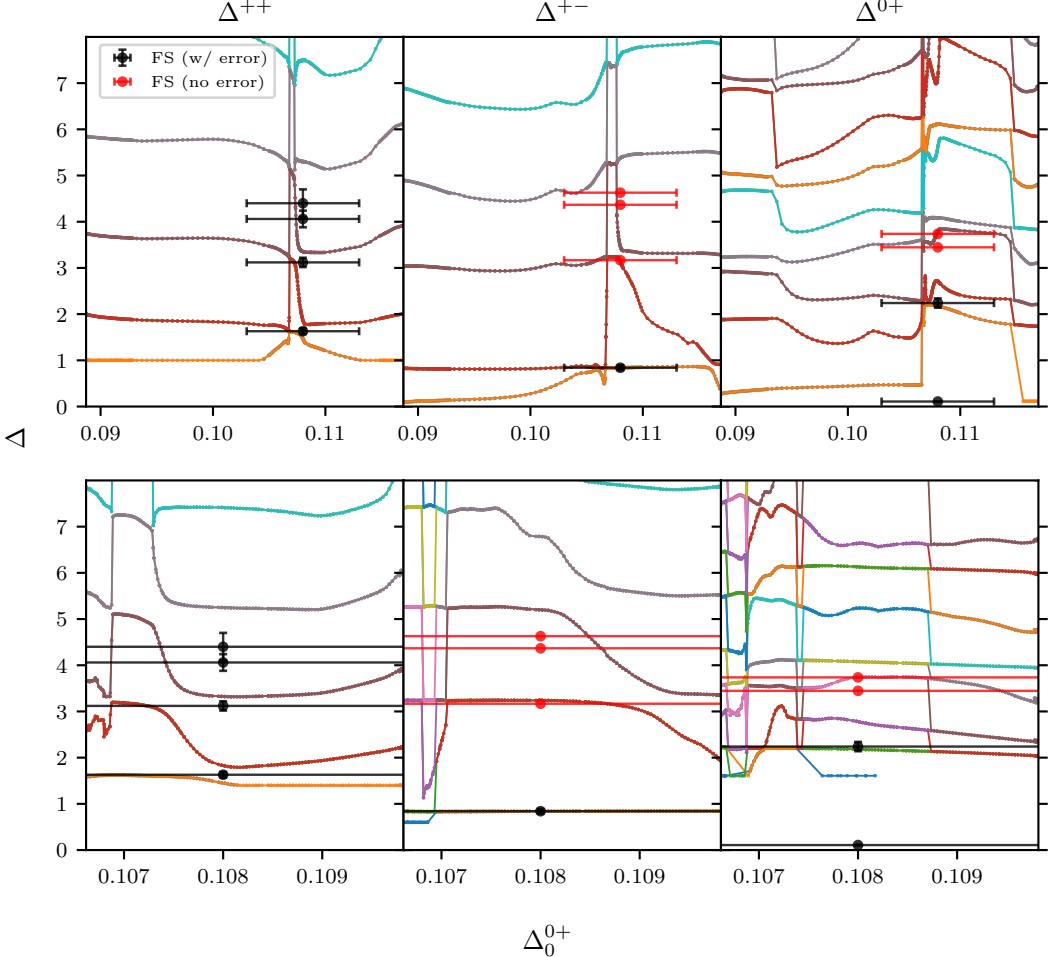

**Figure 14**: Zeros of the optimal functional obtained from $g$-minimization at $\Lambda = 45$ for the defect, domain wall, and endpoint spectra, going from left to right. The spectra in the upper row were obtained upon minimizing $g$ with no assumptions on the defect-changing spectrum other than $\Delta_1^{++} \geq 1$. The spectra in the lower row were obtained assuming $\Delta_1^{++} \geq 1.4$, $\Delta_0^{+-} \geq 0.6$, and $\Delta_1^{0+} \geq \Delta_0^{0+} + 1.5$. We also present the corresponding operator dimensions computed with the fuzzy sphere regularization for comparison. Beyond the low-dimension operators and error estimates from fuzzy sphere listed in table 3, we also plot all higher operators reported in [27, 33]. Again, we default to using reported values and errors obtained from the energy spectrum for these operators.

setup is close to saturating the physical value of $g$, which we expect to live near this kink.

A further indication that the physical dCFT data lie near the kink comes from the spectra produced during $g$ minimization. When $g$ is minimized, the functional $\alpha^*$ which produces

the optimal bound also produces a unitary solution to the crossing equations for free [49, 56]. We show the spectra of defect-changing operators implied from $\alpha^*$ in Fig. 14. When we choose $\Delta_0^{0+}$ to be near the kink, the extracted low-lying spectra show excellent agreement with the fuzzy sphere predictions for $\Delta_1^{++}$, $\Delta_2^{++}$, $\Delta_0^{+-}$, $\Delta_1^{+-}$, $\Delta_0^{0+}$[19], and $\Delta_1^{0+}$. A crude way we estimate the value of $\Delta_0^{0+}$ for which $g$ is globally minimized within the island in the limit $\Lambda \to \infty$[20] is to minimize the difference between our bound at $\Lambda = 35$ with our weakest assumptions (only $\Delta_0^{++} \geq 1$) and our most aggressive assumptions. This choice represents the point at which the bound on $g$ is least sensitive to an increase in constraining power within our setup. At this value of $\Delta_0^{0+}$ nearly all of the other quantities we have computed bounds on have regions that are stable both to increases in $\Lambda$ and to making stronger defect-changing operator spectrum assumptions. This procedure was used to determine the value of $\Delta_0^{0+}$ used to obtain the spectrum extraction results reported in table 2.

Unfortunately, looking beyond the low-lying operators $\phi_1^{++}$, $\phi_2^{++}$, $\phi_0^{+-}$, $\phi_1^{+-}$, $\phi_0^{0+}$, $\phi_1^{0+}$ the fuzzy sphere appears to predict a few operators that we do not find in the extremal spectrum, so our ability to make comparison is rather limited. There are a few reasons that could explain why we seem to miss these operators. One explanation is that we have not yet included enough constraints in our setup. For instance, we have not yet included constraints coming from using the displacement operator or $\phi_0^{+-}, \phi_1^{++}$ as external operators, which would allow more physical assumptions to be made. We also could include more constraints about the bulk, such as fixing more of the bulk descendant OPE coefficients or fully incorporating $3d$ conformal blocks, as we will discuss later. A final possibility is that the bulk CFT data for which we do not allow uncertainty, given in (3.7), are too far from their true values for the $3d$ Ising CFT[14]. It will be interesting in the future to implement these improvements to learn more about the higher operator spectrum.

We finally mention how we obtain the bounds on $g$ following the standard technique to optimize OPE coefficients. To briefly recall how this works, we first act on the crossing equation with a linear functional $\alpha$ and write the result as

$$\frac{1}{g}\alpha[\mathbf{V}_0^{++}] + \alpha[\mathbf{V}_0^{0,0}] + \sum_i \mathrm{Tr}(P_i\alpha[\mathbf{V}_i]) = 0 \tag{3.31}$$

where $\mathbf{V}_i$ runs over all crossing vectors included in (3.12–3.20) in addition to (3.30), representing the remaining terms in the crossing equation. Setting $\alpha[\mathbf{V}_0^{0,0}] = \pm 1$ and imposing $\alpha[\mathbf{V}_i] \succeq 0$ allows us to derive

$$\alpha[\mathbf{V}_0^{++}] \leq \mp g, \tag{3.32}$$

which means $\mp\alpha[\mathbf{V}_0^{++}]$ yields a valid upper/lower bound on $g$, and we use SDPB to find an optimal $\alpha$ subject to the constraints.

---

[19]$\phi_0^{0+}$ is an external operator. Since the OPE coefficients involving $\phi_0^{0+}$ are accounted for in the discrete constraint $\mathbf{V}_{\phi_0^{0+}\sigma\varepsilon}^r$, we do not expect it to necessarily appear as a zero of $\alpha[\mathbf{V}_\Delta^{0+}(x)]$.

[20]$\Lambda$ is the derivative order, defined in (3.1).

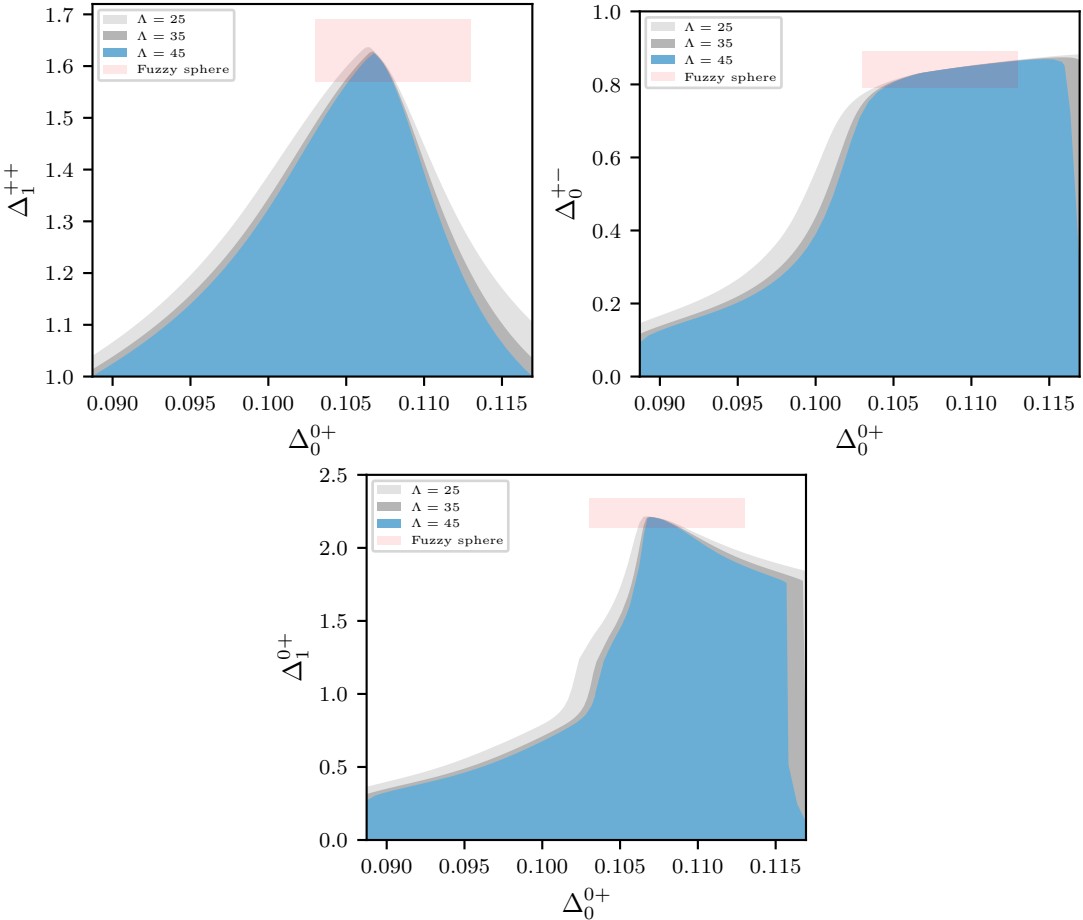

**Figure 15**: Upper bounds on the scaling dimension of the lightest non-trivial defect primary $\Delta_1^{++}$, the lightest domain wall primary $\Delta_0^{+-}$, and the second-lightest endpoint primary $\Delta_1^{0+}$. The only assumption about the defect-changing operator spectra is $\Delta_1^{++} \geq 1$, which is also encoded in the bound on $|\lambda_{\varepsilon 00}^{0+0}/\lambda_{\sigma 00}^{0+0}|$.

### 3.6 Defect-changing operator dimensions

We next discuss our bounds on the scaling dimensions of various defect-changing operators. Our most general results are upper bounds on $\Delta_1^{++}$, $\Delta_0^{+-}$, and $\Delta_1^{0+}$ as a function of $\Delta_0^{0+}$, where we make no assumptions about the defect-changing spectrum other than $\Delta_1^{++} \geq 1$, in addition to making use of the OPE ratio bounds of Fig. 12. These bounds are obtained by imposing (3.12–3.20), (3.30), and (3.26) and finding the smallest disallowed choices of the gaps in the defect-changing sectors using a binary search. The bounds are shown in Fig. 15 and lead to the following general bounds

$$1 \leq \Delta_1^{++} \leq 1.625 \qquad \Delta_0^{+-} \leq 0.870 \qquad \Delta_1^{0+} \leq 2.210.$$

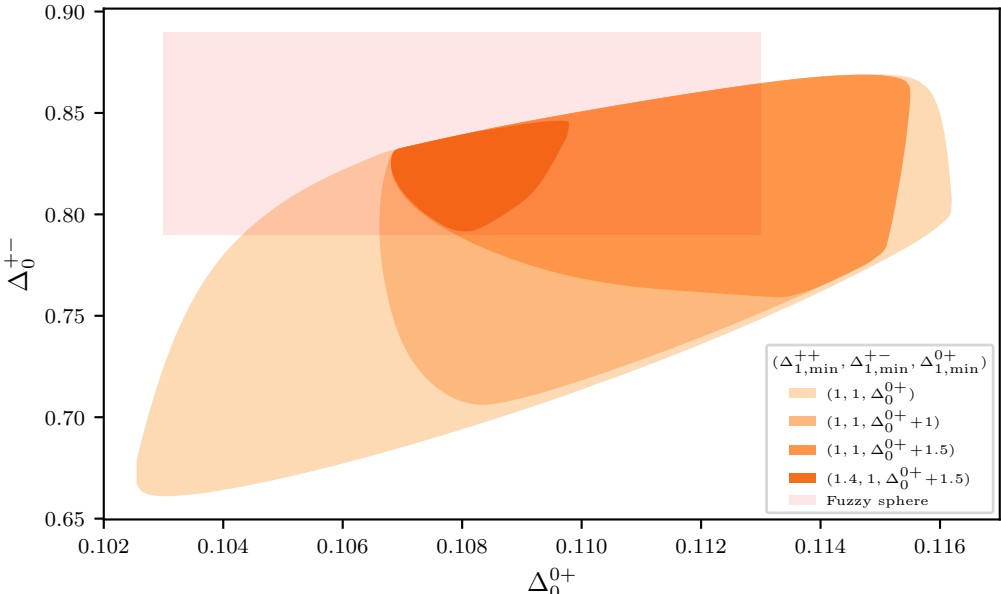

**Figure 16**: Allowed region for the leading domain wall operator with various non-generic gap assumptions. In all instances, we compute a bound assuming only a single relevant domain wall operator. For these bounds, we use the bounds on $r_{\sigma\varepsilon}^{\phi_0^{0+}}$ set according to those computed with matching $\Delta_{1,\min}^{++}$ and $\Delta_{1,\min}^{0+}$ from Fig. 12.

The values at which the above bounds are saturated, as a function of increasing $\Lambda$, agree well with the values predicted by the fuzzy sphere [27]

$$(\Delta_1^{++}, \Delta_0^{+-}, \Delta_1^{0+}) = (1.63(5), 0.84(5), 2.25(7)).$$

The upper bounds on $\Delta_1^{++}$ and $\Delta_1^{0+}$ additionally exhibit sharp kinks in the regions where the fuzzy sphere predicts these quantities to lie, giving further evidence that our bounds are sensitive to the solution corresponding to the true pinning field defect fixed point. There is actually a kink in the $\Delta_0^{+-}$ upper bound as well, albeit a much less dramatic one. It can be seen more clearly in Fig. 16.

The most interesting consequence of these bounds is the proof, with no non-trivial defect-changing spectrum assumptions, that $\phi_0^{+-}$ is a *relevant* operator of the $\mathcal{D}^+ \oplus \mathcal{D}^-$ fixed point, thus ruling out this non-simple defect from supporting stable LRO. We note, however, that technically our results do not rule out the existence of defects $\tilde{\mathcal{D}}^\pm$ with $\Delta_0^{+-} > 1$ lying within the "continent" in Fig. 12 (left) that are not described by an RG flow starting from the trivial line defect in the 3d Ising model. Thus, we cannot categorically rule out the existence of stable LRO defects in the 3d Ising model.

To obtain more refined estimates of $\Delta_0^{+-}$, we further impose $\Delta_1^{+-} \geq 1$, amounting to

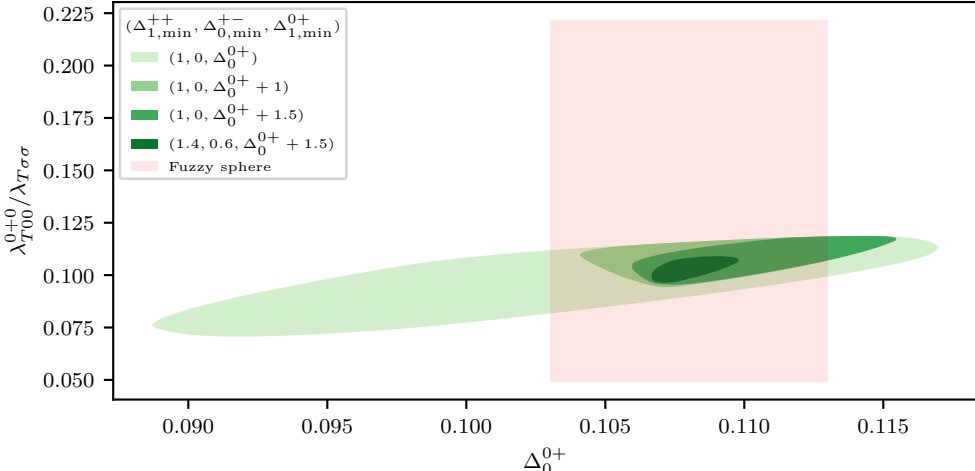

**Figure 17**: The allowed region for the ratio of OPE coefficients $\lambda^{0+0}_{T00}/\lambda_{T\sigma\sigma}$. The bounds are computed at $\Lambda = 45$. The fuzzy sphere data used to compute this ratio are taken from [27, 80].

assuming only a single relevant domain wall is present, as suggested by the fuzzy sphere and $\epsilon$-expansion. The new allowed region in $\Delta^{+-}_0$ is now bounded from below due to these secondary gap assumptions. Note that just the assumptions $\Delta^{+-}_1 \geq 1$, $\Delta^{++}_1 \geq 1$ automatically give $\Delta^{+-}_0 > 0.65$, confirming our strongest assumption on $\Delta^{+-}_0$ in the rest of the paper.

The gap assumptions in Fig. 16 are implemented by respectively replacing (3.20) with

$$\alpha[\mathbf{V}^{+-}_{\Delta^{+-}_0}] \geq 0 \tag{3.33}$$

$$\alpha[\mathbf{V}^{+-}_{\Delta}] \geq 0 \qquad \Delta \geq 1. \tag{3.34}$$

All the other constraints are imposed in the same way as before, and we use the "cutting curve" algorithm again to obtain the final islands using $\Delta^{+-}_0$ as scanning parameters. We also attempted to obtain an island for $\Delta^{++}_1$, but using a conservative choice such as $\Delta^{++}_{2,\min} = 2$ did not yield an improved lower bound on $\Delta^{++}_1$ beyond what we assume to obtain our strongest bounds, i.e. $\Delta^{++}_{1,\min} = 1.4$.

### 3.7  Stress tensor OPE ratio

A final quantity that we bound is $\lambda^{0+0}_{T00}/\lambda_{T\sigma\sigma}$, representing the ratio of OPE coefficients of the stress tensor appearing in the endpoint vs. $\sigma$ OPEs. We obtain the bound in an essentially identical manner as described earlier in the case of $|\lambda^{0+0}_{\varepsilon00}/\lambda^{0+0}_{\sigma00}|$, except we scan over $\theta^T_{\phi^{0+}_0 \sigma}$

defined via $\tan\theta^T_{\phi_0^{0+}\sigma} = \lambda^{0+0}_{T00}/\lambda_{T\sigma\sigma}$ using the crossing vector $\mathbf{V}_T(x,\theta^T_{\phi_0^{0+}\sigma})$, defined as

$$\mathbf{V}_T(x,\theta^T_{\phi_0^{0+}\sigma}) = M_T(\theta^T_{\phi_0^{0+}\sigma}) \cdot \mathbf{V}_T(x) \cdot M_T(\theta^T_{\phi_0^{0+}\sigma})^\mathsf{T} \qquad M_T(\theta^T_{\phi_0^{0+}\sigma}) = \left(\sin\theta^T_{\phi_0^{0+}\sigma} \ \cos\theta^T_{\phi_0^{0+}\sigma}\right).$$
(3.35)

The results are shown in Fig. 17, yielding the range

$$0.09629 \le \lambda^{0+0}_{T00}/\lambda_{T\sigma\sigma} \le 0.10902$$
(3.36)

when computed using the strongest assumptions. Even our agnostic bound gives a range on this quantity that is considerably tighter than the estimate computed using fuzzy sphere data[21]. Our bound indicates that $\lambda^{0+0}_{T00}/\lambda_{T\sigma\sigma}$ takes a value that is less than the analogous quantity computed for distinct bulk scalar primary operators $\mathcal{O}_1, \mathcal{O}_2$, $\frac{\lambda_{T\mathcal{O}_1\mathcal{O}_1}}{\lambda_{T\mathcal{O}_2\mathcal{O}_2}} = \Delta_{\mathcal{O}_1}/\Delta_{\mathcal{O}_2}$. As we will discuss in section 4.4, in $D > 2$ when the three-point function $\langle\phi^{0+}(\infty)x^\mu x^\nu T_{\mu\nu}(x)\phi^{0+}(0)\rangle$ is extended beyond the collinear limit, unlike for the case of bulk scalar operators it is no longer uniform over the sphere $|x| = R$ and, in fact, diverges as $x$ approaches the line defect. Thus, while the integrated three point function of $T$ still satisfies (after an appropriate subtraction) the same Ward identity measuring the dilatation charge as for bulk operators, the relation $\frac{\lambda_{T\mathcal{O}_1\mathcal{O}_1}}{\lambda_{T\mathcal{O}_2\mathcal{O}_2}} = \Delta_{\mathcal{O}_1}/\Delta_{\mathcal{O}_2}$ generally does not hold for the collinear three point function. We will demonstrate these conclusions explicitly within the $4 - \epsilon$ expansion in section 4.4.

# 4 The pinning field defect in $D = 4 - \epsilon$

In this section we study the pinning field defect of the Ising model in dimension $D = 4 - \epsilon$, which allows us to obtain perturbative results in $D = 3$ upon extrapolation. Together with the exact results in $D = 2$ reviewed in section 2.6, our results here motivate the assumptions on the defect operator spectrum made in our numerical bootstrap analysis in section 3.

Our starting point will be massless, real $\phi^4$-theory in $D = 4 - \epsilon$ dimension containing also a term that creates the pinning-field defect along the $\tau$ axis

$$S = \int \left[\frac{1}{2}(\partial\phi)^2 + \frac{\lambda_0}{4!}\phi^4\right] d^D x + \int h(\tau)\phi\, d\tau.$$
(4.1)

Note that we allow $h(\tau)$ to be non-uniform; this will let us create different defect configurations such as domain walls and endpoints, depending on the chosen profile. We will treat both the bulk $\phi^4$ interaction and the pinning field defect term as perturbations of the Gaussian fixed point, using the bulk propagator of the Gaussian theory to compute the required Feynman diagrams

$$G(|x - y|) \equiv \langle\phi(x)\phi(y)\rangle = \frac{c(D)}{|x - y|^{D-2}}, \qquad c(D) = \frac{\Gamma(D/2 - 1)}{4\pi^{D/2}}.$$
(4.2)

---

[21]The large error in this ratio is due primarily to the large relative error for $\lambda^{0+0}_{T00} = 0.044(28)$ reported in [27], which we propagate to the range shown in fig. 17. No error bar was reported in [80] for $\lambda_{T\sigma\sigma}$ so we treat the reported value as a constant in computing the fuzzy sphere error window shown in fig. 17.

Both $\lambda$ and $h$ flow under RG, but the flow of $\lambda$ is unaffected by the presence of the defect. While the physical context and motivation were somewhat different from our work, this particular defect was first studied in [29], where it was shown that the defect RG fixed point, along with the already known bulk fixed-point, occurs at

$$\frac{\lambda_{r,*}}{(4\pi)^2} = \frac{1}{3}\epsilon + \frac{17}{81}\epsilon^2 + O(\epsilon^3), \tag{4.3}$$

$$h_{r,*}^2 = 9 + \frac{73}{6}\epsilon + O(\epsilon^2). \tag{4.4}$$

Here the subscript $r$ denotes renormalized coupling constants in the minimal subtraction scheme, see appendix D for details. In contrast, bare coupling constants are denoted with a subscript 0.

As a final preliminary remark, we point out the tree-level calculation of the one-point function of $\phi$ gives

$$\phi_{\text{cl}}(x) = -\int d\tau' G(x - (\tau', 0))h(\tau'). \tag{4.5}$$

We will make extensive use of this quantity in what follows.

Using this setup, prior works have analytically computed the scaling dimensions $\Delta_0^{0+}$, $\Delta_0^{+-}$ of the leading defect endpoint [29] and domain wall [28] operators, as well as the defect $g$-function [23], all of which are roughly in agreement with our previous bootstrap calculations. As a warm up, in section 4.1 we will reproduce the previous perturbative results for $\Delta_0^{0+}$ and $\Delta_0^{+-}$. In sections 4.2, 4.3, we will generalize the perturbative calculations to include other defect quantities, such as the scaling dimension of the next defect-endpoint primary operator $\Delta_1^{0+}$ and OPE coefficients $\lambda_{\sigma00}^{0+0}$ and $\lambda_{\varepsilon00}^{0+0}$, also to compare with our bootstrap results. To our knowledge, these results are new. Finally, in section 4.4 we discuss the Ward identity satisfied by the correlation function of the stress-tensor with defect-changing operators, and illustrate how it is saturated in perturbation theory.

## 4.1 Leading endpoint and domain wall scaling dimensions to $O(\epsilon)$

First we will study the scaling dimensions of the leading defect creation (endpoint) operator $\phi_0^{0+}$ and domain wall operator $\phi_0^{+-}$. The two point function of these operators can be obtained simply by computing the partition function of the theory with different choices of $h(\tau)$ in (4.1) of the form

$$h^{0+}(\tau) = \begin{cases} h_0, & 0 < \tau < L, \\ 0, & \text{otherwise} \end{cases} \tag{4.6}$$

for the leading endpoint operator and

$$h^{+-}(\tau) = \begin{cases} -h_0, & 0 < \tau < L, \\ h_0, & \text{otherwise}. \end{cases} \tag{4.7}$$

for the leading domain wall. We may treat the partition function as a functional of the field profile $Z[h(\tau)]$, further defining $Z \equiv Z[0]$, $Z^{++} \equiv Z[h]$, $Z^{+-} \equiv Z[h^{+-}(\tau)]$, and $Z^{0+} \equiv Z[h^{0+}(\tau)]$.

For the domain wall case, we expect

$$\frac{Z^{+-}}{Z^{++}} \sim \frac{1}{L^{2\Delta_0^{+-}}}, \tag{4.8}$$

where $\Delta_0^{+-}$ is the scaling dimension of the leading domain wall operator. For the defect creation operator, we expect the partition function to take the form

$$\frac{Z^{0+}}{Z} \approx \frac{e^{-ML}}{L^{2\Delta_0^{0+}}}, \tag{4.9}$$

where $M$ is the non-universal free-energy density of the defect.

To zeroth order in $\lambda$ and $\epsilon$, we have

$$\log Z[h(\tau)] = \frac{1}{2} \int d\tau_1 d\tau_2 h(\tau_1) h(\tau_2) G(\tau_1 - \tau_2) \tag{4.10}$$

where $G(x) = \frac{1}{4\pi^2 x^2}$. Thus,

$$\log \frac{Z^{+-}}{Z^{++}} = \frac{1}{2} \int d\tau_1 d\tau_2 (h(\tau_1) h(\tau_2) - h_0^2) G(\tau_1 - \tau_2) = -2h_0^2 \int_0^L d\tau_1 \left[ \int_L^\infty d\tau_2 + \int_{-\infty}^0 d\tau_2 \right] G(\tau_1 - \tau_2)$$

$$= \frac{-h_0^2}{\pi^2} \log \Lambda L, \tag{4.11}$$

where $\Lambda$ is a UV cutoff used to regulate the divergence at the domain wall. Thus, inserting the fixed point value of $h$ in Eq. (4.4), we obtain

$$\Delta_0^{+-} = \frac{h_{r,*}^2}{2\pi^2} + O(\epsilon) = \frac{9}{2\pi^2} + O(\epsilon) \approx 0.465945 + O(\epsilon). \tag{4.12}$$

An analogous calculation for the defect creation operator gives

$$\Delta_0^{0+} = \frac{h_{r,*}^2}{8\pi^2} + O(\epsilon) = \frac{9}{8\pi^2} + O(\epsilon) \approx 0.113986 + O(\epsilon). \tag{4.13}$$

We provide details of the rather lengthy calculations $Z^{0+}$ and $Z^{+-}$ to $O(\lambda)$ in Appendix D. The results of the calculations are the determinations of $\Delta_0^{0+}$ and $\Delta_0^{+-}$ to $O(\epsilon)$:

$$\Delta_0^{0+} = \frac{9}{8\pi^2} \left( 1 + \epsilon \left( \frac{73}{54} + \frac{1}{2} \left( \frac{3\zeta(3)}{\pi^2} - 1 \right) \right) \right) \approx 0.113986(1 + 1.03454\epsilon),$$

$$\Delta_0^{+-} = \frac{9}{2\pi^2} \left[ 1 + \epsilon \left( \frac{73}{54} + \frac{1}{2} \left( \frac{12\zeta(3)}{\pi^2} - 1 \right) \right) \right] \approx 0.455945(1 + 1.58261\epsilon)$$

Our calculation of $\Delta_0^{0+}$ agrees with Ref. [29] and the calculation of $\Delta_0^{+-}$ agrees with Ref. [28]. As was already pointed out in Ref. [28], the $O(\epsilon)$ correction to $\Delta_0^{0+}$ is positive, while $\Delta_0^{0+}(D =$

$2) = \frac{1}{32} < \Delta_0^{0+}(D = 4^-)$. This means that $\Delta_0^{0+}$ exhibits non-monotonic dependence on $D$. Numerically, the $O(\epsilon)$ corrections to both $\Delta_0^{0+}$ and $\Delta_0^{+-}$ are large and mainly come from the large $O(\epsilon)$ correction to $h_{r,*}^2$ in Eq. (4.4). However, this $O(\epsilon)$ correction to $h_{r,*}^2$ cancels out when we consider

$$\frac{\Delta_0^{0+}}{\Delta_0^{+-}} = \frac{1}{4}\left(1 - \frac{9\zeta(3)}{2\pi^2}\epsilon + O(\epsilon^2)\right) \approx \frac{1}{4}(1 - 0.548072\epsilon) \overset{\epsilon \to 1}{\approx} \approx 0.113 \tag{4.14}$$

Padé resumming and matching to the value $\frac{\Delta_0^{0+}}{\Delta_0^{+-}} = \frac{1}{32}$ at $D = 2$,

$$\left[\frac{\Delta_0^{0+}}{\Delta_0^{+-}}\right]_{1,1} \approx 0.128. \tag{4.15}$$

This can be compared to our rigorous bootstrap result in the first column in Table 2:

$$\Delta_0^{0+} = 0.1079(19), \quad \Delta_0^{+-} = 0.818(28), \quad \frac{\Delta_0^{0+}}{\Delta_0^{+-}} = 0.132(7). \tag{4.16}$$

Thus, the direct $O(\epsilon)$ value (4.14) is already quite close to the bootstrap result, and the $[1, 1]$ Padé resummation agrees with the bootstrap within error bars.

As was pointed out in Ref. [55], in contrast to the non-monotonic behavior of $\Delta_0^{+0}$ across $D$, the $\epsilon$-expansion results for $\Delta_0^{+-}$ are consistent with $\Delta_0^{+-}$ monotonically increasing as $D$ decreases from $D = 4^-$ to $D = 2$. Directly Padé resumming $\Delta_0^{+-}$ and matching to $\Delta_0^{+-} = 1$ in $D = 2$ gives [28]

$$[\Delta_0^{+-}]_{1,1} \approx 0.85, \tag{4.17}$$

in good agreement with our bootstrap result in Table 2.

## 4.2 Next endpoint primary

We now consider the next defect endpoint operator. In the WF theory, such operators can be obtained by fusing powers of $\phi$ and its derivatives with the "minimal" endpoint (implemented via (4.6)). The lowest such operator involves insertion of $\phi$ at the endpoint, which we denote $\phi \cdot \phi_0^{+0}$. However, it is clear from the equation of motion that this is just the first descendant of the minimal endpoint. The next $SO(D - 1)$ singlet endpoint operators are

$$O_1 = \phi^2 \cdot \phi_0^{+0}, \qquad O_2 = \partial_\tau \phi \cdot \phi_0^{+0}. \tag{4.18}$$

To zeroth order in $\epsilon$ both have dimension $\Delta_1^{0+} = \Delta_0^{0+} + 2$. We now calculate first correction in $\epsilon$ to the dimension of these operators. We expect that after taking into account their mixing under RG, one will become the second descendant of $\phi_0^{+0}$ and the other will be a primary. We, indeed, confirm this to order $\epsilon$ and find that the first subleading endpoint primary has dimension

$$\Delta_1^{0+} = \Delta_0^{0+} + 2 + \left(\frac{1}{3} + \frac{3}{4\pi^2}\right)\epsilon + O(\epsilon^2), \tag{4.19}$$

see appendix D.4 for the calculation details. We know that in $D = 2$, the subleading $SL(2, \mathbb{R})$ primary has $\Delta_1^{0+} = \Delta_0^{0+} + 2$, which gives the second term on the RHS of Eqs. (2.47), (2.48), (2.49) (there is also a primary operator with dimension $\Delta_0^{0+} + 3/2$, but it is odd under reflection across the defect axis). Thus, as we lower $D$ from 4 to 2, $\Delta_1^{0+} - \Delta_0^{0+}$ initially increases above 2 and then decreases back to 2. Plugging $\epsilon = 1$ directly into (4.19) gives $\Delta_1^{0+} \approx \Delta_0^{0+} + 2.41$. This is slightly bigger than our bootstrap result $\Delta_1^{0+} - \Delta_0^{0+} \approx 2.10$ from $g$-minimization, see Tab. 2. The fact that $\Delta_1^{0+} - \Delta_0^{0+} = 2$ in $D = 4^-$ and increases with $\epsilon$ makes the strongest gap assumption in this sector employed in our bootstrap study $\Delta_1^{0+} - \Delta_0^{0+} \geq 1.5$ appear quite conservative.

In principle, a similar analysis may be carried out for subleading domain wall operators. Again, the lowest dimension subleading operator $\phi \cdot \phi_0^{+-}$ is an $SL(2, \mathbb{R})$ descendant of $\phi_0^{+-}$ by the equation of motion. The next $SO(D-1)$ singlet domain wall operators are again $\phi^2 \cdot \phi_0^{+-}$ and $\partial_\tau \phi \cdot \phi_0^{+-}$ with dimensions $\Delta_1^{+-} = \Delta_0^{+-} + 2 + O(\epsilon)$. Likewise, in $D = 2$ the transverse reflection even subleading domain wall has $\Delta_1^{+-} = \Delta_0^{+-} + 2$, as manifested in the second term on the RHS of Eq. (2.59). While we will not attempt to compute the $O(\epsilon)$ correction to $\Delta_1^{+-}$ here, the above observations are in accord with the large value of $\Delta_1^{+-} \approx 3.2355$ obtained by our bootstrap $g$-minimization in Tab. 2, and $\Delta_1^{+-} \approx 3.167$ obtained by fuzzy sphere [27]. In this light, the gap assumption $\Delta_1^{+-} \geq 1$ made in Fig. 16 is very conservative.

## 4.3 OPE coefficients

In this section we compute the OPE coefficients $\lambda_{\sigma 00}^{0+0}$ and $\lambda_{\varepsilon 00}^{0+0}$. Consider the collinear three point function of a bulk scalar primary $\mathcal{O}(x)$ and two lowest defect endpoint operators normalized by the two point function of defect endpoint operators:

$$\frac{\langle \phi_0^{0+}(L)\phi_0^{+0}(0)\mathcal{O}(\tau)\rangle}{\langle \phi_0^{0+}(L)\phi_0^{+0}(0)\rangle} = \frac{\lambda_{\mathcal{O}00}^{0+0}}{(-\tau)^{\Delta_\mathcal{O}}(1-\tau/L)^{\Delta_\mathcal{O}}}. \tag{4.20}$$

Here we have assumed $\tau < 0$. The LHS is the expectation value of $\mathcal{O}$ in the presence of the background field $h(\tau)$ given by Eq. (4.6): $\frac{\langle\mathcal{O}(\tau)\rangle_h}{\langle 1\rangle_h}$. Further, we may take the limit $L \to \infty$, $\tau$ - fixed to simplify the calculation. Proceeding in this manner, we obtain

$$\lambda_{\sigma 00}^{0+0} = \frac{3}{2\pi}\left[1 + \epsilon\left(\frac{25}{27} + \frac{3\zeta(3)}{\pi^2}\right) + O(\epsilon^2)\right] \overset{\epsilon \to 1}{=} 1.09402, \tag{4.21}$$

$$\lambda_{\varepsilon 00}^{0+0} = \frac{9\sqrt{2}}{8\pi^2}\left[1 + \epsilon\left(\frac{32}{27} + \frac{6\zeta(3)}{\pi^2}\right) + O(\epsilon^2)\right] \overset{\epsilon \to 1}{=} 0.470054. \tag{4.22}$$

Using $\lambda_{\sigma 00}^{0+0} = 1$, $\lambda_{\varepsilon 00}^{0+0} = \frac{1}{4}$ at $D = 2$ and Pade-resumming to match this value, we obtain:

$$[\lambda_{\sigma 00}^{0+0}]_{1,1} \approx 0.845, \qquad [\lambda_{\varepsilon 00}^{0+0}]_{1,1} \approx 0.239. \tag{4.23}$$

The fuzzy sphere values are [27]:

$$\lambda_{\sigma 00}^{0+0} = 0.869(19), \quad \lambda_{\varepsilon 00}^{0+0} = 0.3334(9). \tag{4.24}$$

Thus, for $\lambda^{0+0}_{\sigma 00}$ both $[1,1]$ and $[2,0]$ Pade estimates are close to the fuzzy sphere result, while for $\lambda^{0+0}_{\varepsilon 00}$ the $[2,0]$ estimate is close.

We may also look at the ratio $(\lambda^{0+0}_{\sigma 00})^2/\lambda^{0+0}_{\varepsilon 00}$, where the numerically large $O(\epsilon)$ correction due to the renormalization of $h^2_{r,*}$ cancels out. We have

$$(\lambda^{0+0}_{\sigma 00})^2/\lambda^{0+0}_{\varepsilon 00} = \sqrt{2}(1 + \frac{2}{3}\epsilon + O(\epsilon^2)) \overset{\epsilon \to 1}{=} 2.35702. \tag{4.25}$$

which is rather close to the fuzzy sphere value,

$$(\lambda^{0+0}_{\sigma 00})^2/\lambda^{0+0}_{\varepsilon 00} = 2.254(2). \tag{4.26}$$

Pade-resumming fixing the value at $D = 2$ actually makes the agreement a bit worse:

$$\left[(\lambda^{0+0}_{\sigma 00})^2/\lambda^{0+0}_{\varepsilon 00}\right]_{1,1} = 2.50466, \quad \left[(\lambda^{0+0}_{\sigma 00})^2/\lambda^{0+0}_{\varepsilon 00}\right]_{2,0} = 2.53206, \quad \left[(\lambda^{0+0}_{\sigma 00})^2/\lambda^{0+0}_{\varepsilon 00}\right]_{0,2} = 2.80012. \tag{4.27}$$

### 4.4 Stress-energy tensor OPE coefficients

An interesting OPE coefficient to study is the coefficient of the stress-energy tensor in the OPE of defect endpoints. For any bulk, scalar operator $\mathcal{O}$ with scaling dimension $\Delta$, the OPE coefficient of the stress tensor in $\bar{\mathcal{O}} \times \mathcal{O}$ is a universal quantity fixed by the conformal Ward identity

$$\lambda_{T\bar{\mathcal{O}}\mathcal{O}} = -\frac{D}{D-1}\frac{\Delta}{S_{D-1}}.$$

In the presence of a non-topological line defect, such a universal result for the OPE no longer holds for a defect-changing operator $\mathcal{O}^{ab}$, however we still want to claim

$$\int_S dS_\mu x^\nu T_{\mu\nu}(x) O^{ab}(0) = -\Delta^{ab} O^{ab}(0) \tag{4.28}$$

holds as an operator equation for both local operators and defect changing operators, when the integration surface $S$ encloses the origin. In fact, such a relation appears necessary if we want the state-operator correspondence for defect operators to be present: the energy of states on a sphere of radius $R$ with defects of types $a$ and $b$ at north and south poles is equal to $\Delta^{ab}_i/R$.

In general, in the presence of a defect a topological symmetry operator may require some counterterms to be tuned away at the junction between the defect and the symmetry operator, in order to preserve the topological property. One issue with Eq. (4.28) is the possibility of such a counter-term at the intersection of the surface $S$ with the defect line [81]. Consider adding

$$\delta T_{\mu\nu}(x) = \delta_{\mu 0}\delta_{\nu 0}\delta^{D-1}(x)\hat{A}(\tau) \tag{4.29}$$

where $\hat{A}$ is some defect operator (here we imagine the defect running along the $\tau$ axis). If the dimension of $\hat{A}$ is bigger than 1 then the addition of $\hat{A}$ will only give rise to subleading

corrections to scaling. For the pinning field defect, in $D < 4$ all the defect operators (except identity) are irrelevant, so we are safe. However, as $D \to 4$, we have a nearly marginal defect operator (exactly marginal when $D = 4$). At fixed order in direct $\epsilon$-expansion, the contribution of operator $\hat{A}$ will not be suppressed. We now treat this issue and demonstrate by an explicit computation that the identity (4.28), indeed, holds for $O^{ab} = \phi_0^{+0}$ to leading order in $\epsilon$.

Consider the correlator ratio

$$T(\theta) = \lim_{L \to \infty} R^{D-2} \frac{\langle \phi^{0+}(L) x^\mu x^\nu T_{\mu\nu}(x) \phi^{+0}(0) \rangle}{\langle \phi^{0+}(L) \phi^{+0}(0) \rangle} \tag{4.30}$$

with the defect operators inserted along the $\vec{x} = 0$ axis with $\tau \geq 0$ and $\tau^2 + |\vec{x}|^2 = R^2$. Here $\theta$ is the "polar" angle between the $\tau$ axis and $x$. $T(\theta)$ is a universal dimensionless function in the defect CFT and formally the identity (4.28) implies

$$\int_{S^{D-1}} T(\theta) d\Omega = -\Delta_0^{0+}, \tag{4.31}$$

where the integral is over a unit sphere around the origin $(d\Omega = \Omega_{D-2}(\sin \theta)^{D-2} d\theta)$. Let's now consider the behavior of $T(\theta)$ in the limit of $\theta \to 0$. For this, we need the bulk to (infinite) defect OPE of $T_{\mu\nu}(x)$. In $D = 4 - \epsilon$ the lowest non-trivial defect operator $\phi_1^{++}$ is an $SO(D-1)$ singlet and has dimension $\Delta_1^{++} = 1 + \epsilon + O(\epsilon^2)$, as can be read off from the $\beta$-function (D.1). All the other defect operators have dimensions $\Delta \geq 2 + O(\epsilon)$. Thus,

$$T_{\tau\tau}(\tau, \vec{x}) = a|\vec{x}|^{-D} + b|\vec{x}|^{-D+\Delta_1^{++}} \phi_1^{++}(\tau) + \ldots, \quad \vec{x} \to 0, \tag{4.32}$$

where $a, b$ are universal OPE coefficients. Inserting this OPE into $T(\theta)$, the leading contribution to $T(\theta) d\Omega$ as $\theta \to 0$ is

$$T(\theta) d\Omega = \Omega_{D-2} \left( a\theta^{-2} + b\lambda_{010}^{++0} \theta^{\Delta_1^{++}-2} + \ldots \right) d\theta, \tag{4.33}$$

where $\lambda_{010}^{++0}$ is the three point function $\langle \phi_0^{0+} \phi_1^{++} \phi_0^{+0} \rangle$. Note that the operators $T_{\tau i}$ and $T_{ij}$ do not contribute to the two leading terms in Eq. (4.33). Clearly, the integral in (4.31) diverges as $\theta \to 0$ due to the leading term in (4.33). If we cut off the integral at a distance $\rho_0 \ll R$ away from the defect (corresponding to minimal angle $\sin \theta_0 = \rho_0/R$),

$$\int_{\rho > \rho_0} T(\theta) d\Omega = \Omega_{D-2} \left( \frac{a}{\theta_0} - \frac{b\lambda_{010}^{++0}}{\Delta_1^{++} - 1} \theta_0^{\Delta_1^{++}-1} \right) + C + \ldots, \tag{4.34}$$

where $C$ does not scale with $\theta_0$ and ellipses denote terms that are further power law suppressed as $\theta_0 \to 0$. The $1/\theta_0 \sim R/\rho_0$ divergence simply corresponds to the non-universal "cosmological constant" in the defect free energy. Eq. (4.31) should be understood as $C = -\Delta_0^{0+}$. Note that since $\Delta_1^{++} > 1$, the second term in parenthesis in Eq. (4.34) goes to zero as $\theta_0 \to 0$ when $\epsilon$ is finite, thus, strictly speaking

$$C = \lim_{\theta_0 \to 0} \left[ \int_{\theta > \theta_0} T(\theta) d\Omega - \frac{a\Omega_{D-2}}{\theta_0} \right]. \tag{4.35}$$

However, since $\Delta_1^{++} = 1 + \epsilon + O(\epsilon^2)$, in a direct perturbation theory expansion, $\theta_0^{\Delta_1^{++}-1}$ will become a sum of powers of $\log \theta_0$ that re-sum to a positive power law. Thus, to compute $C$ in $\epsilon$ expansion, we instead look at

$$C = \lim_{\theta_0 \to 0} \left[ \int_{\theta > \theta_0} T(\theta) d\Omega - \frac{a\Omega_{D-2}}{\theta_0} + \frac{b\lambda_{010}^{++0}\Omega_{D-2}}{\Delta_1^{++} - 1} \theta_0^{\Delta_1^{++}-1} \right], \qquad (4.36)$$

where the expression in square brackets can be directly expanded order by order in $\epsilon$ and leading corrections to the $\theta_0 \to 0$ limit go as first power of $\theta_0$ (up to logarithms).

Utilizing this strategy, we now demonstrate $C = -\Delta_0^{0+}$ to zeroth order in $\epsilon$. This will require the knowledge of $T(\theta)$ to zeroth order in $\epsilon$ and $b\lambda_{010}^{++0}$ to first order in $\epsilon$. The stress tensor of the $\phi^4$ theory is

$$T_{\mu\nu} = \partial_\mu\phi\partial_\nu\phi - \frac{\delta_{\mu\nu}}{2}(\partial_\rho\phi)^2 - \frac{\lambda_0}{4!}\delta_{\mu\nu}\phi^4 + \frac{1}{4}\frac{D-2}{D-1}(\delta_{\mu\nu}\partial^2 - \partial_\mu\partial_\nu)\phi^2$$

$$= \frac{D}{2(D-1)}\partial_\mu\phi\partial_\nu\phi - \frac{1}{2}\frac{D-2}{D-1}\phi\partial_\mu\partial_\nu\phi + \delta_{\mu\nu}\left(-\frac{1}{2(D-1)}(\partial_\rho\phi)^2 + \frac{\lambda_0}{4!}\frac{D-3}{D-1}\phi^4\right) (4.37)$$

where we have used the equation of motion $\partial^2\phi = \frac{\lambda_0}{6}\phi^3$ in the last step[22]. Substituting $\phi_{cl}(x)$, Eqs. (4.5), (D.75) (with $L \to \infty$) into $T_{\mu\nu}$ gives to zeroth order in $\epsilon$

$$[T(\theta)]_0 = -\frac{3}{32\pi^4 \sin^4\theta}\left((\pi - \theta + \frac{1}{2}\sin 2\theta)^2 + \sin^4\theta\right) + O(\epsilon). \qquad (4.38)$$

Clearly, $T(\theta)$ has non-trivial $\theta$-dependence, unlike it would if $\phi^{0+}$ and $\phi^{+0}$ in Eq. (4.30) were replaced by bulk spin zero operators. To this order in $\epsilon$, $\lambda_{00T_{\tau\tau}}^{+00} = T(\theta = \pi) \approx -\frac{\Delta_0^{0+}}{6|\Omega_{D-1}|}$, a factor of 6 smaller than it would be if $T(\theta)$ was uniform and Eq. (4.28) held.[23] From the limit $\theta \to 0$ of $T(\theta)$ we also find that the coefficients $a$ and $b$ in Eq. (4.33) take values $a = -\frac{3}{32\pi^2} + O(\epsilon)$, $b\lambda_{010}^{++0} = O(\epsilon)$. In appendix D.6 we explicitly compute the first order term in $\epsilon$ in $b\lambda_{010}^{++0}$.

Integrating $[T(\theta)]_0$ over the unit sphere, as $\theta_0 \to 0$,

$$\int_{\theta > \theta_0} [T(\theta)]_0 \, d\Omega - \frac{a\Omega_{D-2}}{\theta_0} = \frac{3}{8\pi^2} \approx \frac{\Delta_0^{0+}}{3}. \qquad (4.39)$$

To obtain $C$ at order $\epsilon^0$ we must add to the above result the last term in Eq. (4.36) expanded to order $\epsilon^0$: $\frac{b\lambda_{010}^{++0}\Omega_{D-2}}{\Delta_1^{++}-1}\theta_0^{\Delta_1^{++}-1} \approx \frac{4\pi b\lambda_{010}^{++0}}{\epsilon}$. In appendix D.6 we explicitly compute,

$$\lambda_{010}^{++0} = -\frac{3}{2\pi} + O(\epsilon), \qquad b = \frac{\epsilon}{4\pi^2} + O(\epsilon^2). \qquad (4.40)$$

Thus, the last term in Eq. (4.36) contributes $-\frac{3}{2\pi^2} \approx -\frac{4}{3}\Delta_0^{0+}$ to $C$. Combining this with Eq. (4.39), to $O(\epsilon^0)$, $C = -\Delta_0^{0+}$, as claimed.

---

[22]Strictly speaking, the coefficient of the term $\frac{1}{4}\frac{D-2}{D-1}(\delta_{\mu\nu}\partial^2 - \partial_\mu\partial_\nu)\phi^2$ needs to be further tuned beyond the free-theory form to make the stress-tensor traceless [82]. However, this can be ignored to the order in $\epsilon$ that we are studying here.

[23]Here in $\lambda_{00T_{\tau\tau}}^{+00}$ we are using the normalization of $T_{\tau\tau}$ as in Eq. (4.37), rather than normalizing its two-point function along the $\tau$ direction to identity. We also use the convention in Eq. (2.13) for collinear three point functions, as usual.

## 5 Discussion

In this work, we have demonstrated the power of modern numerical conformal bootstrap tools to probe the properties of conformal line defects in specific CFTs in higher dimensions. We focused on the case of the pinning field defect in the $3d$ Ising CFT, giving essentially rigorous bounds on a variety of the most important quantities characterizing this defect. Our most physically important result is the proof that the non-simple "LRO defect" $\mathcal{D}^+ \oplus \mathcal{D}^-$ related to the pinning field defect is not stable due to the RG-relevance of the domain-wall creation operator. The only assumptions underlying this result are that the $3d$ Ising pinning field defect lies inside the bootstrap island that we find, and further that the bulk $3d$ Ising CFT data that enter our calculations are sufficiently close to the true values. Our work paves the way for future studies of this defect that use even more sophisticated bootstrap methods, which can yield estimates of the dCFT data with much higher precision.

There are a number of interesting directions that are worth more study in the future. One direction is to pursue improvements to our setup. Some options that stand out are firstly to include more external defect-changing operators such as the displacement operator, $\phi_1^{++}$, or $\phi_0^{+-}$. While the dimensions of these operators are quite large compared to that of $\phi_0^{0+}$, making them generally somewhat less constraining for the quantities studied in this work without more assumptions, including them opens the possibility of studying a number of new dCFT quantities. Including the displacement operator is especially interesting since its correlation functions are strongly constrained by bulk conformal invariance [83, 84]. Another interesting possibility is to include the full $3d$ conformal blocks in correlation functions that only contain bulk operators. One can fantasize that, since augmenting the $3d$ bulk Ising bootstrap setup with the defect-changing operators of the pinning field strictly strengthens any bulk bounds in principle, the complete system of bulk and defect-changing operators could lead to a shrinkage of the $3d$ Ising bulk bootstrap allowed region compared with the existing, purely bulk results [48, 76]. To attempt this, it will be important to also upgrade to a navigator bootstrap setup to more efficiently scan over the rather large number of parameters that would be involved [74].

Beyond the $3d$ Ising pinning field, which is the simplest example of a line defect in a strongly coupled CFT in higher dimensions, a rich set of defects exist in other CFTs that merit future study. Spin impurities are a particularly physical example of a line defect, naturally appearing in e.g. the $O(N)$ WF theories, but are difficult to study analytically due to quantum effects [85–87]. Bootstrap does not suffer from issues of strong-coupling, so our method seems like an especially effective way to study these defects. Other theories that deserve attention from the perspective taken in this work are conformal gauge theories. The problem of bootstrapping such gauge theories has already proved to be a challenge; using the standard approach of imposing unitarity and crossing symmetry on correlation functions of local operators gives results that are not nearly as strong compared with e.g. the $O(N)$ CFTs [78, 88, 89]. One natural guess as to why previous bootstrap studies of gauge theories have not had as much success is that, when they have a flavor symmetry such as $PSU(N)$, the $SU(N)$

flavor fundamental is missing from the spectrum of local operators, but can alternatively be realized at the endpoint of a Wilson line. Using the techniques developed here adapted for e.g. Wilson line endpoints, which recently were studied perturbatively in [90], we expect that a deeper understanding of both the bulk and defect operator spectrum of conformal gauge theories can be achieved.

Finally, we mention that the problem of studying line defects in two-dimensional theories is still quite open. It has proven difficult to solve conformal boundary conditions beyond the rational boundaries of rational models and toroidal compactifications, so numerical bootstrap techniques are a natural next resort. There is a much richer set of constraints on line defects in two dimensions coming from e.g. the modular transformation properties of the annulus partition function [35]. We point out the recent work [91] as a concrete example of progress along this direction.

## Acknowledgements

We thank Gabriel Cuomo, Yin-Chen He, Zohar Komargodski and Zheng Zhou for discussions. We are grateful to the Kavli Institute for Theoretical Physics (KITP), where this work was initiated. We also thank the organizers of the workshops "Paths to quantum field theory" in Durham, UK, "Defects, from condensed matter to quantum gravity," Pollica Physics Centre, Italy, and the IHES summer school "Symmetries and anomalies: a modern perspective" in Bures-sur-Yvette, France for allowing the authors to interact in person. This research was supported in part by grant NSF PHY-2309135 to the KITP. SL acknowledges support from the Gordon and Betty Moore Foundation under Grant No. GBMF8690, the National Science Foundation under Grant No. NSF PHY-1748958, the Simons Foundation under an award to Xie Chen (Award No. 828078), and a start-up grant from the Institute of Physics at Chinese Academy of Sciences. MM was supported in part by the National Science Foundation under grant number DMR-1847861. The numerical calculations were performed on the Hyak computing cluster at the University of Washington (UW), funded by the UW student technology fee. Research at Perimeter Institute is supported in part by the Government of Canada through the Department of Innovation, Science and Economic Development and by the Province of Ontario through the Ministry of Economic Development, Job Creation and Trade.

## A  2D Ising Virasoro characters

$$\chi_0(\tau) = \frac{1}{2}\left(\frac{\vartheta_3(0, e^{\pi i \tau})}{\eta(\tau)}\right)^{1/2} + \frac{1}{2}\left(\frac{\vartheta_4(0, e^{\pi i \tau})}{\eta(\tau)}\right)^{1/2}$$

$$\chi_{1/2}(\tau) = \frac{1}{2}\left(\frac{\vartheta_3(0, e^{\pi i \tau})}{\eta(\tau)}\right)^{1/2} - \frac{1}{2}\left(\frac{\vartheta_4(0, e^{\pi i \tau})}{\eta(\tau)}\right)^{1/2}$$

$$\chi_{1/16}(\tau) = \frac{1}{\sqrt{2}}\left(\frac{\vartheta_2(0, e^{\pi i \tau})}{\eta(\tau)}\right)^{1/2}, \tag{A.1}$$

where $\vartheta_i(u, q)$ are Jacobi $\vartheta$-functions (see Ref. [92], section 8.180.)

## B  Crossing equations and crossing vectors

In terms of conformal blocks (2.20), the full set of crossing equations is

$$0 = \left(\frac{1}{g} \pm 1\right) F_{0,\mp}^{\phi\phi\phi\phi}(x) + \sum_{\mathcal{O}\in[++;r^\tau=0]} (\lambda_{\phi\phi\mathcal{O}}^{0++})^2 F_{\Delta,\mp}^{\phi\phi\phi\phi}(x) \pm \sum_{\mathcal{O}\in\left[00;{r^\tau=0 \atop q=0,1}\right]} (\lambda_{\phi\phi\mathcal{O}}^{+00})^2 F_{\Delta,\mp}^{\phi\phi\phi\phi}(x) \quad \text{(B.1)}$$

$$0 = \pm F_{0,\mp}^{\phi\phi\phi\phi}(x) + \sum_{\mathcal{O}\in[-+]} (\lambda_{\phi\phi\mathcal{O}}^{0-+})^2 F_{\Delta,\mp}^{\phi\phi\phi\phi}(x) \pm \sum_{\left[00;{r^\tau=0 \atop q=0,1}\right]} (-1)^q (\lambda_{\phi\phi\mathcal{O}}^{+00})^2 F_{\Delta,\mp}^{\phi\phi\phi\phi}(x) \quad \text{(B.2)}$$

$$0 = \pm F_{0,\mp}^{\sigma\sigma\phi\phi}(x) + \sum_{\mathcal{O}\in[0+]} |\lambda_{\sigma\phi\mathcal{O}}^{00+}|^2 F_{\Delta,\mp}^{\sigma\phi\phi\sigma}(x) \pm \sum_{\mathcal{O}\in\left[00;{r^\tau=0 \atop q=0}\right]} \lambda_{\phi\phi\mathcal{O}}^{+00}\lambda_{\sigma\sigma\mathcal{O}} F_{\Delta,\mp}^{\sigma\sigma\phi\phi}(x) \quad \text{(B.3)}$$

$$0 = \sum_{\mathcal{O}\in[0+]} \lambda_{\epsilon\phi\mathcal{O}}^{00+}(\lambda_{\sigma\phi\mathcal{O}}^{00+})^* F_{\Delta,\mp}^{\epsilon\phi\phi\sigma}(x) \pm \sum_{\mathcal{O}\in\left[00;{r^\tau=0 \atop q=1}\right]} \lambda_{\phi\phi\mathcal{O}}^{+00}\lambda_{\epsilon\sigma\mathcal{O}} F_{\Delta,\mp}^{\sigma\epsilon\phi\phi}(x) \quad \text{(B.4)}$$

$$0 = \pm F_{0,\mp}^{\epsilon\epsilon\phi\phi}(x) + \sum_{\mathcal{O}\in[0+]} |\lambda_{\epsilon\phi\mathcal{O}}^{00+}|^2 F_{\Delta,\mp}^{\epsilon\phi\phi\epsilon}(x) \pm \sum_{\mathcal{O}\in\left[00;{r^\tau=0 \atop q=0}\right]} \lambda_{\phi\phi\mathcal{O}}^{+00}\lambda_{\epsilon\epsilon\mathcal{O}} F_{\Delta,\mp}^{\epsilon\epsilon\phi\phi}(x) \quad \text{(B.5)}$$

$$0 = F_{0,-}^{\sigma\sigma\sigma\sigma}(x) + \sum_{\mathcal{O}\in\left[00;{r^\tau=0 \atop q=0}\right]} \lambda_{\sigma\sigma\mathcal{O}}^2 F_{\Delta,-}^{\sigma\sigma\sigma\sigma}(x) \quad \text{(B.6)}$$

$$0 = F_{0,\mp}^{\sigma\sigma\epsilon\epsilon}(x) + \sum_{\mathcal{O}\in\left[00;{r^\tau=0 \atop q=0}\right]} \lambda_{\sigma\sigma\mathcal{O}}\lambda_{\epsilon\epsilon\mathcal{O}} F_{\Delta,\mp}^{\sigma\sigma\epsilon\epsilon}(x) \pm \sum_{\mathcal{O}\in\left[00;{r^\tau=0,1 \atop q=1}\right]} |\lambda_{\epsilon\sigma\mathcal{O}}|^2 F_{\Delta,\mp}^{\epsilon\sigma\sigma\epsilon}(x) \quad \text{(B.7)}$$

$$0 = \sum_{\mathcal{O}\in\left[00;{r^\tau=0,1 \atop q=1}\right]} (-1)^{r^\tau} |\lambda_{\epsilon\sigma\mathcal{O}}|^2 F_{\Delta,-}^{\sigma\epsilon\sigma\epsilon}(x) \quad \text{(B.8)}$$

$$0 = F_{0,-}^{\epsilon\epsilon\epsilon\epsilon}(x) + \sum_{\mathcal{O}\in\left[00;{r^\tau=0 \atop q=0}\right]} \lambda_{\epsilon\epsilon\mathcal{O}}^2 F_{\Delta,-}^{\epsilon\epsilon\epsilon\epsilon}(x). \quad \text{(B.9)}$$

The crossing equations lead to the following crossing vectors listed below. Note that for crossing vectors whose elements are $n \times n$ matrices with $n > 1$, we use "0" to represent the

zero matrix. Note that we also use $\phi$ as shorthand to represent $\phi_0^{0+}$. Also, we define $r_T = \frac{\Delta_\epsilon}{\Delta_\sigma}$.

$$
\mathbf{V}_T(x) =
\begin{pmatrix}
\begin{pmatrix} F_{3,-}^{\phi\phi\phi\phi} & 0 \\ 0 & 0 \end{pmatrix} \\[4pt]
\begin{pmatrix} -F_{3,+}^{\phi\phi\phi\phi} & 0 \\ 0 & 0 \end{pmatrix} \\[4pt]
\begin{pmatrix} F_{3,-}^{\phi\phi\phi\phi} & 0 \\ 0 & 0 \end{pmatrix} \\[4pt]
\begin{pmatrix} -F_{3,+}^{\phi\phi\phi\phi} & 0 \\ 0 & 0 \end{pmatrix} \\[4pt]
\begin{pmatrix} 0 & \frac{1}{2}F_{3,-}^{\sigma\sigma\phi\phi} \\ \frac{1}{2}F_{3,-}^{\sigma\sigma\phi\phi} & 0 \end{pmatrix} \\[4pt]
\begin{pmatrix} 0 & -\frac{1}{2}F_{3,+}^{\sigma\sigma\phi\phi} \\ -\frac{1}{2}F_{3,+}^{\sigma\sigma\phi\phi} & 0 \end{pmatrix} \\[2pt]
0 \\
0 \\
\begin{pmatrix} 0 & \frac{r_T}{2}F_{3,-}^{\epsilon\epsilon\phi\phi} \\ \frac{r_T}{2}F_{3,-}^{\epsilon\epsilon\phi\phi} & 0 \end{pmatrix} \\[4pt]
\begin{pmatrix} 0 & -\frac{r_T}{2}F_{3,+}^{\epsilon\epsilon\phi\phi} \\ -\frac{r_T}{2}F_{3,+}^{\epsilon\epsilon\phi\phi} & 0 \end{pmatrix} \\[4pt]
\begin{pmatrix} 0 & 0 \\ 0 & F_{3,-}^{\sigma\sigma\sigma\sigma} \end{pmatrix} \\[4pt]
\begin{pmatrix} 0 & 0 \\ 0 & r_T F_{3,-}^{\sigma\sigma\epsilon} \end{pmatrix} \\[4pt]
\begin{pmatrix} 0 & 0 \\ 0 & r_T F_{3,+}^{\sigma\sigma\epsilon} \end{pmatrix} \\[2pt]
0 \\
\begin{pmatrix} 0 & 0 \\ 0 & r_T^2 F_{3,-}^{\epsilon\epsilon\epsilon\epsilon} \end{pmatrix} \\[2pt]
0
\end{pmatrix}
\qquad
\mathbf{V}_\Delta^{0,0}(x) =
\begin{pmatrix}
\begin{pmatrix} F_{\Delta,-}^{\phi\phi\phi\phi} & 0 & 0 \\ 0 & 0 & 0 \\ 0 & 0 & 0 \end{pmatrix} \\[4pt]
\begin{pmatrix} -F_{\Delta,+}^{\phi\phi\phi\phi} & 0 & 0 \\ 0 & 0 & 0 \\ 0 & 0 & 0 \end{pmatrix} \\[4pt]
\begin{pmatrix} F_{\Delta,-}^{\phi\phi\phi\phi} & 0 & 0 \\ 0 & 0 & 0 \\ 0 & 0 & 0 \end{pmatrix} \\[4pt]
\begin{pmatrix} -F_{\Delta,+}^{\phi\phi\phi\phi} & 0 & 0 \\ 0 & 0 & 0 \\ 0 & 0 & 0 \end{pmatrix} \\[4pt]
\begin{pmatrix} 0 & \frac{1}{2}F_{\Delta,-}^{\sigma\sigma\phi\phi} & 0 \\ \frac{1}{2}F_{\Delta,-}^{\sigma\sigma\phi\phi} & 0 & 0 \\ 0 & 0 & 0 \end{pmatrix} \\[4pt]
\begin{pmatrix} 0 & -\frac{1}{2}F_{\Delta,+}^{\sigma\sigma\phi\phi} & 0 \\ -\frac{1}{2}F_{\Delta,+}^{\sigma\sigma\phi\phi} & 0 & 0 \\ 0 & 0 & 0 \end{pmatrix} \\[2pt]
0 \\
0 \\
\begin{pmatrix} 0 & 0 & \frac{1}{2}F_{\Delta,-}^{\epsilon\epsilon\phi\phi} \\ 0 & 0 & 0 \\ \frac{1}{2}F_{\Delta,-}^{\epsilon\epsilon\phi\phi} & 0 & 0 \end{pmatrix} \\[4pt]
\begin{pmatrix} 0 & 0 & -\frac{1}{2}F_{\Delta,+}^{\epsilon\epsilon\phi\phi} \\ 0 & 0 & 0 \\ -\frac{1}{2}F_{\Delta,+}^{\epsilon\epsilon\phi\phi} & 0 & 0 \end{pmatrix} \\[4pt]
\begin{pmatrix} 0 & 0 & 0 \\ 0 & F_{\Delta,-}^{\sigma\sigma\sigma\sigma} & 0 \\ 0 & 0 & 0 \end{pmatrix} \\[4pt]
\begin{pmatrix} 0 & 0 & 0 \\ 0 & 0 & \frac{1}{2}F_{\Delta,-}^{\sigma\sigma\epsilon} \\ 0 & \frac{1}{2}F_{\Delta,-}^{\sigma\sigma\epsilon} & 0 \end{pmatrix} \\[4pt]
\begin{pmatrix} 0 & 0 & 0 \\ 0 & 0 & \frac{1}{2}F_{\Delta,+}^{\sigma\sigma\epsilon} \\ 0 & \frac{1}{2}F_{\Delta,+}^{\sigma\sigma\epsilon} & 0 \end{pmatrix} \\[2pt]
0 \\
\begin{pmatrix} 0 & 0 & 0 \\ 0 & 0 & 0 \\ 0 & 0 & F_{\Delta,-}^{\epsilon\epsilon\epsilon\epsilon} \end{pmatrix} \\[2pt]
0
\end{pmatrix}
$$

$$
\mathbf{V}_\Delta^{1,0}(x) = \begin{pmatrix} \begin{pmatrix} F_{\Delta,-}^{\phi\phi\phi\phi} & 0 \\ 0 & 0 \end{pmatrix} \\ \begin{pmatrix} -F_{\Delta,+}^{\phi\phi\phi\phi} & 0 \\ 0 & 0 \end{pmatrix} \\ \begin{pmatrix} -F_{\Delta,-}^{\phi\phi\phi\phi} & 0 \\ 0 & 0 \end{pmatrix} \\ \begin{pmatrix} F_{\Delta,+}^{\phi\phi\phi\phi} & 0 \\ 0 & 0 \end{pmatrix} \\ 0 \\ 0 \\ \begin{pmatrix} 0 & \frac{1}{2}F_{\Delta,-}^{\sigma\epsilon\phi\phi} \\ \frac{1}{2}F_{\Delta,-}^{\sigma\epsilon\phi\phi} & 0 \end{pmatrix} \\ \begin{pmatrix} 0 & -\frac{1}{2}F_{\Delta,+}^{\sigma\epsilon\phi\phi} \\ -\frac{1}{2}F_{\Delta,+}^{\sigma\epsilon\phi\phi} & 0 \end{pmatrix} \\ 0 \\ 0 \\ 0 \\ \begin{pmatrix} 0 & 0 \\ 0 & F_{\Delta,-}^{\epsilon\sigma\sigma\epsilon} \end{pmatrix} \\ \begin{pmatrix} 0 & 0 \\ 0 & -F_{\Delta,+}^{\epsilon\sigma\sigma\epsilon} \end{pmatrix} \\ \begin{pmatrix} 0 & 0 \\ 0 & F_{\Delta,-}^{\epsilon\sigma\epsilon\sigma} \end{pmatrix} \\ 0 \\ 0 \end{pmatrix}
\qquad
\mathbf{V}_\Delta^{1,1}(x) = \begin{pmatrix} 0 \\ 0 \\ 0 \\ 0 \\ 0 \\ 0 \\ 0 \\ 0 \\ 0 \\ 0 \\ 0 \\ 0 \\ F_{\Delta,-}^{\epsilon\sigma\sigma\epsilon} \\ -F_{\Delta,+}^{\epsilon\sigma\sigma\epsilon} \\ -F_{\Delta,-}^{\epsilon\sigma\epsilon\sigma} \\ 0 \\ 0 \end{pmatrix}
$$

$$\mathbf{V}_{\varepsilon',\Delta_{\varepsilon'}}(x) = \begin{pmatrix} \begin{pmatrix} F^{\phi\phi\phi\phi}_{\Delta_{\varepsilon'},-} & 0 & 0 & 0 \\ 0 & 0 & 0 & 0 \\ 0 & 0 & 0 & 0 \\ 0 & 0 & 0 & F^{\phi\phi\phi\phi}_{\Delta_{\varepsilon'}+2,-} \end{pmatrix} \\ \begin{pmatrix} -F^{\phi\phi\phi\phi}_{\Delta_{\varepsilon'},+} & 0 & 0 & 0 \\ 0 & 0 & 0 & 0 \\ 0 & 0 & 0 & 0 \\ 0 & 0 & 0 & -F^{\phi\phi\phi\phi}_{\Delta_{\varepsilon'}+2,+} \end{pmatrix} \\ \begin{pmatrix} F^{\phi\phi\phi\phi}_{\Delta_{\varepsilon'},-} & 0 & 0 & 0 \\ 0 & 0 & 0 & 0 \\ 0 & 0 & 0 & 0 \\ 0 & 0 & 0 & F^{\phi\phi\phi\phi}_{\Delta_{\varepsilon'}+2,-} \end{pmatrix} \\ \begin{pmatrix} -F^{\phi\phi\phi\phi}_{\Delta_{\varepsilon'},+} & 0 & 0 & 0 \\ 0 & 0 & 0 & 0 \\ 0 & 0 & 0 & 0 \\ 0 & 0 & 0 & -F^{\phi\phi\phi\phi}_{\Delta_{\varepsilon'}+2,+} \end{pmatrix} \\ \begin{pmatrix} 0 & \frac{1}{2}F^{\sigma\sigma\phi\phi}_{\Delta_{\varepsilon'},-} & 0 & 0 \\ \frac{1}{2}F^{\sigma\sigma\phi\phi}_{\Delta_{\varepsilon'},-} & 0 & 0 & \frac{1}{2}\alpha^{(2)}_{\mathcal{O}\sigma\sigma}F^{\sigma\sigma\phi\phi}_{\Delta_{\varepsilon'}+2,-} \\ 0 & 0 & 0 & 0 \\ 0 & \frac{1}{2}\alpha^{(2)}_{\mathcal{O}\sigma\sigma}F^{\sigma\sigma\phi\phi}_{\Delta_{\varepsilon'}+2,-} & 0 & 0 \end{pmatrix} \\ \begin{pmatrix} 0 & -\frac{1}{2}F^{\sigma\sigma\phi\phi}_{\Delta_{\varepsilon'},+} & 0 & 0 \\ -\frac{1}{2}F^{\sigma\sigma\phi\phi}_{\Delta_{\varepsilon'},+} & 0 & 0 & -\frac{1}{2}\alpha^{(2)}_{\mathcal{O}\sigma\sigma}F^{\sigma\sigma\phi\phi}_{\Delta_{\varepsilon'}+2,+} \\ 0 & 0 & 0 & 0 \\ 0 & -\frac{1}{2}\alpha^{(2)}_{\mathcal{O}\sigma\sigma}F^{\sigma\sigma\phi\phi}_{\Delta_{\varepsilon'}+2,+} & 0 & 0 \end{pmatrix} \\ 0 \\ \begin{pmatrix} 0 & 0 & \frac{1}{2}F^{\varepsilon\varepsilon\phi\phi}_{\Delta_{\varepsilon'},-} & 0 \\ 0 & 0 & 0 & 0 \\ \frac{1}{2}F^{\varepsilon\varepsilon\phi\phi}_{\Delta_{\varepsilon'},-} & 0 & 0 & \frac{1}{2}\alpha^{(2)}_{\mathcal{O}\varepsilon\varepsilon}F^{\varepsilon\varepsilon\phi\phi}_{\Delta_{\varepsilon'}+2,-} \\ 0 & 0 & \frac{1}{2}\alpha^{(2)}_{\mathcal{O}\varepsilon\varepsilon}F^{\varepsilon\varepsilon\phi\phi}_{\Delta_{\varepsilon'}+2,-} & 0 \end{pmatrix} \\ \begin{pmatrix} 0 & 0 & -\frac{1}{2}F^{\varepsilon\varepsilon\phi\phi}_{\Delta_{\varepsilon'},+} & 0 \\ 0 & 0 & 0 & 0 \\ -\frac{1}{2}F^{\varepsilon\varepsilon\phi\phi}_{\Delta_{\varepsilon'},+} & 0 & 0 & -\frac{1}{2}\alpha^{(2)}_{\mathcal{O}\varepsilon\varepsilon}F^{\varepsilon\varepsilon\phi\phi}_{\Delta_{\varepsilon'}+2,+} \\ 0 & 0 & -\frac{1}{2}\alpha^{(2)}_{\mathcal{O}\varepsilon\varepsilon}F^{\varepsilon\varepsilon\phi\phi}_{\Delta_{\varepsilon'}+2,+} & 0 \end{pmatrix} \\ \begin{pmatrix} 0 & 0 & 0 & 0 \\ 0 & F^{\sigma\sigma\sigma\sigma}_{\Delta_{\varepsilon'},-} + (\alpha^{(2)}_{\mathcal{O}\sigma\sigma})^2 F^{\sigma\sigma\sigma\sigma}_{\Delta_{\varepsilon'}+2,-} & 0 & 0 \\ 0 & 0 & 0 & 0 \\ 0 & 0 & 0 & 0 \end{pmatrix} \\ \begin{pmatrix} 0 & 0 & 0 & 0 \\ 0 & 0 & \frac{1}{2}F^{\sigma\sigma\varepsilon\varepsilon}_{\Delta_{\varepsilon'},-} + \frac{1}{2}\alpha^{(2)}_{\mathcal{O}\varepsilon\varepsilon}\alpha^{(2)}_{\mathcal{O}\sigma\sigma}F^{\sigma\sigma\varepsilon\varepsilon}_{\Delta_{\varepsilon'}+2,-} & 0 \\ 0 & \frac{1}{2}F^{\sigma\sigma\varepsilon\varepsilon}_{\Delta_{\varepsilon'},-} + \frac{1}{2}\alpha^{(2)}_{\mathcal{O}\varepsilon\varepsilon}\alpha^{(2)}_{\mathcal{O}\sigma\sigma}F^{\sigma\sigma\varepsilon\varepsilon}_{\Delta_{\varepsilon'}+2,-} & 0 & 0 \\ 0 & 0 & 0 & 0 \end{pmatrix} \\ \begin{pmatrix} 0 & 0 & 0 & 0 \\ 0 & 0 & \frac{1}{2}F^{\sigma\sigma\varepsilon\varepsilon}_{\Delta_{\varepsilon'},+} + \frac{1}{2}\alpha^{(2)}_{\mathcal{O}\varepsilon\varepsilon}\alpha^{(2)}_{\mathcal{O}\sigma\sigma}F^{\sigma\sigma\varepsilon\varepsilon}_{\Delta_{\varepsilon'}+2,+} & 0 \\ 0 & \frac{1}{2}F^{\sigma\sigma\varepsilon\varepsilon}_{\Delta_{\varepsilon'},+} + \frac{1}{2}\alpha^{(2)}_{\mathcal{O}\varepsilon\varepsilon}\alpha^{(2)}_{\mathcal{O}\sigma\sigma}F^{\sigma\sigma\varepsilon\varepsilon}_{\Delta_{\varepsilon'}+2,+} & 0 & 0 \\ 0 & 0 & 0 & 0 \end{pmatrix} \\ 0 \\ \begin{pmatrix} 0 & 0 & 0 & 0 \\ 0 & 0 & 0 & 0 \\ 0 & 0 & F^{\varepsilon\varepsilon\varepsilon\varepsilon}_{\Delta_{\varepsilon'},-} + (\alpha^{(2)}_{\mathcal{O}\varepsilon\varepsilon})^2 F^{\varepsilon\varepsilon\varepsilon\varepsilon}_{\Delta_{\varepsilon'}+2,-} & 0 \\ 0 & 0 & 0 & 0 \end{pmatrix} \\ 0 \end{pmatrix}$$

$$\mathbf{V}_\Delta^{++}(x) = \begin{pmatrix} F_{\Delta,-}^{\phi\phi\phi\phi} \\ F_{\Delta,+}^{\phi\phi\phi\phi} \\ 0 \\ 0 \\ 0 \\ 0 \\ 0 \\ 0 \\ 0 \\ 0 \\ 0 \\ 0 \\ 0 \\ 0 \\ 0 \\ 0 \end{pmatrix} \qquad \mathbf{V}_\Delta^{+-}(x) = \begin{pmatrix} 0 \\ 0 \\ F_{\Delta,-}^{\phi\phi\phi\phi} \\ F_{\Delta,+}^{\phi\phi\phi\phi} \\ 0 \\ 0 \\ 0 \\ 0 \\ 0 \\ 0 \\ 0 \\ 0 \\ 0 \\ 0 \\ 0 \\ 0 \end{pmatrix} \qquad \mathbf{V}_\Delta^{0+}(x) = \begin{pmatrix} 0 \\ 0 \\ 0 \\ 0 \\ \begin{pmatrix} F_{\Delta,-}^{\sigma\phi\phi\sigma} & 0 \\ 0 & 0 \end{pmatrix} \\ \begin{pmatrix} F_{\Delta,+}^{\sigma\phi\phi\sigma} & 0 \\ 0 & 0 \end{pmatrix} \\ \begin{pmatrix} 0 & \frac{1}{2}F_{\Delta,-}^{\epsilon\phi\phi\sigma} \\ \frac{1}{2}F_{\Delta,-}^{\epsilon\phi\phi\sigma} & 0 \end{pmatrix} \\ \begin{pmatrix} 0 & \frac{1}{2}F_{\Delta,+}^{\epsilon\phi\phi\sigma} \\ \frac{1}{2}F_{\Delta,+}^{\epsilon\phi\phi\sigma} & 0 \end{pmatrix} \\ \begin{pmatrix} 0 & 0 \\ 0 & F_{\Delta,-}^{\epsilon\phi\phi\epsilon} \end{pmatrix} \\ \begin{pmatrix} 0 & 0 \\ 0 & F_{\Delta,+}^{\epsilon\phi\phi\epsilon} \end{pmatrix} \\ 0 \\ 0 \\ 0 \\ 0 \\ 0 \\ 0 \end{pmatrix}$$

$$
\mathbf{V}_{\phi_0^{0+}\sigma\epsilon}(x) =
\begin{pmatrix}
\begin{pmatrix}
F^{\phi\phi\phi\phi}_{\Delta_\sigma,-} & 0 & 0 & 0 & 0 \\
0 & F^{\phi\phi\phi\phi}_{\Delta_\varepsilon,-} & 0 & 0 & 0 \\
0 & 0 & 0 & 0 & 0 \\
0 & 0 & 0 & F^{\phi\phi\phi\phi}_{\Delta_\sigma+2,-} & 0 \\
0 & 0 & 0 & 0 & F^{\phi\phi\phi\phi}_{\Delta_\varepsilon+2,-}
\end{pmatrix} \\
\begin{pmatrix}
-F^{\phi\phi\phi\phi}_{\Delta_\sigma,+} & 0 & 0 & 0 & 0 \\
0 & -F^{\phi\phi\phi\phi}_{\Delta_\varepsilon,+} & 0 & 0 & 0 \\
0 & 0 & 0 & 0 & 0 \\
0 & 0 & 0 & -F^{\phi\phi\phi\phi}_{\Delta_\sigma+2,+} & 0 \\
0 & 0 & 0 & 0 & -F^{\phi\phi\phi\phi}_{\Delta_\varepsilon+2,+}
\end{pmatrix} \\
\begin{pmatrix}
-F^{\phi\phi\phi\phi}_{\Delta_\sigma,-} & 0 & 0 & 0 & 0 \\
0 & F^{\phi\phi\phi\phi}_{\Delta_\varepsilon,-} & 0 & 0 & 0 \\
0 & 0 & 0 & 0 & 0 \\
0 & 0 & 0 & -F^{\phi\phi\phi\phi}_{\Delta_\sigma+2,-} & 0 \\
0 & 0 & 0 & 0 & F^{\phi\phi\phi\phi}_{\Delta_\varepsilon+2,-}
\end{pmatrix} \\
\begin{pmatrix}
F^{\phi\phi\phi\phi}_{\Delta_\sigma,+} & 0 & 0 & 0 & 0 \\
0 & -F^{\phi\phi\phi\phi}_{\Delta_\varepsilon,+} & 0 & 0 & 0 \\
0 & 0 & 0 & 0 & 0 \\
0 & 0 & 0 & F^{\phi\phi\phi\phi}_{\Delta_\sigma+2,+} & 0 \\
0 & 0 & 0 & 0 & -F^{\phi\phi\phi\phi}_{\Delta_\varepsilon+2,+}
\end{pmatrix} \\
\begin{pmatrix}
F^{\sigma\phi\phi\sigma}_{\Delta_0^{0+},-} & 0 & 0 & 0 & 0 \\
0 & 0 & \frac{1}{2}F^{\sigma\sigma\phi\phi}_{\Delta_\varepsilon,-} & 0 & 0 \\
0 & \frac{1}{2}F^{\sigma\sigma\phi\phi}_{\Delta_\varepsilon,-} & 0 & 0 & \frac{1}{2}\alpha^{(2)}_{\varepsilon\sigma\sigma}F^{\sigma\sigma\phi\phi}_{\Delta_\varepsilon+2,-} \\
0 & 0 & 0 & 0 & 0 \\
0 & 0 & \frac{1}{2}\alpha^{(2)}_{\varepsilon\sigma\sigma}F^{\sigma\sigma\phi\phi}_{\Delta_\varepsilon+2,-} & 0 & 0
\end{pmatrix} \\
\begin{pmatrix}
F^{\sigma\phi\phi\sigma}_{\Delta_0^{0+},+} & 0 & 0 & 0 & 0 \\
0 & 0 & -\frac{1}{2}F^{\sigma\sigma\phi\phi}_{\Delta_\varepsilon,+} & 0 & 0 \\
0 & -\frac{1}{2}F^{\sigma\sigma\phi\phi}_{\Delta_\varepsilon,+} & 0 & 0 & -\frac{1}{2}\alpha^{(2)}_{\varepsilon\sigma\sigma}F^{\sigma\sigma\phi\phi}_{\Delta_\varepsilon+2,+} \\
0 & 0 & 0 & 0 & 0 \\
0 & 0 & -\frac{1}{2}\alpha^{(2)}_{\varepsilon\sigma\sigma}F^{\sigma\sigma\phi\phi}_{\Delta_\varepsilon+2,+} & 0 & 0
\end{pmatrix} \\
\begin{pmatrix}
0 & \frac{1}{2}F^{\varepsilon\phi\phi\sigma}_{\Delta_0^{0+},-} & \frac{1}{2}F^{\sigma\varepsilon\phi\phi}_{\Delta_\sigma,-} & 0 & 0 \\
\frac{1}{2}F^{\varepsilon\phi\phi\sigma}_{\Delta_0^{0+},-} & 0 & 0 & 0 & 0 \\
\frac{1}{2}F^{\sigma\varepsilon\phi\phi}_{\Delta_\sigma,-} & 0 & 0 & \frac{1}{2}\alpha^{(2)}_{\sigma\varepsilon\sigma}F^{\sigma\varepsilon\phi\phi}_{\Delta_\sigma+2,-} & 0 \\
0 & 0 & \frac{1}{2}\alpha^{(2)}_{\sigma\varepsilon\sigma}F^{\sigma\varepsilon\phi\phi}_{\Delta_\sigma+2,-} & 0 & 0 \\
0 & 0 & 0 & 0 & 0
\end{pmatrix} \\
\begin{pmatrix}
0 & \frac{1}{2}F^{\varepsilon\phi\phi\sigma}_{\Delta_0^{0+},+} & -\frac{1}{2}F^{\sigma\varepsilon\phi\phi}_{\Delta_\sigma,+} & 0 & 0 \\
\frac{1}{2}F^{\varepsilon\phi\phi\sigma}_{\Delta_0^{0+},+} & 0 & 0 & 0 & 0 \\
-\frac{1}{2}F^{\sigma\varepsilon\phi\phi}_{\Delta_\sigma,+} & 0 & 0 & -\frac{1}{2}\alpha^{(2)}_{\sigma\varepsilon\sigma}F^{\sigma\varepsilon\phi\phi}_{\Delta_\sigma+2,+} & 0 \\
0 & 0 & -\frac{1}{2}\alpha^{(2)}_{\sigma\varepsilon\sigma}F^{\sigma\varepsilon\phi\phi}_{\Delta_\sigma+2,+} & 0 & 0 \\
0 & 0 & 0 & 0 & 0
\end{pmatrix} \\
\begin{pmatrix}
0 & 0 & 0 & 0 & 0 \\
0 & F^{\varepsilon,\phi,\phi,\varepsilon}_{\Delta_0^{0+},-} & \frac{1}{2}r_{\sigma\varepsilon}F^{\varepsilon\varepsilon\phi\phi}_{\Delta_\varepsilon,-} & 0 & 0 \\
0 & \frac{1}{2}r_{\sigma\varepsilon}F^{\varepsilon\varepsilon\phi\phi}_{\Delta_\varepsilon,-} & 0 & 0 & \frac{1}{2}r_{\sigma\varepsilon}\alpha^{(2)}_{\varepsilon\varepsilon\varepsilon}F^{\varepsilon\varepsilon\phi\phi}_{\Delta_\varepsilon+2,-} \\
0 & 0 & 0 & 0 & 0 \\
0 & 0 & \frac{1}{2}r_{\sigma\varepsilon}\alpha^{(2)}_{\varepsilon\varepsilon\varepsilon}F^{\varepsilon\varepsilon\phi\phi}_{\Delta_\varepsilon+2,-} & 0 & 0
\end{pmatrix} \\
\begin{pmatrix}
0 & 0 & 0 & 0 & 0 \\
0 & F^{\varepsilon,\phi,\phi,\varepsilon}_{\Delta_0^{0+},+} & -\frac{1}{2}r_{\sigma\varepsilon}F^{\varepsilon\varepsilon\phi\phi}_{\Delta_\varepsilon,+} & 0 & 0 \\
0 & -\frac{1}{2}r_{\sigma\varepsilon}F^{\varepsilon\varepsilon\phi\phi}_{\Delta_\varepsilon,+} & 0 & 0 & -\frac{1}{2}r_{\sigma\varepsilon}\alpha^{(2)}_{\varepsilon\varepsilon\varepsilon}F^{\varepsilon\varepsilon\phi\phi}_{\Delta_\varepsilon+2,+} \\
0 & 0 & 0 & 0 & 0 \\
0 & 0 & -\frac{1}{2}r_{\sigma\varepsilon}\alpha^{(2)}_{\varepsilon\varepsilon\varepsilon}F^{\varepsilon\varepsilon\phi\phi}_{\Delta_\varepsilon+2,+} & 0 & 0
\end{pmatrix} \\
\begin{pmatrix}
0 & 0 & 0 & 0 & 0 \\
0 & 0 & 0 & 0 & 0 \\
0 & 0 & F^{\sigma\sigma\sigma\sigma}_{\Delta_\varepsilon,-}+(\alpha^{(2)}_{\varepsilon\sigma\sigma})^2 F^{\sigma\sigma\sigma\sigma}_{\Delta_\varepsilon+2,-} & 0 & 0 \\
0 & 0 & 0 & 0 & 0 \\
0 & 0 & 0 & 0 & 0
\end{pmatrix}
\end{pmatrix}
$$

$$\left( \begin{array}{c} \begin{pmatrix} 0 & 0 & 0 & 0 & 0 \\ 0 & 0 & 0 & 0 & 0 \\ 0 & 0 & F_{\Delta_\sigma,-}^{\varepsilon\sigma\sigma\varepsilon} + r_{\sigma\varepsilon}\left(F_{\Delta_\varepsilon,-}^{\sigma\sigma\varepsilon\varepsilon} + \alpha_{\varepsilon\varepsilon\varepsilon}^{(2)}\alpha_{\varepsilon\sigma\sigma}^{(2)}F_{\Delta_\varepsilon+2,-}^{\sigma\sigma\varepsilon\varepsilon}\right) + (\alpha_{\sigma\varepsilon\sigma}^{(2)})^2 F_{\Delta_\sigma+2,-}^{\varepsilon\sigma\sigma\varepsilon} & 0 & 0 \\ 0 & 0 & 0 & 0 & 0 \\ 0 & 0 & 0 & 0 & 0 \end{pmatrix} \\[2em] \begin{pmatrix} 0 & 0 & 0 & 0 & 0 \\ 0 & 0 & 0 & 0 & 0 \\ 0 & 0 & -F_{\Delta_\sigma,+}^{\varepsilon\sigma\sigma\varepsilon} + r_{\sigma\varepsilon}\left(F_{\Delta_\varepsilon,+}^{\sigma\sigma\varepsilon\varepsilon} + \alpha_{\varepsilon\varepsilon\varepsilon}^{(2)}\alpha_{\varepsilon\sigma\sigma}^{(2)}F_{\Delta_\varepsilon+2,+}^{\sigma\sigma\varepsilon\varepsilon}\right) - (\alpha_{\sigma\varepsilon\sigma}^{(2)})^2 F_{\Delta_\sigma+2,+}^{\varepsilon\sigma\sigma\varepsilon} & 0 & 0 \\ 0 & 0 & 0 & 0 & 0 \\ 0 & 0 & 0 & 0 & 0 \end{pmatrix} \\[2em] \oplus \quad \begin{array}{c} \begin{pmatrix} 0 & 0 & 0 & 0 & 0 \\ 0 & 0 & 0 & 0 & 0 \\ 0 & 0 & F_{\Delta_\sigma,-}^{\sigma\varepsilon\sigma\varepsilon} + (\alpha_{\sigma\varepsilon\sigma}^{(2)})^2 F_{\Delta_\sigma+2,-}^{\sigma\varepsilon\sigma\varepsilon} & 0 & 0 \\ 0 & 0 & 0 & 0 & 0 \\ 0 & 0 & 0 & 0 & 0 \end{pmatrix} \\[2em] \begin{pmatrix} 0 & 0 & 0 & 0 & 0 \\ 0 & 0 & 0 & 0 & 0 \\ 0 & 0 & r_{\sigma\varepsilon}^2\left(F_{\Delta_\varepsilon,-}^{\varepsilon\varepsilon\varepsilon\varepsilon} + (\alpha_{\varepsilon\varepsilon\varepsilon}^{(2)})^2 F_{\Delta_\varepsilon+2,-}^{\varepsilon\varepsilon\varepsilon\varepsilon}\right) & 0 & 0 \\ 0 & 0 & 0 & 0 & 0 \\ 0 & 0 & 0 & 0 & 0 \end{pmatrix} \\[2em] \begin{pmatrix} 0 & 0 & 0 & 0 & 0 \\ 0 & 0 & 0 & 0 & 0 \\ 0 & 0 & 1 & 0 & 0 \\ 0 & 0 & 0 & 0 & 0 \\ 0 & 0 & 0 & 0 & 0 \end{pmatrix} \end{array} \end{array} \right)$$

## C   Cutting-curve algorithm

Here we describe an algorithm to solve an optimization problem frequently encountered in this work. Throughout this section, we will assume all scaling dimensions of external operators are held fixed. Let $\mathbf{V}(x;\gamma)$ be a crossing vector that depends on some *internal* parameter $\gamma$ that may take values within a finite interval $\gamma \in [\gamma_l, \gamma_h]$ (i.e. it does not represent the scaling dimension of an external operator—if it did, we expect the algorithm described in this section to be substantially less efficient than alternative approaches such as the navigator method.). We will either take $\gamma$ to be the scaling dimension of an internal operator or a ratio of two OPE coefficients. Next, let $\mathbf{V}_{\Delta,i}(x)$ represent the remaining set of crossing vectors we will impose positivity on within some ranges $S_i$ of the internal scaling dimension $\Delta \in S_i$, where $S_i$ may be disconnected. Finally let $\mathcal{N}(x)$ be a normalization crossing vector.

We wish to numerically solve problems of the form

$$\text{(Maximize/Minimize) } \gamma \text{ subject to}$$
$$\alpha[\mathcal{N}(x)] = 1$$
$$\alpha[\mathbf{V}(x;\gamma)] \succeq 0 \tag{C.1}$$
$$\alpha[\mathbf{V}_{\Delta,i}(x)] \succeq 0 \qquad \Delta \in S_i$$

where $\alpha$ is defined as in the main text in (3.1). We will assume, for this implementation, that the set of allowed values of $\gamma$ is connected—we have found no evidence to the contrary in the examples studied in this work. For the purpose of defining our algorithm, this means that when the scaling dimensions of all external operators are fixed, the allowed ranges of $\gamma$ form an interval. Note that we may not simply optimize $\gamma$ using SDPB directly, since $\gamma$ is not explicitly bounded by the value of a linear functional acting on some crossing vector, as is the case when, say, optimizing the magnitude of OPE coefficients.

We require an algorithm that can efficiently do the following:

1. If there is actually no allowed $\gamma \in [\gamma_l, \gamma_h]$, rigorously rule out the entire range.

2. If there is an allowed $\gamma \in [\gamma_l, \gamma_h]$, find a single allowed point.

When there is at least one allowed point, the hardest part of the calculation is finding such an allowed point. Once an allowed point $\gamma_*$ is found, since we assume the entire range of allowed values to be an interval we know $\gamma_l \leq \gamma_{\min} \leq \gamma_* \leq \gamma_{\max} \leq \gamma_h$, where $\gamma_{\min}, \gamma_{\max}$ represent the optimal lower/upper bounds. From there we may efficiently approximate $\gamma_{\min}, \gamma_{\max}$, to within some desired tolerance, by performing a standard binary search that tests for feasibility of different choices of $\gamma$. At each step, if such an $\alpha$ is found, the semidefinite program is feasible (i.e. all of the constraints can be satisfied), which proves $\gamma$ in the above is *disallowed* via the standard numerical bootstrap logic. Depending on whether we seek to approximate the lower or upper bounds, we then raise or lower $\gamma$ and repeat.

To find the allowed point, we use the following strategy, which is essentially equivalent to the $m = 2$ case of the "cutting-surface" algorithm described in [79], although there is a very minor technical difference in our problem which is that $\alpha[\mathbf{V}(x; \gamma)]$ will not be a quadratic form in $\gamma$[24]. Define $\mathcal{A}_0 = [\gamma_l, \gamma_h]$, the set of initially allowed $\gamma$, and let $\gamma_0 = (\gamma_h - \gamma_l)/2$ be the initial value of $\gamma$ to check. The trick noticed in [79] is that any time some linear functional $\alpha$ rules out a point $\gamma$, it will also rule out points in some neighborhood of $\gamma$. We define this neighborhood by computing the nearest roots below and above $\gamma$ of $\det \alpha[\mathbf{V}(x; \gamma)]$, defining some interval $\mathcal{I}_0$ that is totally ruled out since the eigenvalues of $\alpha[\mathbf{V}(x; \gamma)]$ are necessarily positive at points continuously connected to $\gamma_0$ without encountering a root of $\det \alpha[\mathbf{V}(x; \gamma)]$, which is a smooth function of $\gamma$ (this is actually a weaker condition than could possibly be imposed, but we find it sufficient.). The set of allowed points then becomes $\mathcal{A}_1 = \mathcal{A}_0 \backslash \mathcal{I}_0$. To proceed in the algorithm, each $\mathcal{A}_i$ representing the allowed region after checking $\gamma_i$ will be expressible as a union of intervals $\mathcal{A}_i = \bigcup_n \mathcal{J}_n^i = A_0 \backslash \bigcup_{j=0}^i \mathcal{I}_j$. There are different ways to choose $\gamma_{i+1}$, but we found in practice that choosing $\gamma_{i+1}$ to be the midpoint of the widest $\mathcal{J}_n^i$ the most efficient for finding an allowed point. Once we find an allowed $\gamma$, we enter the binary search phase described previously.

Sometimes the aforementioned criteria for choosing $\gamma_{i+1}$ struggles to fully exclude all values of $\gamma$ when there is no allowed value. In practice, we consider a point ruled out if it does not find an allowed value within 75 iterations. This seems to happen due to the fact that we use "hot-starting," i.e. for each call to SDPB we recycle $\alpha$ from the previous run since it will necessarily be positive when evaluated on all the fixed constraints $\alpha[\mathbf{V}_{\Delta,i}(x)] \succeq 0$. As the algorithm proceeds, $\mathcal{I}_i$ typically shrinks in width[25], whereas $\mathcal{I}_0$ usually is sufficiently wide such that $\mathcal{A}_1$ is still connected. We experimented with letting SDPB run longer for each check of $\gamma$ by saving a checkpoint at termination only after every few runs, but it is unclear the

---

[24] When we are computing OPE ratios with this method we will let the OPE ratio equal e.g. $\cot \gamma, \tan \gamma$, which compactifies the possible values that must be scanned over. When $\gamma$ is an internal scaling dimension $\alpha[\mathbf{V}(x; \gamma)]$ will be a higher-degree polynomial.

[25] Strangely, this seems to be the opposite behavior to what is described in [79], where the functional near the boundary of the allowed region apparently rules out roughly half of the search space at each step.

ultimate effect this choice has on the overall efficiency. There are also other choices of $\gamma_{i+1}$ that seem to perform somewhat better at ruling out the full range of $\gamma$ when applicable, but we find choosing a method optimized for disallowed points less desirable since another way to judge when there should be no allowed $\gamma$ is the point, as a function of the external dimension, at which the upper and lower bounds on $\gamma$ agree, although this is arguably not fully rigorous. However, in our problem the external dimension is $\Delta_0^{0+}$, and all evidence suggests that there are two connected components of the allowed values of $\Delta_0^{0+}$ which we confirmed at small $\Lambda$, so we find the approach described here satisfactory.

## D   Perturbative calculation details

This appendix expands on the calculation of pinning field defect properties in the $4 - \epsilon$ expansion, described in section 4. Starting with Eq. (4.1) with $h(\tau) = h_0$, we use the standard $\beta$-function for $\lambda$ and the $\beta$-function for $h$ computed by Ref. [23, 29]:

$$\beta(\lambda_r) = -\epsilon \lambda_r + \frac{N+8}{3} \frac{\lambda_r^2}{(4\pi)^2} - \frac{3N+14}{3} \frac{\lambda^3}{(4\pi)^4} + O(\lambda^4),$$

$$\beta(h_r) = -\frac{\epsilon}{2} h_r + \frac{\lambda_r}{(4\pi)^2} \frac{h_r^3}{6} + \frac{\lambda_r^2}{(4\pi)^4} \left( \frac{N+2}{36} h_r - \frac{N+8}{36} h_r^3 - \frac{h_r^5}{12} \right) + O(\lambda^3). \quad \text{(D.1)}$$

where

$$\lambda_0 = \mu^\epsilon \left( \lambda_r + \frac{\delta\lambda}{\epsilon} + \frac{\delta_2\lambda}{\epsilon^2} + \dots \right),$$

$$\delta\lambda = \frac{N+8}{3} \frac{\lambda_r^2}{(4\pi)^2} - \frac{3N+14}{6} \frac{\lambda_r^3}{(4\pi)^4} + O(\lambda_r^4),$$

$$\delta_2\lambda = \frac{(N+8)^2}{9} \frac{\lambda_r^3}{(4\pi)^4} + O(\lambda_r^4), \quad \text{(D.2)}$$

and

$$h_0 = \mu^{\epsilon/2} \left( h_r + \frac{\delta h}{\epsilon} + \frac{\delta_2 h}{\epsilon^2} + \dots \right),$$

$$\delta h = \frac{\lambda_r}{(4\pi)^2} \frac{h_r^3}{12} + \frac{\lambda_r^2}{(4\pi)^4} \left( \frac{N+2}{72} h_r - \frac{N+8}{108} h_r^3 - \frac{h_r^5}{48} \right) + O(\lambda_r^3),$$

$$\delta_2 h = \frac{\lambda_r^2}{(4\pi)^4} \left( \frac{N+8}{108} h_r^3 + \frac{h_r^5}{96} \right) + O(\lambda_r^3). \quad \text{(D.3)}$$

We use dimensional regularization throughout this appendix. These expressions are given for the pinning field defect in the $O(N)$ model. We are interested in the Ising model, $N = 1$. We will not actually explicitly need the $\beta$-functions for $h$ and $\lambda$ to this high order, but we will use the result

$$h_{r,*}^2 = (N+8) + \epsilon \frac{4N^2 + 45N + 170}{2(N+8)}, \quad \text{(D.4)}$$

which follows from these $\beta$-functions.

## D.1 RG formalism for leading defect operators

As described in section 4.1, scaling dimensions of the leading defect endpoint and domain wall operators can be extracted from the defect partition functions $Z^{0+}$ and $Z^{+-}$. The computation beyond the leading order in $\epsilon$ requires a more involved analysis of the renormalization of $F[h(\tau)] = -\log Z[h(\tau)]$. The procedure will apply to both $Z^{0+}$ and $Z^{+-}$. $F$ receives additive renormalizations (corresponding to multiplicative renormalization of $\phi_0^{0+}$ or $\phi_0^{+-}$). We define

$$F = F_r(\mu, \lambda_r, h_r, L) + C(\mu, \lambda_r, h_r, \Lambda), \tag{D.5}$$

where $\Lambda$ is the UV cut-off, $\mu$ is the renormalization scale, $L$ is the defect length, and $F_r(\mu, \lambda_r, h_r, L)$ is finite. Applying the operator $\mu \frac{d}{d\mu}\Big|_{\lambda_0, h_0, \Lambda}$ to the left-hand-side of the above equation, we obtain the inhomogeneous Callan-Symanzik equation:

$$\left(\mu \frac{\partial}{\partial \mu} + \beta(\lambda_r)\frac{\partial}{\partial \lambda_r} + \beta(h_r)\frac{\partial}{\partial h_r}\right) F_r(\mu, \lambda_r, h_r) = B(\lambda_r, h_r), \tag{D.6}$$

where

$$B(\lambda_r, h_r) = -\mu \frac{dC}{d\mu}\Big|_{\lambda_0, h_0, \Lambda} = -\left(\mu \frac{\partial}{\partial \mu} + \beta(\lambda_r)\frac{\partial}{\partial \lambda_r} + \beta(h_r)\frac{\partial}{\partial h_r}\right) C(\mu, \lambda_r, h_r, \Lambda). \tag{D.7}$$

Note that $B(\lambda_r, h_r)$ must be finite, as it appears on the right-hand-side of (D.6). In dimensional regularization $C(\lambda_r, h_r)$ is independent of $\Lambda$ (and, therefore, of $\mu$), so

$$B(\lambda_r, h_r) = -\left(\beta(\lambda_r)\frac{\partial}{\partial \lambda_r} + \beta(h_r)\frac{\partial}{\partial h_r}\right) C(\lambda_r, h_r). \tag{D.8}$$

Then, solving the Callan-Symanzik equation

$$F_r(\mu, \lambda_{r,*}, h_{r,*}) = B(\lambda_{r,*}, h_{r,*}) \log \mu L + \text{const.}, \tag{D.9}$$

where $L$ is the length of the defect, so the scaling dimension of the defect operator is

$$\Delta = \frac{B(\lambda_{r,*}, h_{r,*})}{2}. \tag{D.10}$$

Armed with this general RG formalism, we now compute the defect free energy to order $\lambda$.

$$F = F_0 + F_1 + O(\lambda^2), \tag{D.11}$$

where

$$F_0 = -\frac{1}{2}\int d\tau d\tau' G(\tau - \tau')h(\tau)h(\tau') = \frac{1}{2}\int d\tau \phi_{cl}(\tau, 0)h(\tau), \tag{D.12}$$

$$F_1 = \frac{\lambda_0}{24}\int d^D x\, \phi_{cl}^4(x), \tag{D.13}$$

where

$$\phi_{cl}(x) = -\int d\tau'\, G(x - (\tau', 0))h(\tau'), \tag{D.14}$$

and, $G(x) = \frac{c(D)}{x^{D-2}}$, $c(D) = \frac{\Gamma(D/2-1)}{4\pi^{D/2}}$, is the free propagator.

### D.2 Leading defect endpoint dimension to $O(\epsilon)$

We begin with the defect endpoint operator:

$$h(\tau) = \begin{cases} h_0, & 0 < \tau < L, \\ 0, & \text{otherwise.} \end{cases} \tag{D.15}$$

Letting $\rho$ denote the distance to the defect axis,

$$\begin{aligned} \phi_{cl}(\tau, \rho) &= -h_0 c(D) \int_0^L d\tau' \frac{1}{((\tau - \tau')^2 + \rho^2)^{(D-2)/2}} \\ &= -h_0 c(D) \rho^{3-D} (f(\tau/\rho) + f((L-\tau)/\rho)), \end{aligned} \tag{D.16}$$

where

$$f(u) = u \, F(1/2, D/2 - 1, 3/2, -u^2) = \text{sgn}(u) \left( \frac{\sqrt{\pi} \Gamma((D-3)/2)}{2\Gamma(D/2 - 1)} + \frac{|u|^{3-D}}{3-D} \mathcal{F}(u^{-1}) \right), \tag{D.17}$$

and

$$\mathcal{F}(u) = F(D/2 - 1, (D-3)/2, (D-1)/2, -u^2), \tag{D.18}$$

where $F(\alpha, \beta, \gamma, x)$ is the hyper-geometric function. Thus, for $\rho \to 0$,

$$\begin{aligned} \phi_{cl}(\tau, \rho) &\approx -h_0 c(D) \left[ \frac{\sqrt{\pi} \Gamma((D-3)/2)}{2\Gamma(D/2 - 1)} (\text{sgn}(\tau) + \text{sgn}(L - \tau)) \rho^{3-D} \right. \\ &\left. + \frac{1}{3-D} \left( \text{sgn}(\tau)|\tau|^{3-D} + \text{sgn}(L - \tau)|L - \tau|^{3-D} \right) \right]. \end{aligned} \tag{D.19}$$

Using dimensional regularization,

$$\phi_{cl}(\tau, 0) = -\frac{h_0 c(D)}{3-D} \left( \text{sgn}(\tau)|\tau|^{3-D} + \text{sgn}(L - \tau)|L - \tau|^{3-D} \right). \tag{D.20}$$

Therefore, from Eq. (D.12),

$$F_0^{0+} = \frac{h_0^2 c(D)}{(D-3)(4-D)} L^{4-D} = \frac{h_0^2 \mu^{-\epsilon}}{4\pi^2 \epsilon} \left[ 1 + \epsilon \left( \frac{1}{2} \log \pi + \frac{\gamma}{2} + 1 + \log \mu L \right) + O(\epsilon^2) \right]. \tag{D.21}$$

This means that to leading order in $\lambda$ $(\epsilon)$,

$$C^{0+} = \frac{h_r^2}{4\pi^2 \epsilon}, \quad B^{0+} = \frac{h_r^2}{4\pi^2}, \quad \Delta_0^{0+} = \frac{h_{r,*}^2}{8\pi^2} = \frac{9}{8\pi^2} \approx 0.113986, \tag{D.22}$$

which matches the more direct calculation in section 4.1 performed using a hard cut-off.

We now proceed to compute $F_1$ in Eq. (D.13). We will only need the divergent part of $F_1$. Since $\phi_{cl}(\tau, \rho)$ is finite for $\rho \neq 0$ (and the integral in (D.13) is infra-red finite), we can just evaluate

$$F_{1,div} = \frac{\lambda_0}{24} \int_{\rho < \rho_*} d^D x \, \phi_{cl}^4(x), \tag{D.23}$$

where we fix $\rho_*$ to be $\epsilon$ independent and $\rho^* \ll L$. We begin by computing

$$F_{1,div}^{(-\infty,0)} = \frac{\lambda_0}{24} \int_{\rho<\rho_*,\tau<0} d^D x \, \phi_{cl}^4(x). \tag{D.24}$$

For $\tau < 0$,

$$\phi_{cl}(\tau,\rho) = \frac{h_0 c(D)}{D-3} \left( -|\tau|^{3-D} \mathcal{F}\left(\frac{\rho}{\tau}\right) + |L-\tau|^{3-D} \mathcal{F}\left(\frac{\rho}{L-\tau}\right) \right). \tag{D.25}$$

For $\rho \ll L$, the first term in the above equation dominates, so we approximate,

$$\phi_{cl}(\tau,\rho) \approx -\frac{h_0 c(D)|\tau|^{3-D}}{D-3} \mathcal{F}\left(\frac{\rho}{\tau}\right), \qquad \rho < \rho_*, \tau < 0. \tag{D.26}$$

We now evaluate

$$
\begin{aligned}
F_{1,div}^{0+,(-\infty,0)} &= \frac{\lambda_0}{24} \Omega_{D-2} \left(\frac{h_0 c(D)}{D-3}\right)^4 \int_0^{\rho_*} \rho^{D-2} d\rho \int_0^\infty d\tau \, \tau^{12-4D} \mathcal{F}^4\left(\frac{\rho}{\tau}\right) \\
&= \frac{\lambda_0}{24} \Omega_{D-2} \left(\frac{h_0 c(D)}{D-3}\right)^4 \int_0^{\rho_*} \rho^{11-3D} d\rho \int_0^\infty du \, u^{12-4D} \mathcal{F}^4\left(u^{-1}\right).
\end{aligned}
\tag{D.27}
$$

The $u$ integral is now finite and the only divergence comes from the $\rho$ integral. We have

$$x F(1, 1/2, 3/2, -x^2) = \tan^{-1} x. \tag{D.28}$$

Therefore,

$$F_{1,div}^{0+,(-\infty,0)} = \frac{\lambda_0 h_0^4 c(D)^4 \Omega_{D-2}}{24} \frac{\rho_*^{3\epsilon}}{3\epsilon} \int_0^\infty du \left(\tan^{-1}\frac{1}{u}\right)^4. \tag{D.29}$$

We have

$$\int_0^\infty du \left(\tan^{-1}\frac{1}{u}\right)^4 = \int_0^{\pi/2} \frac{z^4 dz}{\sin^2 z} = \frac{1}{4}(2\pi^3 \log 2 - 9\pi\zeta(3)). \tag{D.30}$$

Thus,

$$F_{1,div}^{0+,(-\infty,0)} = \frac{\lambda_r h_r^4 c(D)^4 \Omega_{D-2}}{288\epsilon}(2\pi^3 \log 2 - 9\pi\zeta(3)). \tag{D.31}$$

An equal contribution to $F_1$ comes from the region $\tau > L$.

Now we tackle the contribution to $F_1$ from region $0 < \tau < L$:

$$F_{1,div}^{0+,(0,L)} = \frac{\lambda_0}{24} \int_0^L d\tau \int_{\rho<\rho_*} d^{D-1}\rho \, \phi_{cl}^4(\tau,\rho). \tag{D.32}$$

First, from (D.19) we observe that there is a divergence coming from the $\rho \to 0$ limit with $0 < \tau < L$ – fixed. This is a "nested" divergence, which will be eventually cancelled by renormalization of $h_0$ in $F_0$. To compute it, let

$$
\begin{aligned}
F_{1,div}^{0+,nest} \equiv \frac{\lambda_0 h_0^4 c(D)^4}{24} \Omega_{D-2} \int_0^L d\tau \int_0^{\rho_*} d\rho \rho^{D-2} &\left[ \left(\frac{\sqrt{\pi}\Gamma((D-3)/2)}{\Gamma(D/2-1)}\right)^4 \rho^{12-4D} \right. \\
&\left. + \frac{4}{3-D} \left(\frac{\sqrt{\pi}\Gamma((D-3)/2)}{\Gamma(D/2-1)}\right)^3 \rho^{9-3D}(\tau^{3-D} + (L-\tau)^{3-D}) \right].
\end{aligned}
\tag{D.33}
$$

The first term in the integrand integrates to $L$ times a UV divergent constant that is zero in dimensional regularization. The second term gives

$$F_{1,div}^{0+,nest} = \frac{\lambda_0 h_0^4 c(D)^4}{6(3-D)} \Omega_{D-2} \left( \frac{\sqrt{\pi}\Gamma((D-3)/2)}{\Gamma(D/2-1)} \right)^3 \frac{L^\epsilon \rho_*^{2\epsilon}}{\epsilon^2}. \tag{D.34}$$

Now, we consider $F_{1,div}^{0+,(0,L)} - F_{1,div}^{0+,nest}$, using

$$\phi_{cl}(x) = \phi_1(x) + \phi_2(x) + \phi_3(x), \qquad 0 < \tau < L, \tag{D.35}$$

$$\phi_1(\rho,\tau) = -h_0 c(D) \frac{\sqrt{\pi}\Gamma(\frac{D-3}{2})}{\Gamma(\frac{D}{2}-1)} \rho^{3-D},$$

$$\phi_2(\rho,\tau) = \frac{h_0 c(D)}{D-3} \tau^{3-D} \mathcal{F}\left(\frac{\rho}{\tau}\right),$$

$$\phi_3(\rho,\tau) = \frac{h_0 c(D)}{D-3} (L-\tau)^{3-D} \mathcal{F}\left(\frac{\rho}{L-\tau}\right). \tag{D.36}$$

The only divergences in $F_{1,div}^{0+,(0,L)} - F_{1,div}^{0+,nest}$ now come from the regions $\tau \to 0, \rho \to 0$ and $\tau \to L, \rho \to 0$. By symmetry these two regions contribute equally, so let's focus on the $\tau \to 0, \rho \to 0$ region. Then

$$F_{1,div}^{0+,(0,L)} - F_{1,div}^{0+,nest} = 2 \cdot \frac{\lambda_0 h_0^4 c(D)^4}{24} \Omega_{D-2} \int_0^{\rho_*} d\rho \rho^{D-2} \int_0^\infty d\tau$$

$$\left[ \frac{4}{3-D} \left( \frac{\sqrt{\pi}\Gamma((D-3)/2)}{\Gamma(D/2-1)} \right)^3 \rho^{9-3D} \tau^{3-D} \left( \mathcal{F}\left(\frac{\rho}{\tau}\right) - 1 \right) \right.$$

$$+ \frac{6}{(3-D)^2} \left( \frac{\sqrt{\pi}\Gamma((D-3)/2)}{\Gamma(D/2-1)} \right)^2 \rho^{6-2D} \tau^{6-2D} \mathcal{F}^2\left(\frac{\rho}{\tau}\right)$$

$$\left. + \frac{4}{(3-D)^3} \frac{\sqrt{\pi}\Gamma((D-3)/2)}{\Gamma(D/2-1)} \rho^{3-D} \tau^{9-3D} \mathcal{F}^3\left(\frac{\rho}{\tau}\right) + \frac{1}{(3-D)^4} \tau^{12-4D} \mathcal{F}^4\left(\frac{\rho}{\tau}\right) \right]$$

$$= 2 \cdot \frac{\lambda_0 h_0^4 c(D)^4}{24} \Omega_{D-2} \int_0^{\rho_*} d\rho \rho^{11-3D} (I_1 + I_2 + I_3 + I_4)$$

$$= \frac{\lambda_0 h_0^4 c(D)^4}{12} \Omega_{D-2} \frac{\rho_*^{3\epsilon}}{3\epsilon} (I_1 + I_2 + I_3 + I_4), \tag{D.37}$$

where

$$I_1 = \frac{4}{3-D} \left( \frac{\sqrt{\pi}\Gamma((D-3)/2)}{\Gamma(D/2-1)} \right)^3 \int_0^\infty du\, u^{3-D} \left( \mathcal{F}(u^{-1}) - 1 \right),$$

$$I_2 = \frac{6}{(3-D)^2} \left( \frac{\sqrt{\pi}\Gamma((D-3)/2)}{\Gamma(D/2-1)} \right)^2 \int_0^\infty du\, u^{6-2D} \mathcal{F}^2(u^{-1}),$$

$$I_3 = \frac{4}{(3-D)^3} \frac{\sqrt{\pi}\Gamma((D-3)/2)}{\Gamma(D/2-1)} \int_0^\infty du\, u^{9-3D} \mathcal{F}^3(u^{-1}),$$

$$I_4 = \frac{1}{(3-D)^4} \int_0^\infty du\, u^{12-4D} \mathcal{F}^4(u^{-1}). \tag{D.38}$$

The integrals $I_{1-4}$ need to be evaluated to $O(\epsilon^0)$. For $I_{2-4}$ we can just directly take the $\epsilon \to 0$ limit:

$$I_2 = 6\pi^2 \int_0^\infty du \left(\tan^{-1}\frac{1}{u}\right)^2 = 6\pi^2 \int_0^{\pi/2} \frac{z^2 dz}{\sin^2 z} = 6\pi^3 \log 2.$$

$$I_3 = -4\pi \int_0^\infty du \left(\tan^{-1}\frac{1}{u}\right)^3 = -4\pi \int_0^{\pi/2} \frac{z^3 dz}{\sin^2 z} = -\frac{3\pi}{2}(2\pi^2 \log 2 - 7\zeta(3)).$$

$$I_4 = \int_0^\infty du \left(\tan^{-1}\frac{1}{u}\right)^4 = \int_0^{\pi/2} \frac{z^4 dz}{\sin^2 z} = \frac{\pi}{4}(2\pi^2 \log 2 - 9\zeta(3)). \tag{D.39}$$

For $I_1$, we divide the $u$ integral into $u \in (0, u_*)$ and $u \in (u_*, \infty)$ with $u_* \ll 1$ – fixed and $\epsilon$-independent. Then for $u > u_*$ we may take the $\epsilon \to 0$ limit of the integrand. For $u < u_*$, we use

$$u^{3-D} F(D/2 - 1, (D-3)/2, (D-1)/2, -u^{-2})$$
$$= (3 - D)u F(D/2 - 1, 1/2, 3/2, -u^2) + \frac{\sqrt{\pi}\Gamma((D-1)/2)}{\Gamma(D/2 - 1)} = O(u^0). \tag{D.40}$$

Therefore to $O(\epsilon^0)$,

$$I_1 = \frac{4}{3 - D} \left(\frac{\sqrt{\pi}\Gamma((D-3)/2)}{\Gamma(D/2 - 1)}\right)^3 \left[-\int_0^{u_*} du\, u^{3-D} + \int_{u_*}^\infty du \left(\tan^{-1}\left(\frac{1}{u}\right) - \frac{1}{u}\right)\right]$$
$$= \frac{4}{\epsilon} \left(\frac{\sqrt{\pi}\Gamma((D-3)/2)}{\Gamma(D/2 - 1)}\right)^3. \tag{D.41}$$

Combining all terms,

$$F_{1,div}^{0+,(0,L)} - F_{1,div}^{0+,nest} = \frac{\lambda_0 h_0^4 c(D)^4}{12} \Omega_{D-2} \frac{\rho_*^{3\epsilon}}{3\epsilon} \left[\frac{4}{\epsilon}\left(\frac{\sqrt{\pi}\Gamma((D-3)/2)}{\Gamma(D/2-1)}\right)^3 + \frac{7\pi^3}{2}\log 2 + \frac{33\pi}{4}\zeta(3)\right], \tag{D.42}$$

and

$$F_{1,div}^{0+} = -\frac{\lambda_r h_r^4 c(D)^4 \Omega_{D-2}}{18\epsilon^2} \left(\frac{\sqrt{\pi}\Gamma((D-3)/2)}{\Gamma(D/2-1)}\right)^3 \left(1 + \epsilon(3\log \mu L - 2\log 2 - \frac{3\zeta(3)}{\pi^2} + 3)\right). \tag{D.43}$$

Now, combining $F_{1,div}^{0+}$ with $F_0^{0+}$ and expressing $h_0$ in $F_0^{0+}$ in terms of $h_r$ using Eq. (D.3), we obtain

$$C^{0+}(\lambda_r, h_r) = (F_0^{0+} + F_1^{0+})_{div} = \frac{h_r^2}{4\pi^2 \epsilon} + \frac{\lambda_r h_r^4}{576\pi^4 \epsilon^2} \left(1 + \frac{\epsilon}{2}\left(\frac{3\zeta(3)}{\pi^2} - 1\right)\right), \tag{D.44}$$

so

$$B^{0+}(\lambda_r, h_r) = \frac{h_r^2}{4\pi^2} + \frac{\lambda_r h_r^4}{384\pi^4}\left(\frac{3\zeta(3)}{\pi^2} - 1\right) + O(\lambda_r^2). \tag{D.45}$$

Therefore,

$$\Delta_0^{0+} = \frac{B^{0+}(\lambda_{r,*}, h_{r,*})}{2} \approx \frac{h_{r,*}^2}{8\pi^2}\left(1 + \frac{\lambda_{r,*}h_{r,*}^2}{6(4\pi)^2}\left(\frac{3\zeta(3)}{\pi^2} - 1\right)\right)$$

$$= \frac{9}{8\pi^2}\left(1 + \epsilon\left\{\frac{73}{54} + \frac{1}{2}\left(\frac{3\zeta(3)}{\pi^2} - 1\right)\right\} + O(\epsilon^2)\right). \tag{D.46}$$

Note that the $O(\epsilon)$ correction to the scaling dimension in curly brackets above contains two terms: the first comes from the $O(\epsilon)$ correction to $h_{r,*}^2$ and the second from $F_1$. Numerically

$$\Delta_0^{0+} \approx 0.113986(1 + \epsilon(1.35185 - 0.317309)) = 0.113986(1 + 1.03454\epsilon). \tag{D.47}$$

We see that the $O(\epsilon)$ correction to $\Delta_0^{0+}$ is of the same order as the bare term. Moreover, this correction increases the scaling dimension. However, we know that for $D = 2$, $\Delta_0^{0+} = 0.03125$, i.e. as $D$ is lowered from 4 to 2, $\Delta_0^{0+}$ has to eventually decrease. We conclude that $\Delta_0^{0+}$ actually has non-monotonic dependence on $D$.

### D.3 Leading domain wall dimension to $O(\epsilon)$

We now proceed to the leading domain wall defect. We still have equations (D.12),(D.13), (D.14), except now

$$h(\tau) = \begin{cases} -h_0, & 0 < \tau < L, \\ h_0, & \text{otherwise.} \end{cases} \tag{D.48}$$

Then

$$\phi_{cl}(\tau, \rho) = 2h_0 c(D)\rho^{3-D}(f(\tau/\rho) + f((L-\tau)/\rho) - f(\infty)), \tag{D.49}$$

with $f(u)$ given by Eq. (D.17) and $f(\infty) = \frac{\sqrt{\pi}\Gamma((D-3)/2)}{2\Gamma(D/2-1)}$. In other words,

$$\phi_{cl}(\tau, \rho) = 2h_0 c(D)\left[\frac{\sqrt{\pi}\Gamma((D-3)/2)}{2\Gamma(D/2-1)}(\text{sgn}(\tau) + \text{sgn}(L-\tau) - 1)\rho^{3-D}\right.$$

$$\left. + \frac{\text{sgn}(\tau)|\tau|^{3-D}}{3-D}\mathcal{F}(\rho/\tau) + \frac{\text{sgn}(L-\tau)|L-\tau|^{3-D}}{3-D}\mathcal{F}(\rho/(L-\tau))\right], \tag{D.50}$$

with $\mathcal{F}$ given by Eq. (D.18). Using dimensional regularization,

$$\phi_{cl}(\tau, 0) = \frac{2h_0 c(D)}{3-D}(\text{sgn}(\tau)|\tau|^{3-D} + \text{sgn}(L-\tau)|L-\tau|^{3-D}). \tag{D.51}$$

Therefore, from (D.12),

$$F_0^{+-} = \frac{1}{2}\int d\tau \phi_{cl}(\tau)h(\tau) = \frac{4h_0^2 c(D)}{D-3}\frac{L^\epsilon}{\epsilon} = 4F_0^{0+} = \frac{h_0^2\mu^{-\epsilon}}{\pi^2\epsilon}\left[1 + \epsilon(\frac{1}{2}\log\pi + \frac{\gamma}{2} + 1 + \log\mu L) + O(\epsilon^2)\right]. \tag{D.52}$$

We now proceed to $F_1^{+-}$. The divergent part is again given by Eq. (D.23) with fixed $\rho_* \ll L$. We begin by extracting the non-local divergence. From (D.50),

$$\phi_{cl}(\tau,\rho) \overset{\rho \to 0}{\approx} 2h_0 c(D)\left[\frac{\sqrt{\pi}\Gamma((D-3)/2)}{2\Gamma(D/2-1)}(\text{sgn}(\tau)+\text{sgn}(L-\tau)-1)\rho^{3-D} \right.$$
$$\left. +\frac{\text{sgn}(\tau)|\tau|^{3-D}}{3-D}+\frac{\text{sgn}(L-\tau)|L-\tau|^{3-D}}{3-D}\right]. \tag{D.53}$$

We then let

$$F_{1,div}^{+-,nest} \equiv \frac{\lambda_0}{24}\frac{8h_0^4 c(D)^4}{3-D}\left(\frac{\sqrt{\pi}\Gamma((D-3)/2)}{\Gamma(D/2-1)}\right)^3 \int_{\rho<\rho_*} d\tau d^{D-1}\Big[\rho\left(\text{sgn}(\tau)+\text{sgn}(L-\tau)-1\right)$$
$$\times\ (\text{sgn}(\tau)|\tau|^{3-D}+\text{sgn}(L-\tau)|L-\tau|^{3-D})\rho^{9-3D}\Big]$$
$$= \frac{2}{3}\frac{\lambda_0 h_0^4 c(D)^4 \Omega_{D-2}}{3-D}\left(\frac{\sqrt{\pi}\Gamma((D-3)/2)}{\Gamma(D/2-1)}\right)^3 \frac{\rho_*^{2\epsilon}L^\epsilon}{\epsilon^2}. \tag{D.54}$$

Notice that this is four times bigger than the corresponding expression (D.34) for the defect endpoint operator.

We can now evaluate $F_{1,div}^{+-} - F_{1,div}^{+-,nest}$. This will only receive divergent contributions from $\rho \to 0$, $\tau \to 0$ and $\rho \to 0$, $\tau \to L$ regions. By reflection symmetry the contribution from both regions is the same. So we may just focus on the $\tau \to 0$ region:

$$F_{1,div}^{+-} - F_{1,div}^{+-,nest} = 2\cdot\frac{\lambda_0}{24}\cdot 16h_0^4 c(D)^4 \int_{\rho<\rho_*} d\tau d^{D-1}\rho\Bigg[$$
$$\frac{4}{3-D}\left(\frac{\sqrt{\pi}\Gamma((D-3)/2)}{2\Gamma(D/2-1)}\right)^3 \rho^{9-3D}|\tau|^{3-D}(\mathcal{F}(\rho/\tau)-1)$$
$$+\frac{6}{(3-D)^2}\left(\frac{\sqrt{\pi}\Gamma((D-3)/2)}{2\Gamma(D/2-1)}\right)^2 \rho^{6-2D}|\tau|^{6-2D}\mathcal{F}(\rho/\tau)^2$$
$$+\frac{4}{(3-D)^3}\frac{\sqrt{\pi}\Gamma((D-3)/2)}{2\Gamma(D/2-1)}\rho^{3-D}|\tau|^{9-3D}\mathcal{F}(\rho/\tau)^3$$
$$+\frac{1}{(3-D)^4}|\tau|^{12-4D}\mathcal{F}(\rho/\tau)^4\Bigg]$$
$$= \frac{8}{9\epsilon}\lambda_0 h_0^4 c(D)^4 \Omega_{D-2}\rho_*^{3\epsilon}\left(\frac{1}{8}I_1 + \frac{1}{4}I_2 + \frac{1}{2}I_3 + I_4\right)$$
$$= \frac{4}{9\epsilon}\lambda_0 h_0^4 c(D)^4 \Omega_{D-2}\rho_*^{3\epsilon}\left(\frac{1}{\epsilon}\left(\frac{\sqrt{\pi}\Gamma((D-3)/2)}{\Gamma(D/2-1)}\right)^3 + \pi^3 \log 2 + 6\pi\zeta(3)\right). \tag{D.55}$$

Combining (D.55) and (D.54),

$$F_{1,div}^{+-} = -\frac{2\pi^3}{9\epsilon^2}\lambda_0 h_0^4 c(D)^4 \Omega_{D-2}\mu^{-3\epsilon}\left(1 + \epsilon(3\log \mu L + 3 + \log 2 - \frac{12\zeta(3)}{\pi^2})\right). \tag{D.56}$$

Further combining this with (D.52),

$$C^{+-}(\lambda_r, h_r) \approx (F_0^{+-} + F_1^{+-})_{div} = \frac{h_r^2}{\pi^2 \epsilon} + \frac{\lambda_r h_r^4}{144\pi^4 \epsilon^2}\left(1 + \epsilon(\frac{6\zeta(3)}{\pi^2} - \frac{1}{2})\right), \tag{D.57}$$

so

$$B^{+-}(\lambda_r, h_r) = \frac{h_r^2}{\pi^2} + \frac{\lambda_r h_r^4}{96\pi^4}\left(\frac{12\zeta(3)}{\pi^2} - 1\right). \tag{D.58}$$

Therefore,

$$\Delta_0^{+-} = \frac{9}{2\pi^2}\left[1 + \epsilon\left\{\frac{73}{54} + \frac{1}{2}\left(\frac{12\zeta(3)}{\pi^2} - 1\right)\right\}\right] \approx 0.455945(1 + \epsilon(1.35185 + 0.230763))$$
$$= 0.455945(1 + 1.58261\epsilon).$$

Here the first term in the curly brackets comes from the $O(\epsilon)$ correction to $h_{r,*}^2$ and is numerically large, and the second term comes from the $O(\lambda_r)$ correction to $B^{+-}$ and is numerically smaller.

### D.4   Next defect endpoint dimension to $O(\epsilon)$

In this section we compute the scaling dimension of the next defect endpoint primary operator $\phi_1^{0+}$ to $O(\epsilon)$. As explained in section 4.2, we need to consider mixing of the operators $O_1$ and $O_2$ in Eq. (4.18). To find the anomalous dimension matrix we consider the correlators:

$$\langle \phi_0^{0+}(L)O_{1,2}(0)\phi(x)\rangle. \tag{D.59}$$

Here $x$ need not lie on the defect axis. We write $\phi(x) = \phi_{cl}(x) + \eta(x)$, where $\phi_{cl}$ is given by (D.16). The action in the presence of the defect becomes:

$$S = F_0^{0+} + \int d^D x \left[\frac{1}{2}(\partial_\mu \eta)^2 + \frac{\lambda_0}{24}(\phi_{cl}^4 + 4\phi_{cl}^3\eta + 6\phi_{cl}^2\eta^2 + 4\phi_{cl}\eta^3 + \eta^4)\right], \tag{D.60}$$

where $F_0^{0+}$ is given by Eqs. (D.12), (D.21). We compute the correlators (D.59) to first order in $\lambda$:

$$\langle \phi_0^{0+}(L)O_1(0)\phi(x)\rangle = e^{-F_0^{0+}-F_1^{0+}}\left[\phi_{cl}^2(0)\phi_{cl}(x) + 2\phi_{cl}(0)G(x) - \frac{\lambda_0}{6}\phi_{cl}^2(0)\int d^D y\, \phi_{cl}^3(y)G(x-y)\right.$$
$$- \frac{\lambda_0}{3}(\phi_{cl}(0)\phi_{cl}(x) + G(x))\int d^D y\, \phi_{cl}^3(y)G(y) - \lambda_0\phi_{cl}(0)\int d^D y\, \phi_{cl}^2(y)G(y)G(y-x)$$
$$\left. - \frac{\lambda_0}{2}\phi_{cl}(x)\int d^D y\, \phi_{cl}^2(y)G(y)^2 - \lambda_0\int d^D y\, \phi_{cl}(y)G(y-x)G(y)^2\right].$$

$$\langle \phi_0^{0+}(L)O_2(0)\phi(x)\rangle = e^{-F_0^{0+}-F_1^{0+}}\left[\partial_\tau \phi_{cl}(0)\phi_{cl}(x) - \partial_\tau G(x) - \frac{\lambda_0}{6}\partial_\tau\phi_{cl}(0)\int d^D y\, \phi_{cl}^3(y)G(x-y)\right.$$
$$\left. + \frac{\lambda_0}{6}\phi_{cl}(x)\int d^D y\, \phi_{cl}^3(y)\partial_\tau G(y) + \frac{\lambda_0}{2}\int d^D y\, \phi_{cl}^2(y)G(y-x)\partial_\tau G(y)\right], \tag{D.61}$$

where $F_1^{0+}$ is given by Eq. (D.13). The calculation of the divergent parts of the integrals above is similar to that in sections D.2, D.3, yielding:

$$\int d^D y\, \phi_{cl}^3(y) G(x-y) \to \frac{h_0^2 \mu^{-2\epsilon}}{32\pi^2\epsilon}\phi_{cl}(x),$$

$$\int d^D y\, \phi_{cl}^3(y) G(y) \to \frac{h_0^3 \mu^{-2\epsilon}}{64\pi^4\epsilon L^{D-3}},$$

$$\int d^D y\, \phi_{cl}^2(y) G(y) \to \frac{h_0^2 \mu^{-2\epsilon}}{96\pi^2\epsilon},$$

$$\int d^D y\, \phi_{cl}^2(y) G(y)^2 \to \frac{h_0^2 \mu^{-3\epsilon}}{256\pi^6\epsilon L^2},$$

$$\int d^D y\, \phi_{cl}(y) G(y)^2 G(x-y) \to \frac{h_0 \mu^{-\epsilon}}{32\pi^4\epsilon L^{D-3}} G(x) + \frac{h_0 \mu^{-2\epsilon}}{128\pi^4\epsilon}\partial_\tau G(x),$$

$$\int d^D y\, \phi_{cl}^3(y) \partial_\tau G(y) \to \frac{h_0^3 \mu^{-3\epsilon}}{256\pi^6\epsilon L^2}(3 - 4\pi^2),$$

$$\int d^D y\, \phi_{cl}^2(y) \partial_\tau G(y) G(x-y) \to \frac{h_0^2 \mu^{-2\epsilon}}{16\pi^4\epsilon L} G(x) + \frac{h_0^2 \mu^{-2\epsilon}}{64\pi^4\epsilon}(1 + \tfrac{2}{3}\pi^2)\partial_\tau G(x). \quad \text{(D.62)}$$

Using $\phi_{cl}(0) = \frac{h_0 c(D) L^{3-D}}{D-3}$, $\partial_\tau \phi_{cl}(0) = c(D) h_0 L^{2-D}$, we then obtain

$$
\begin{aligned}
\langle \phi_0^{0+}(L) O_1(0)\phi(x)\rangle e^{F_0^{0+}+F_1^{0+}} &= \frac{h_0^2 c(D)^2}{L^{2(D-3)}}\phi_{cl}(x)\left(1 - \frac{5}{192\pi^2\epsilon}\lambda_0 h_0^2 \mu^{-2\epsilon} - \frac{1}{32\pi^2\epsilon}\lambda_0\mu^{-\epsilon}\right) \\
&\quad + \frac{2h_0 c(D)}{L^{D-3}}G(x)\left(1 - \frac{1}{64\pi^2\epsilon}\lambda_0 h_0^2 \mu^{-2\epsilon} - \frac{1}{16\pi^2\epsilon}\lambda_0\mu^{-\epsilon}\right) \\
&\quad - \frac{\lambda_0 h_0 \mu^{-2\epsilon}}{128\pi^4\epsilon}\partial_\tau G(x).
\end{aligned}
$$
$$(\text{D.63})$$

$$
\begin{aligned}
\langle \phi_0^{0+}(L) O_2(0)\phi(x)\rangle e^{F_0^{0+}+F_1^{0+}} &= \frac{h_0 c(D)}{L^{D-2}}\phi_{cl}(x)\left(1 + \frac{1 - 2\pi^2}{128\pi^4\epsilon}\lambda_0 h_0^2 \mu^{-2\epsilon}\right) \\
&\quad - \partial_\tau G(x)\left(1 - \frac{2\pi^2 + 3}{384\pi^4\epsilon}\lambda_0 h_0^2 \mu^{-2\epsilon}\right) + \frac{\lambda_0 h_0^2 \mu^{-\epsilon}}{32\pi^4\epsilon L^{D-3}}G(x).
\end{aligned}
$$
$$(\text{D.64})$$

Here we have only kept the divergent part of the $O(\lambda)$ terms. We are now ready to find the $O_1, O_2$ mixing matrix. Let

$$O_i = \mathcal{Z}_{ij}(\mu, \lambda_r, h_r) O_{j,r}. \tag{D.65}$$

We will choose $\mathcal{Z}_{ij}$ in such a way that $O_{1,r}$ and $O_{2,r}$ have the same engineering dimension $\Delta_{\text{eng}} = [O_1]_{\text{eng}} = D - 2$. The anomalous dimension matrix is:

$$\gamma_{ij} = \mathcal{Z}_{ik}^{-1}\left[\mu\frac{\partial}{\partial\mu} + \beta(\lambda_r)\frac{\partial}{\partial\lambda_r} + \beta(h_r)\frac{\partial}{\partial h_r}\right]\mathcal{Z}_{kj}. \tag{D.66}$$

Letting $\gamma^{(p)}$, $p = 1, 2$ be the eigenvalues of the matrix $\gamma(\lambda_{r,*}, h_{r,*})$, the dimensions of the scaling operators are

$$\Delta^{(p)} = \gamma^{(p)} + \Delta_{\text{eng}}. \tag{D.67}$$

We recall from section D.1 that

$$\phi_0^{0+} = e^{-C^{0+}/2}\phi_{0,r}^{0+}, \tag{D.68}$$

where $C^{0+}$ is defined in (D.5) and explicitly given by Eq. (D.44). From (D.63) and (D.64), after expressing $h_0$ in terms of $h_r$, we obtain:

$$\mathcal{Z}^{-1} = e^{C^{0+}/2} \begin{pmatrix} 1 + \frac{\lambda_r(h_r^2+6)}{96\pi^2\epsilon} & -\frac{\lambda_r h_r \mu^{-\epsilon/2}}{128\pi^4\epsilon} \\ -\frac{\lambda_r h_r}{16\pi^2\epsilon} & \mu^{-\epsilon/2}\left(1 + \frac{2\pi^2+3}{384\pi^4\epsilon}\lambda_r h_r^2\right). \end{pmatrix}. \tag{D.69}$$

$$\gamma = \frac{1}{2}B^{0+}(h_r, \lambda_r)\mathbb{1} + \begin{pmatrix} \frac{\lambda_r(h_r^2+3)}{48\pi^2} & -\frac{\lambda_r h_r}{64\pi^4} \\ -\frac{\lambda_r h_r}{16\pi^2} & \frac{\epsilon}{2} + \frac{2\pi^2+3}{192\pi^4}\lambda_r h_r^2 \end{pmatrix}. \tag{D.70}$$

Substituting the fixed point values $\lambda_{r,*}$ and $h_{r,*}$ in Eq. (4.4), we obtain:

$$\gamma^{(1)} = \Delta_0^{0+} + \epsilon, \qquad \gamma^{(2)} = \Delta_0^{0+} + \left(\frac{4}{3} + \frac{3}{4\pi^2}\right)\epsilon. \tag{D.71}$$

From this

$$\Delta^{(1)} = \Delta_0^{0+} + 2, \qquad \Delta^{(2)} = \Delta_0^{0+} + 2 + \left(\frac{1}{3} + \frac{3}{4\pi^2}\right)\epsilon + O(\epsilon^2). \tag{D.72}$$

Clearly, $\Delta^{(1)}$ corresponds to the second descendant of $\phi_0^{0+}$. $\Delta_1^{0+} = \Delta^{(2)}$ corresponds to the next endpoint primary.

## D.5   OPE coefficients

In this appendix, we compute $\lambda_{\sigma00}^{0+0}$ and $\lambda_{\epsilon00}^{0+0}$ to $O(\epsilon)$, proceeding as outlined below Eq. (4.20). As in section D.4, we write $\phi(x) = \phi_{cl}(x) + \eta(x)$ and use Eq. (D.60). For $\mathcal{O} = \sigma$ we obtain to first order in $\lambda$

$$\frac{\langle\phi(\tau)\rangle_h}{\langle 1\rangle_h} \approx \phi_{cl}(\tau) - \frac{\lambda_0}{6}\int d^D y\, \phi_{cl}^3(y)G(\tau - y_\tau, \vec{y}), \tag{D.73}$$

while for $\mathcal{O} = \epsilon$,

$$\frac{\langle\phi^2(\tau)\rangle_h}{\langle 1\rangle_h} \approx \phi_{cl}^2(\tau) - \frac{\lambda_0}{3}\phi_{cl}(\tau)\int d^D y\, \phi_{cl}^3(y)G(\tau - y_\tau, \vec{y}) - \frac{\lambda_0}{2}\int d^D y\, \phi_{cl}^2(y)G^2(\tau - y_\tau, \vec{y}). \tag{D.74}$$

To evaluate these integrals, we use Eq. (D.16) and take $L \to \infty$ from the outset to obtain:

$$\phi_{cl}(\tau, \rho) = -c(D)h_0\left[\theta(\tau)\frac{\sqrt{\pi}\Gamma((D-3)/2)}{\Gamma(D/2-1)}\frac{1}{\rho^{D-3}} - \frac{1}{D-3}\frac{\text{sgn}(\tau)}{|\tau|^{D-3}}\mathcal{F}(\rho/\tau))\right], \tag{D.75}$$

with $\mathcal{F}$ given by Eq. (D.18).

## D.5.1 $\lambda_{\sigma 00}^{0+0}$

We begin by evaluating the integral in Eq. (D.73). Let's start with the contribution of the $y_\tau < 0$ region. Recalling we are interested in $\tau < 0$,

$$\left[\frac{\langle \phi(\tau)\rangle_h}{\langle 1\rangle_h}\right]_{1,y_\tau<0} = \frac{\lambda_0 c(D)^4 h_0^3}{6(D-3)^3}\int_0^\infty dy_\tau \int d^{D-1}\vec{y}\,\frac{1}{|y_\tau|^{3(D-3)}}\mathcal{F}^3(|\vec{y}|/y_\tau)\frac{1}{((|\tau|-y_\tau)^2+\vec{y}^2)^{D/2-1}}. \tag{D.76}$$

The subscript 1 on the LHS denotes a first order contribution in $\lambda$. This is actually a non-divergent integral, so to the order in $\epsilon$ that we are working here we can set $D = 4$, so that $\mathcal{F}(u) = \tan^{-1}(u)/u$. Changing variables to $|\vec{y}|/y_\tau = v$ and $y_\tau = \tau \tilde{y}_\tau$ (and dropping the tilde),

$$\left[\frac{\langle \phi(\tau)\rangle_h}{\langle 1\rangle_h}\right]_{1,y_\tau<0} = \frac{1}{6}\lambda_0 c(4)^4 h_0^3 \Omega_2 |\tau|^{-1+3\epsilon}\int_0^\infty dy_\tau \int_0^\infty dv\, v^2 \mathcal{F}^3(v)\frac{1}{(1-y_\tau)^2+y_\tau^2 v^2}. \tag{D.77}$$

The $y_\tau$ integral gives:

$$\int_0^\infty dy_\tau \frac{1}{(1-y_\tau)^2+y_\tau^2 v^2} = \frac{1}{v}(\pi - \tan^{-1}v), \tag{D.78}$$

so that

$$\left[\frac{\langle \phi(\tau)\rangle_h}{\langle 1\rangle_h}\right]_{1,y_\tau<0} = \frac{1}{6(4\pi)^4}\lambda_0 h_0^3 |\tau|^{-1+3\epsilon}\left(\log 2 - \frac{3\zeta(3)}{2\pi^2}\right). \tag{D.79}$$

Next, we compute the contribution of the $y_\tau > 0$ region to the integral in Eq. (D.73).

$$\left[\frac{\langle \phi(\tau)\rangle_h}{\langle 1\rangle_h}\right]_{1,y_\tau>0} = \frac{1}{6}\lambda_0 c(D)^4 h_0^3\int_0^\infty dy_\tau \int d^{D-1}\vec{y}\left(\frac{\sqrt{\pi}\Gamma((D-3)/2)}{\Gamma(D/2-1)}y^{3-D} - \frac{1}{D-3}|y_\tau|^{3-D}\mathcal{F}(|\vec{y}|/y_\tau)\right)^3 \times$$

$$\times\frac{1}{((|\tau|+y_\tau)^2+\vec{y}^2)^{D/2-1}} = \frac{1}{6}\lambda_0 c(D)^4 h_0^3 \Omega_{D-2}|\tau|^{-1+3\epsilon}\int_0^\infty dy_\tau \int_0^\infty \frac{dy}{y}\left[y^{2\epsilon}\left(\frac{\sqrt{\pi}\Gamma((D-3)/2)}{\Gamma(D/2-1)}\right)^3\right.$$

$$\left.-3\pi^2\tan^{-1}(y/y_\tau) + 3\pi\left(\tan^{-1}(y/y_\tau)\right)^2 - \left(\tan^{-1}(y/y_\tau)\right)^3\right]\frac{1}{((1+y_\tau)^2+y^2)^{D/2-1}}$$

$$= \frac{1}{6}\lambda_0 c(D)^4 h_0^3 \Omega_{D-2}|\tau|^{-1+3\epsilon}\left[\left(\frac{\sqrt{\pi}\Gamma((D-3)/2)}{\Gamma(D/2-1)}\right)^3\frac{1}{D-3}\int_0^\infty dy\, y^{-1+2\epsilon}\mathcal{F}(y) + \right.$$

$$\left.+\int_0^\infty dy_\tau \int_0^\infty \frac{dv}{v}\left(-3\pi^2\tan^{-1}v + 3\pi(\tan^{-1}v)^2 - (\tan^{-1}v)^3\right)\frac{1}{(1+y_\tau)^2+v^2 y_\tau^2}\right]. \tag{D.80}$$

In the first step above we've set $D = 4$ in all but the first term in the square brackets, since these terms do not give a divergence as $D \to 4$. In the second step, we've performed the integral over $y_\tau$ for the first term in square brackets (note this is an integral we had already encountered in Eq. (D.16)) and changed variables to $v = y/y_\tau$ for the remaining terms. Now, $\int_0^\infty dy\, y^{-1+2\epsilon}\mathcal{F}(y) = \frac{1}{2\epsilon}+1+O(\epsilon)$, and $\int_0^\infty dy_\tau \frac{1}{(1+y_\tau)^2+v^2 y_\tau^2} = \frac{\tan^{-1}v}{v}$. The remaining integrals over $v$ in the last line of Eq. (D.80) are (up to prefactors) the same as $I_2, I_3, I_4$ in Eq. (D.39), so

$$\left[\frac{\langle \phi(\tau)\rangle_h}{\langle 1\rangle_h}\right]_{1,y_\tau>0} = \frac{1}{6}\lambda_0 c(D)^4 h_0^3 \Omega_{D-2}\pi^3|\tau|^{-1+3\epsilon}\left(\frac{1}{2\epsilon}(1+3\epsilon)\left(\frac{\Gamma((D-3)/2)}{\sqrt{\pi}\Gamma(D/2-1)}\right)^3 - \frac{5}{4}\log 2 - \frac{45}{8\pi^2}\zeta(3)\right). \tag{D.81}$$

Combining this with the $y_\tau < 0$ contribution (D.79),

$$\left[\frac{\langle\phi(\tau)\rangle_h}{\langle 1\rangle_h}\right]_1 = \frac{1}{6}\lambda_0 c(D)^4 h_0^3 \Omega_{D-2}\pi^3 |\tau|^{-1+3\epsilon}\left(\frac{1}{2\epsilon}(1+3\epsilon)\left(\frac{\Gamma((D-3)/2)}{\sqrt{\pi}\Gamma(D/2-1)}\right)^3 - \log 2 - \frac{6\zeta(3)}{\pi^2}\right).$$
(D.82)

Combining this with the zeroth order contribution in $\lambda$ in Eq. (D.75) and expressing $h_0$ and $\lambda_0$ in terms of $h_r$ and $\lambda_r$,

$$\frac{\langle\phi(\tau)\rangle_h}{\langle 1\rangle_h} \approx -\frac{1}{4\pi^2}|\tau|^{-1+\epsilon/2}\left[h_r\left(1 + \frac{\epsilon}{2}(\log\mu|\tau|+2+\gamma_E+\log\pi)\right)\right.$$
$$\left.- \frac{\lambda_r h_r^3}{12(4\pi)^2}\left(2\log\mu|\tau|+3+\gamma_E+\log\pi - \frac{12\zeta(3)}{\pi^2}\right)\right].$$
(D.83)

Notice that the $1/\epsilon$ divergence has disappeared. Recall that to first order in $\lambda$ the bulk two point function of $\phi$ does not receive any corrections at the critical point. Thus, the bulk operator $\sigma$ normalized to identity is to first order in $\epsilon$, $\sigma = c(D)^{-1/2}\phi$. Using this normalization and inserting the fixed point values of $h_r$ and $\lambda_r$ in Eq. (4.4), we obtain[26]

$$\frac{\langle\sigma(\tau)\rangle_h}{\langle 1\rangle_h} = |\tau|^{-1+\epsilon/2}\frac{3}{2\pi}\left[1 + \epsilon\left(\frac{25}{27} + \frac{3\zeta(3)}{\pi^2}\right) + O(\epsilon^2)\right].$$
(D.84)

Eq. (4.21) then follows from (4.20).

## D.5.2 $\lambda_{\epsilon 00}^{0+0}$

We begin by normalizing the $\phi^2(x)$ operator in the bulk. We have

$$\langle\phi^2(x)\phi^2(0)\rangle = 2G^2(x) - \lambda_0\int d^d y\, G^2(x-y)G^2(y) + O(\lambda^2).$$
(D.85)

A standard calculation then yields in minimal subtraction

$$[\phi^2]_r = \left(1 + \frac{\lambda_r}{(4\pi)^2\epsilon} + O(\lambda^2)\right)\phi^2,$$
(D.86)

and normalizing the bulk two-point function of $\epsilon$ to 1,

$$\epsilon = 2\sqrt{2}\pi^2\mu^{\epsilon/3}\left(1 + \epsilon(\frac{1}{6} - \frac{1}{3}\gamma_E - \frac{1}{3}\log\pi)\right)[\phi^2]_r,$$
(D.87)

with $\Delta_\epsilon = 2 - \frac{2}{3}\epsilon + O(\epsilon^2)$.

Next, we proceed to Eq. (D.74). Again, we will be interested in evaluating this at $\tau < 0$. The integral in the second term on the RHS of (D.74) has already been evaluated when considering $\lambda_{\sigma 00}^{0+0}$, so it remains to compute the last term,

$$\left[\frac{\langle\phi^2(\tau)\rangle_h}{\langle 1\rangle_h}\right]_{1,\text{II}} \equiv -\frac{\lambda_0}{2}\int d^D y\, \phi_{cl}^2(y)G^2(\tau-y_\tau,\vec{y}).$$
(D.88)

---

[26]Note, we choose $h < 0$ for the $+$ defect, corresponding to a positive $\langle\phi\rangle$ profile for an infinite straight defect.

We first compute the contribution to the above integral from the $y_\tau < 0$ region. The only divergence in this region is a bulk divergence from $(y_\tau, y)$ approaching $(\tau, 0)$.

$$\left[\frac{\langle \phi^2(\tau)\rangle_h}{\langle 1\rangle_h}\right]_{1,\text{II},y_\tau<0} = -\frac{\lambda_0 c(D)^4 h_0^2}{2(D-3)^2}\int_0^\infty dy_\tau \int d^{D-1}\vec{y}\, y_\tau^{-2+2\epsilon}\mathcal{F}^2(|\vec{y}|/y_\tau)\frac{1}{((|\tau|-y_\tau)^2+\vec{y}^2)^{D-2}}.$$
(D.89)

As in section D.5.1 we pull out the $\tau$ scale dependence and switch variables to $v = |\vec{y}|/y_\tau$:

$$\left[\frac{\langle \phi^2(\tau)\rangle_h}{\langle 1\rangle_h}\right]_{1,\text{II},y_\tau<0} = -\frac{\lambda_0 c(D)^4 h_0^2 \Omega_{D-2}}{2(D-3)^3}|\tau|^{-2+3\epsilon}\int_0^\infty dv\, v^{2-\epsilon}\mathcal{F}^2(v)p(v,\epsilon),$$
(D.90)

where

$$p(v,\epsilon) = \int_0^\infty dy_\tau \frac{y_\tau^{1+\epsilon}}{((1-y_\tau)^2 + v^2 y_\tau^2)^{D-2}}.$$
(D.91)

Now,

$$p(v,\epsilon=0) = \frac{1}{2}\left(v^{-2} + v^{-3}(\pi - \tan^{-1}v)\right),$$

$$p(v,\epsilon) = \frac{\sqrt{\pi}\Gamma(D-5/2)}{\Gamma(D-2)}v^{-3+2\epsilon}, \qquad v \to 0.$$
(D.92)

Now the integral (D.90) can be performed to first subleading order in $\epsilon$:

$$\left[\frac{\langle \phi^2(\tau)\rangle_h}{\langle 1\rangle_h}\right]_{1,\text{II},y_\tau<0} = -\frac{\lambda_0 c(D)^4 h_0^2 \Omega_{D-2}\pi}{2(D-3)^3}|\tau|^{-2+3\epsilon}\left(\frac{\Gamma(3/2-\epsilon)}{\sqrt{\pi}\Gamma(2-\epsilon)}\frac{1}{\epsilon} - \frac{\log 2}{4} - \frac{\pi^2}{32} + \frac{3}{4}\right).$$
(D.93)

The contribution to (D.88) from the region $y_\tau > 0$ is non-divergent and evaluates to

$$\left[\frac{\langle \phi^2(\tau)\rangle_h}{\langle 1\rangle_h}\right]_{1,\text{II},y_\tau>0} = -\frac{\pi^2}{16}\lambda_0 c(4)^4 h_0^2|\tau|^{-2+3\epsilon}(\pi^2 + 16 - 8\log 2).$$
(D.94)

Combining this with Eq. (D.93),

$$\left[\frac{\langle \phi^2(\tau)\rangle_h}{\langle 1\rangle_h}\right]_{1,\text{II}} = -\frac{\lambda_0 h_0^2}{256\pi^6}|\tau|^{-2+3\epsilon}\left(\frac{1}{\epsilon} + \frac{9}{2} + \frac{3\gamma_E}{2} + \frac{3}{2}\log\pi\right).$$
(D.95)

We now combine all three terms on the RHS of Eq. (D.74) (using Eq. (D.82) for the second term), and trade $h_0$, $\lambda_0$, for $h_r$, $\lambda_r$ to obtain

$$\left[\frac{\langle [\phi^2]_r(\tau)\rangle_h}{\langle 1\rangle_h}\right] = \frac{h_r^2}{16\pi^4\tau^{2-\epsilon}}\left(1 + \epsilon(2 + \gamma_E + \log\pi + \log\mu|\tau|)\right)$$
$$+ \frac{\lambda_r h_r^4}{3\cdot 256\pi^6|\tau|^{2-\epsilon}}\left(-\frac{3}{2} - \frac{1}{2}\gamma_E - \frac{1}{2}\log\pi + \frac{6\zeta(3)}{\pi^2} - \log\mu|\tau|\right)$$
$$+ \frac{\lambda_r h_r^2}{256\pi^6|\tau|^{2-\epsilon}}\left(-\frac{5}{2} - \frac{1}{2}\gamma_E - \frac{1}{2}\log\pi - \log\mu|\tau|\right) + O(\lambda^2).$$
(D.96)

Note that the $1/\epsilon$ divergences have cancelled upon taking the renormalization of $\phi^2$ (D.86) into account. Substituting the fixed point values of $\lambda_r$ and $h_r$ and using the normalization of $\epsilon$ in Eq. (D.87),

$$\left[\frac{\langle\epsilon(\tau)\rangle_h}{\langle 1\rangle_h}\right] = \frac{9\sqrt{2}}{8\pi^2|\tau|^{\Delta_\epsilon}}\left(1 + \epsilon\left(\frac{32}{27} + \frac{6\zeta(3)}{\pi^2}\right) + O(\epsilon^2)\right), \tag{D.97}$$

which in turn immediately yields $\lambda_{\epsilon 00}^{0+0}$ in Eq. (4.22).

### D.6 Stress energy tensor OPE

In this appendix we compute $\lambda_{010}^{++0}$ corresponding to the three point function $\langle\phi_0^{0+}\phi_1^{++}\phi_0^{+0}\rangle$ and the OPE coefficient $b$ in Eq. (4.32) to leading order in $\epsilon$.

We recall that $\phi_1^{++}(\tau) \sim \phi(\tau, 0)$ in $\epsilon$ expansion. More precisely, we may write $\phi_1^{++}(\tau) = \mathcal{N}_{++}\mathcal{Z}_{++}(\lambda_r)\mu^{\Delta_1^{++}-(D-2)/2}\phi(\tau, 0)$, where $\mathcal{Z}_{++}(\lambda_r) = 1 + O(\lambda_r)$ is a renormalization factor in minimal subtraction and $\mathcal{N}_{++}$ is a finite factor to fix the $\phi_1^{++}$ two point function to 1. To leading order in $\lambda$ in dimensional regularization,

$$\frac{\langle\phi(\tau, 0)\phi(0)\rangle_h}{\langle 1\rangle_h} = G(\tau), \tag{D.98}$$

where the expectation value is in the presence of a uniform $h$ (infinite defect). Thus, $N_{++} = 2\pi + O(\epsilon)$. (Recall, $g = 1 + O(\epsilon)$,[23], so we don't have to take the $g$ factor into account to this order.)

To compute $\lambda_{010}^{++0}$, analogously to Eq. (4.20) we use

$$\lambda_{010}^{++0} = \lim_{L\to\infty}\tau^{\Delta_1^{++}}\frac{\langle\phi_0^{0+}(L)\phi_1^{++}(\tau)\phi_0^{+0}(0)\rangle}{\langle\phi_0^{0+}(L)\phi_0^{+0}(0)\rangle} = \tau^{\Delta_1^{++}}\frac{\langle\phi_1^{++}(\tau)\rangle_h}{\langle 1\rangle_h} \tag{D.99}$$

with $h(\tau) = h_0\theta(\tau)$. To leading order,

$$\frac{\langle\phi(\tau, 0)\rangle_h}{\langle 1\rangle_h} = \phi_{cl}(\tau, 0) = \frac{c(D)h_0}{(D-3)\tau^{D-3}}, \tag{D.100}$$

where we used Eq. (D.75). Setting $h_0$ to it's fixed point value (see footnote 26 for our sign convention) and using the normalization $\mathcal{N}_{++}$, we get

$$\lambda_{010}^{++0} = -\frac{3}{2\pi} + O(\epsilon). \tag{D.101}$$

We now proceed to compute the coefficient $b$ in Eq. (4.32), which may be extracted from the two-point function

$$\frac{\langle T_{\tau\tau}(\tau, \vec{x})\phi_1^{++}(0)\rangle_h}{\langle 1\rangle_h} \sim \frac{b}{g}\frac{1}{|\vec{x}|^{D-\Delta_1^{++}}\tau^{2\Delta_1^{++}}} + \ldots, \quad \vec{x}\to 0, \tag{D.102}$$

where from here on $h = h_0$ is uniform. Below we will denote $\langle\!\langle O\rangle\!\rangle = \frac{\langle O\rangle_h}{\langle 1\rangle_h}$. We write $\phi(x) = \phi_{cl}(x) + \eta(x)$, where $\phi_{cl}$ is given by Eq. (4.5),

$$\phi_{cl}(x) = -\frac{c(D)\sqrt{\pi}\Gamma((D-3)/2)}{\Gamma(D/2-1)}\frac{h_0}{|\vec{x}|^{D-3}}. \tag{D.103}$$

The action then takes the form in Eq. (D.60) (apart from the different expression for $\phi_{cl}(x)$) and

$$
\begin{aligned}
\langle\!\langle T_{\tau\tau}(x)\phi(0)\rangle\!\rangle = \Bigg\langle\!\!\Bigg\langle\!\!\Bigg\langle\! & \left[\frac{1}{2}(\partial_\tau\eta(x))^2 - \frac{1}{D-1}\partial_i\phi_{cl}(x)\partial_i\eta(x) - \frac{1}{2(D-1)}(\partial_i\eta(x))^2\right. \\
& -\frac{D-2}{2(D-1)}\phi_{cl}(x)\partial_\tau^2\eta(x) - \frac{D-2}{2(D-1)}\eta(x)\partial_\tau^2\eta(x) \\
& \left.\frac{\lambda_0(D-3)}{24(D-1)}\left(4\phi_{cl}^3(x)\eta(x) + 6\phi_{cl}^2(x)\eta^2(x) + 4\phi_{cl}(x)\eta^3(x) + \eta^4(x)\right)\right]\eta(0)\Bigg\rangle\!\!\Bigg\rangle\!\!\Bigg\rangle .
\end{aligned}
$$
(D.104)

To zeroth order in $\lambda$ we get:

$$
\langle\!\langle T_{\tau\tau}(x)\phi(0)\rangle\!\rangle_0 = -\frac{1}{D-1}\partial_i\phi_{cl}(x)\partial_i G(x) - \frac{D-2}{2(D-1)}\phi_{cl}(x)\partial_\tau^2 G(x).
$$
(D.105)

Both terms in the equation above diverge as $1/|\vec{x}|^{D-3}$ as $\vec{x}\to 0$, which is slower than the $\sim 1/|\vec{x}|^3$ divergence of Eq. (D.102). We conclude $b\sim O(\epsilon)$. We now proceed to evaluate Eq. (D.104) to $O(\lambda^1)$:

$$
\begin{aligned}
\langle\!\langle T_{\tau\tau}(x)\phi(0)\rangle\!\rangle_1 = {} & \frac{\lambda_0(D-3)}{6(D-1)}\phi_{cl}^3(x)G(x) - \frac{1}{D-1}\partial_i\phi_{cl}(x)\partial_i\langle\!\langle\eta(x)\eta(0)\rangle\!\rangle_1 - \frac{D-2}{2(D-1)}\phi_{cl}(x)\partial_\tau^2\langle\!\langle\eta(x)\eta(0)\rangle\!\rangle_1 \\
& -\frac{1}{D-1}\partial_i G(x)\partial_i\langle\!\langle\eta(x)\rangle\!\rangle_1 - -\frac{D-2}{2(D-1)}\partial_\tau^2 G(x)\langle\!\langle\eta(x)\rangle\!\rangle_1 \\
& +\frac{\lambda_0}{2}\int d^D y\,\phi_{cl}(y)G(y)\left[\frac{1}{D-1}(\partial_i G(x-y))^2 + \frac{D-2}{D-1}G(x-y)\partial_\tau^2 G(x-y) - (\partial_\tau G(x-y))^2\right].
\end{aligned}
$$
(D.106)

Since (up to logs), $\langle\eta(x)\rangle_1 \sim \frac{1}{|\vec{x}|}$ the two terms in the second line above diverge slower than (D.102) and do not contribute to $b$. The expression in square brackets in the last line vanishes identically. Thus, we only need to consider the contribution from the first line. We have,

$$
\langle\!\langle\eta(x)\eta(0)\rangle\!\rangle_1 = -\frac{\lambda_0}{2}\int d^D y\,\phi_{cl}^2(y)G(x-y)G(y).
$$
(D.107)

In $D=4$ the integral has a UV divergence as $y\to 0$ associated with the renormalization of the defect operator $\phi(0)$. This divergence is not of interest to us here since it does not lead to any singularities as $|\vec{x}|\to 0$. The most singular contribution in this limit comes from the region $|x-y|\ll|x_\tau|$. Setting $G(y)=G(x)$ in Eq. (D.107), integrating over $y_\tau$ and cutting off the $|\vec{y}|$ integral at $|\vec{y}|\sim|x_\tau|$,

$$
\langle\!\langle\eta(x)\eta(0)\rangle\!\rangle_1 \sim \frac{\lambda_r h_r^2}{32\pi^2}\log\left(\frac{|\vec{x}|}{|x_\tau|}\right)G(x).
$$
(D.108)

As a check of this result, we note that in DCFT, as $\vec{x}\to 0$,

$$
\sigma(\vec{x},\tau) \sim |\vec{x}|^{\Delta_1^{++}-\Delta_\sigma}\phi_1^{++}(\tau),
$$
(D.109)

and $\Delta_1^{++} - \Delta_\sigma \approx \frac{3\epsilon}{2}$. Now to first order in $\lambda$

$$\langle\langle\phi(x)\phi(0)\rangle\rangle_{0+1} \approx \left(1 + \frac{\lambda_r h_r^2}{32\pi^2} \log\left(\frac{|\vec{x}|}{|x_\tau|}\right)\right) G(x), \quad \vec{x} \to 0, \qquad (D.110)$$

which matches the exponent in Eq. (D.109) upon substituting the fixed point values of $\lambda_r, h_r$ in Eq. (4.4).

Returning to Eq. (D.106), we see that the third term on the first line is not singular enough as $\vec{x} \to 0$ to match Eq. (D.102), but the second term is. Combining it with the first term and taking into account the normalization of $\phi_1^{++}$, we obtain

$$b \approx -\frac{\lambda_{r,*} h_{r,*}^3}{576\pi^4} = \frac{\epsilon}{4\pi^2} + O(\epsilon^2). \qquad (D.111)$$

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
