# Peer review of "The beginning of the endpoint bootstrap for conformal line defects"

_SciPost Physics_

## Round 2 · Referee Report · Anonymous (Referee 1) · 2025-12-16

Strengths

See report.

Weaknesses

See report.

Report

In this paper the authors create and demonstrate a novel numerical conformal bootstrap setup for conformal defects, which incorporates correlation functions of bulk operators and defect-ending or defect-changing operators. This results in a positive semi-definite system that can be analyzed in the usual numerical conformal bootstrap framework, which is able to constrain defect CFT data, and encode information from the bulk CFT. This solves an important problem in the numerical study of conformal defects, where crossing equations of correlation functions either did not have the required positivity, only focused on defect operators without input from the bulk, or numerical bootstrap setups were only applicable to special examples of defects.

This bootstrap setup can be applied to endable defects. In this paper, the authors focus on the example of the pinning field defect in the 3d Ising CFT, and also draw conclusions about the stability of a $Z_2$-symmetric defect exhibiting long range order, which can be constructed out of two pinning field defects. The paper is complete and detailed, presenting both numerical bounds on various defect observables including the g-function and additional perturbative computations in $D = 4 - \varepsilon$, and comparing their results to state-of-the-art predictions from e.g. the fuzzy sphere in the literature. The paper is well written and well structured, and clearly and extensively describes the new numerical setup and assumptions that went into the obtained bounds.This is an important and novel step in the nonperturbative study of conformal defects.

I have a few suggestions and minor comments, as well as some questions out of curiosity. Although the list might appear long, they are mainly small points. I split them into requested changes, minor points and curiosities.

Requested changes

Requested changes

1) On page 4, “and others that have studied the properties of the pinning field defect more broadly [23, 29-33]”. In particular [Gimenez-Grau 2022], [Gimenez-Grau, Lauria, Liendo, van Vliet 2022], [Nishioka, Okuyama, Shimamori 2022] also study this defect in $d=4-\varepsilon$. Also, later on, “ … using so-called bulk-to-defect/boundary crossing symmetry…”, one should include [Meineri, Radhakrishnan 2025]. Lastly, right after “... while others have studied the consequences of the usual crossing symmetry adapted to defect/boundary operators”, some contributions that are missing are e.g. [Cavaglia, Gromov, Julius, Preti 2021] and [Bartlett-Tisdale, Herzog, Schaub 2023.]

2) In section 2.4, right above section 2.5: the g-function has also been bounded in the setup considered in [Meineri, Radhakrishnan 2025]. A comment would be in place.

3) In section 2.5 the analysis of bulk operators as SL(2,R) primaries is done for D = 3, specifying representations of $SO(3)$ and $SO(2)_T$. Can this discussion not be generalized to arbitrary D? Even though D = 3 is the case of interest for the numerics, the following paragraph discusses D = 2, and the perturbative section focuses on computations in $D = 4-\varepsilon$.

4) In section 3.1 on page 30: “ Next, we have assumed that the ratio of OPE coefficients $r_{\sigma \varepsilon}$ … is known exactly, …” in combination with eq. (3.5). It looks like that in the end, you input both the numerical values for $\lambda_{\varepsilon \varepsilon \varepsilon}$ and $\lambda_{\varepsilon \sigma \sigma}$ given in Table 4. However, the wording in this section 3.1 (“...remove the dependence on $\lambda_{\varepsilon \varepsilon \varepsilon}$... “) makes it look like you are treating these two OPE coefficients differently. Also, in the last paragraph of section 3.1: “... it is not possible to completely eliminate it …”. Why would you expect you could eliminate this OPE coefficient if it is a piece of CFT data present in the crossing equation?

5) In figure 13: you are only scanning over the values for $\Delta_{0}^{0+}$ which produce the island in figure 12. The caption of this figure could contain a comment that there is also a continent that is not shown here. Did you look at this continent and does it contain any robust features under increasing $\Lambda$, in contrast to figures appearing earlier in the paper? In addition, the comment about the robust kink in the zoomed-in plot in figure 13 that is now mentioned in the text could also be repeated in the caption so that it becomes immediately obvious to the reader while studying the figure why this region is highlighted.

6) Footnote 19 (page 41): what do you mean with “...not expect it to necessarily appear as a zero…”? If you include it in the discrete constraint, you explicitly exclude it from $V_{\Delta}^{0+}$. So you do not expect it to appear as a zero of $\alpha[V_{\Delta}^{0+}]$ at all, except if there is a degenerate operator. Did you end up seeing it appear as a zero or not? Furthermore, I would recommend not including it in the list of extracted low-lying spectra compared to fuzzy sphere predictions, because as you say, it is an external value which you choose and fix before extracting the spectrum from the extremal functional.

7) In section 4 you make several predictions for observables using a Padé approximation. While the paper is very detailed, here it seems there is minimal information about the calculation. At the top of page 50, two Padé approximations ([1,1] and [2,0]) are mentioned, but only one ([1,1]) result is actually given in the text. For other observables appearing earlier in this section, again only the [1,1] result is quoted. Why is this value chosen? Purely based on the agreement with fuzzy sphere results, or additionally because other Padés contain e.g. poles? I assume the Padé values quoted are for D=3, but this is not explicitly mentioned in the text. There is also inconsistent use of Padé and Pade (without the accent).

8) In section 4.3, the OPE coefficients $\lambda^{0+0}_{\sigma 00}$ and $\lambda^{0+0}_{\varepsilon 00}$ for $L \to \infty$ are the one-point functions of $\sigma, \varepsilon$ in the presence of a half-infinite line. Such one-point functions were computed perturbatively for the infinite pinning line defect in [Gimenez-Grau 2022]. Is there a relation between these one-point functions one can extract? Some integrals might also have been computed there already.

Minor points:

9) In section 2.2, on page 15: has such an example of an LRO defect been studied before? If so, where?

10) In section 2.3, eq. (2.5): define $Z^0$ as well.

11) In section 2.6, page 24, bottom line: the order $\tau_1 < \tau_2$ is flipped with respect to earlier conventions $\tau_i \geq \tau_{i+1}$.

12) On page 30, first equation, $\langle \phi^{+0} \phi^{0+} \phi^{+0} \phi^{0+}\rangle$ should this be $\pm$? If not, why are you not including $\langle \phi^{-0} \phi^{0-} \phi^{-0} \phi^{0-}\rangle$ but are including the mixed correlator $\langle \phi^{+0} \phi^{0-}\phi^{-0} \phi^{0+} \rangle$?

13) On page 38, on using the ratio of OPE coefficients as a continuous parameter. Has this been done before and could you provide a reference?

14) Figure 14: do the different colors have a certain meaning? Or are they just meant to better distinguish different lines?

Curiosities:

15) What about the pinning defect for O(N)? Generalizing the perturbative results should be straightforward, can you estimate how severely the reduction in precision of bulk CFT data is expected to affect the bounds? A small comment could be nice.

16) In section 2.2, on page 14: ".. a notion of SSB for line defects. We will first discuss the properties that we expect for such a line defect, and then show how we may construct ...". The properties are now discussed only for this specific construction. Are some of them valid for general SSB for line defects? Or is it hard to make general statements?

17) With regards to the continent in figure 12: you say there are no robust features with increasing $\Lambda$, but what about the kink around $\Delta_{0}^{0+} = 0.33?$ It seems to stay sharp for increasing $\Lambda$. Also, do you know of other theories that lie within the continent, not necessarily at the edge? Perhaps the pinning line defect in the Gross-Neveu-Yukawa CFT for example?

18) On page 33, middle: “ .. bulk $Z_2$ even tensor primaries with odd $\ell$ are kinematically excluded from these OPEs, and in principle our setup would be sensitive to such operators.” This could also be a feature. Did you check what happened if you did not a priori exclude them? Do the bounds just become weak or can you somehow rule them out or put gaps?

19) Did you do a spectrum extraction for the OPE maximization in section 3.4? If so, how did it compare to the spectrum extraction from g-minimization?

20) On page 38, at the bottom: could you think of a way to fix the uniqueness problem of $\sigma, \varepsilon$ for all components of the OPE vector given in eq. (3.21).

Recommendation

Ask for minor revision

---

## Editorial Decision

in_refereeing